# A Block Coordinate Descent Method for Nonsmooth Composite Optimization under Orthogonality Constraints

**Ganzhao Yuan**
Shenzhen University of Advanced Technology (SUAT), China
yuanganzhao@foxmail.com

## Abstract

Nonsmooth composite optimization with orthogonality constraints has a wide range of applications in statistical learning and data science. However, this problem is challenging due to its nonsmooth objective and computationally expensive, non-convex constraints. In this paper, we propose a new approach called **OBCD**, which leverages Block Coordinate Descent to address these challenges. **OBCD** is a feasible method with a small computational footprint. In each iteration, it updates $k$ rows of the solution matrix, where $k \geq 2$, by globally solving a small nonsmooth optimization problem under orthogonality constraints. We prove that the limiting points of **OBCD**, referred to as (global) block-$k$ stationary points, offer stronger optimality than standard critical points. Furthermore, we show that **OBCD** converges to $\epsilon$-block-$k$ stationary points with an iteration complexity of $\mathcal{O}(1/\epsilon)$. Additionally, under the Kurdyka-Lojasiewicz (KL) inequality, we establish the non-ergodic convergence rate of **OBCD**. We also demonstrate how novel breakpoint search methods can be used to solve the subproblem in **OBCD**. Empirical results show that our approach consistently outperforms existing methods.[1]

## 1 Introduction

We consider the following nonsmooth composite optimization problem under orthogonality constraints ('$\triangleq$' means define):

$$\min_{\mathbf{X} \in \mathbb{R}^{n \times r}} F(\mathbf{X}) \triangleq f(\mathbf{X}) + h(\mathbf{X}), \ s.t. \ \mathbf{X}^\mathsf{T}\mathbf{X} = \mathbf{I}_r. \tag{1}$$

Here, $n \geq r$, $n \geq 2$, and $\mathbf{I}_r$ is a $r \times r$ identity matrix. We do not assume convexity of $f(\mathbf{X})$ and $h(\mathbf{X})$. For brevity, the orthogonality constraints $\mathbf{X}^\mathsf{T}\mathbf{X} = \mathbf{I}_r$ in Problem (1) is rewritten as $\mathbf{X} \in \mathrm{St}(n,r) \triangleq \{\mathbf{X} \in \mathbb{R}^{n \times r} \mid \mathbf{X}^\mathsf{T}\mathbf{X} = \mathbf{I}_r\}$, where $\mathcal{M} \triangleq \mathrm{St}(n,r)$ is the Stiefel manifold in the literature (Edelman et al., 1998; Absil et al., 2008; Wen & Yin, 2013; Hu et al., 2020). We impose the following assumptions on Problem (1) throughout this paper. (Asm-i) For any $\mathbf{X}$ and $\mathbf{X}^+$, where $\mathbf{X}$ and $\mathbf{X}^+$ only differ at most by $k$ rows with $k \geq 2$, we assume $f : \mathbb{R}^{n \times r} \mapsto \mathbb{R}$ is differentiable and $\mathbf{H}$-smooth with $\mathbf{H} \in \mathbb{R}^{nr \times nr}$ such that:

$$f(\mathbf{X}^+) \leq \mathcal{Q}(\mathbf{X}^+; \mathbf{X}) \triangleq f(\mathbf{X}) + \langle \mathbf{X}^+ - \mathbf{X}, \nabla f(\mathbf{X}) \rangle + \tfrac{1}{2}\|\mathbf{X}^+ - \mathbf{X}\|_\mathbf{H}^2, \tag{2}$$

where $\|\mathbf{H}\|_{\mathsf{sp}} \leq L_f$ for some constant $L_f > 0$ and $\|\mathbf{X}\|_\mathbf{H}^2 \triangleq \mathrm{vec}(\mathbf{X})^\mathsf{T}\mathbf{H}\mathrm{vec}(\mathbf{X})$ [2]. Here, $\|\mathbf{H}\|_{\mathsf{sp}}$ is the spectral norm of $\mathbf{H}$. Notably, when $\mathbf{H} = L_f \cdot \mathbf{I}_{nr}$, this condition simplifies to the standard $L_f$-smoothness (Nesterov, 2003). (Asm-ii) The function $h(\mathbf{X}) : \mathbb{R}^{n \times r} \mapsto \mathbb{R}$ is proper, lower semicontinuous, and potentially non-smooth. Additionally, it is coordinate-wise separable, such that $h(\mathbf{X}) = \sum_{i,j} h(\mathbf{X}_{ij})$. Typical examples of $h(\mathbf{X})$ include the $\ell_p$ norm $h(\mathbf{X}) = \|\mathbf{X}\|_p$ with $p \in \{0,1\}$, the capped-$\ell_1$ function $h(\mathbf{X}) = \sum_{i,j} \max(|\mathbf{X}_{ij}|, \tau)$ with $\tau > 0$, and the indicator function

---

[1] Future versions of this paper can be found at `https://arxiv.org/abs/2304.03641`.
[2] Consider $f(\mathbf{X}) = \frac{1}{2}\mathrm{tr}(\mathbf{X}^\mathsf{T}\mathbf{C}\mathbf{X}\mathbf{D}) = \frac{1}{2}\|\mathbf{X}\|_\mathbf{H}^2$, where $\mathbf{H} = \mathbf{D} \otimes \mathbf{C}$, and $\mathbf{C} \in \mathbb{R}^{n \times n}$, $\mathbf{D} \in \mathbb{R}^{r \times r}$ are symmetric. Clearly, $f(\mathbf{X})$ satisfies (2) with equality, i.e., $f(\mathbf{X}^+) = \mathcal{Q}(\mathbf{X}^+; \mathbf{X})$ for all $\mathbf{X}$ and $\mathbf{X}^+$.

for non-negativity constraints $h(\mathbf{X}) = \iota_{\geq 0}(\mathbf{X})$. (Asm-iii) The following small-sized subproblem can be solved exactly and efficiently:

$$\min_{\mathbf{V} \in \mathrm{St}(k,k)} \mathcal{P}(\mathbf{V}) \triangleq \tfrac{1}{2}\|\mathbf{V}\|_{\tilde{\mathbf{Q}}}^2 + \langle \mathbf{V}, \mathbf{P} \rangle + h(\mathbf{VZ}) \tag{3}$$

for any given $\mathbf{Z} \in \mathbb{R}^{k \times r}$, $\mathbf{P} \in \mathbb{R}^{k \times k}$, and $\tilde{\mathbf{Q}} \in \mathbb{R}^{k^2 \times k^2}$. Here, we employ a notational simplification by defining $h(\mathbf{VZ}) \triangleq \sum_{i,j} h([\mathbf{VZ}]_{ij})$, given the coordinate-wise separability of $h(\cdot)$. This assumption is analogous to the "prox-friendly" condition in (variable-metric) proximal gradient methods (Beck & Teboulle, 2009; Raguet et al., 2013), but instead of evaluating a standard proximal operator for *a single nonsmooth term* in the *full* space, our subproblem jointly handles *two nonsmooth components* (the function $h(\cdot)$ and the orthogonality constraint) in a low-dimensional $k \times k$ space.

Problem (1) is an optimization framework that plays a crucial role in a variety of statistical learning and data science models, such as sparse Principal Component Analysis (PCA) (Journée et al., 2010; Shalit & Chechik, 2014), nonnegative PCA (Zass & Shashua, 2006; Qian et al., 2021), deep neural networks (Cogswell et al., 2016; Cho & Lee, 2017; Massart & Abrol, 2022; Huang & Gao, 2023), electronic structure calculation (Zhang et al., 2014; Liu et al., 2014), Fourier transforms approximation (Frerix & Bruna, 2019), orthogonal nonnegative matrix factorization (Jiang et al., 2022), $K$-indicators clustering (Jiang et al., 2016), and dictionary learning (Zhai et al., 2020).

## 1.1 MOTIVATING APPLICATIONS

Many machine learning and data science models can be cast as instances of Problem (1). Below, we present two representative examples: $L_0$-regularized sparse PCA and $L_1$-regularized sparse PCA. An additional example on nonnegative PCA is provided in Appendix Section G.1.

▶ $L_0$**-Regularized Sparse PCA**. $L_0$-regularized Sparse PCA (SPCA) is a method that uses $\ell_0$ norm to produce modified principal components with sparse loadings, which helps reduce model complexity and increase model interpretability (d'Aspremont et al., 2008; Chen et al., 2016). It can be formulated as: $\min_{\mathbf{X} \in \mathrm{St}(n,r)} -\langle \mathbf{X}, \mathbf{CX} \rangle + \lambda\|\mathbf{X}\|_0$, where $\mathbf{C} = \mathbf{A}^\mathsf{T}\mathbf{A} \in \mathbb{R}^{n \times n}$ is the covariance of the data matrix $\mathbf{A} \in \mathbb{R}^{m \times n}$ and $\lambda > 0$.

▶ $L_1$**-Regularized Sparse PCA**. As the $L_1$ norm provides the tightest convex relaxation for the $L_0$-norm over the unit ball in the sense of $L_\infty$-norm, some researchers replace the non-convex and discontinuous $L_0$ norm function with a convex but non-smooth function (Chen et al., 2016; Vu et al., 2013; Lu & Zhang, 2012). This leads to the following optimization problem of $L_1$-regularized SPCA: $\min_{\mathbf{X} \in \mathrm{St}(n,r)} -\langle \mathbf{X}, \mathbf{CX} \rangle + \lambda\|\mathbf{X}\|_1$, where $\mathbf{C} \in \mathbb{R}^{n \times n}$ is the covariance matrix of the data, and $\lambda > 0$.

## 1.2 RELATED WORK

We now present some related algorithms from the literature.

▶ **Minimizing Smooth Functions under Orthogonality Constraints.** One of the main challenges in solving Problem (1) stems from the nonconvexity of the orthogonality constraints. Existing approaches for addressing this difficulty can be broadly grouped into four classes: *(i)* Geodesic-like methods (Abrudan et al., 2008; Edelman et al., 1998; Absil et al., 2008). Computing exact geodesics typically involves solving ordinary differential equations, which can be computationally expensive. To avoid this, geodesic-like methods approximate the geodesic path by computing the geodesic logarithm using simpler linear algebraic operations. *(ii)* Projection-like methods (Absil et al., 2008; Golub & Van Loan, 2013; Jiang & Dai, 2015). These include techniques such as projection onto the nearest orthogonal matrix, polar decomposition, and QR-based projection. At each iteration, these methods descend along the Euclidean or Riemannian gradient direction and subsequently apply a projection step to enforce orthogonality. *(iii)* Multiplier correction methods (Gao et al., 2018; 2019; Xiao et al., 2022). These methods exploit the fact that the Lagrange multiplier associated with the orthogonality constraint is symmetric and admits a closed-form expression at first-order stationarity. They update the multiplier after achieving sufficient decrease in the objective, resulting in efficient feasible or infeasible first-order methods. *(iv)* Landing methods (Ablin & Peyré, 2022; Vary et al.; Ablin et al., 2024). These methods avoid explicit retractions by working in the ambient Euclidean space while adding a penalty that attracts iterates toward the orthogonal manifold. Each update combines a descent direction for the objective with a corrective term that reduces constraint violation,

and, with appropriate step sizes, the iterates converge to points that are nearly orthogonal and nearly stationary for the original problem.

▶ **Minimizing Nonsmooth Functions under Orthogonality Constraints.** Another major challenge in solving Problem (1) arises from the nonsmoothness of the objective function. Existing approaches for handling this issue can be broadly categorized into four classes: *(i)* Subgradient methods (Hwang et al., 2015; Li et al., 2021; Cheung et al., 2024). These methods generalize gradient descent to nonsmooth settings. Many of the previously mentioned geodesic-like and projection-based strategies can be incorporated into subgradient frameworks on manifolds. *(ii)* Proximal gradient methods (Chen et al., 2020; Li et al., 2024b; Lyu & Li, 2025). These methods compute a descent direction by solving a strongly convex subproblem over the tangent space, often using a semi-smooth Newton method. The resulting point is then mapped back onto the manifold via a retraction to preserve orthogonality. *(iii)* Block Majorization Minimization (BMM) on Riemannian manifolds (Li et al., 2024b; 2023; Breloy et al., 2021; Gutman & Ho-Nguyen, 2023). This class of methods iteratively constructs a tangential majorizing surrogate for a block of the objective, takes an approximate descent step in the corresponding tangent space, and retracts the iterate back to the manifold. *(iv)* Operator splitting methods (Lai & Osher, 2014; Chen et al., 2016; Zhang et al., 2019). These methods reformulate the original problem by introducing auxiliary variables and linear constraints, decomposing it into simpler subproblems that can be solved separately and often exactly. Prominent examples include the Alternating Direction Method of Multipliers (ADMM) (He & Yuan, 2012), Riemannian ADMM (RADMM) (Li et al., 2024a), and Penalty-based Splitting Method (PSM) (Yuan, 2024; Chen, 2012).

▶ **Block Coordinate Descent Methods.** (Block) coordinate descent is a classical and powerful algorithm that solves optimization problems by iteratively performing minimization along (block) coordinate directions (Tseng & Yun, 2009; Xu & Yin, 2013). The BCD methods have recently gained attention in solving nonconvex optimization problems, including sparse optimization (Yuan, 2024; Yuan et al., 2020), $k$-means clustering (Nie et al., 2022), structured nonconvex minimization (Yuan, 2023b;a), recurrent neural network (Massart & Abrol, 2022), and multi-layer convolutional networks (Bibi et al., 2019; Zeng et al., 2019). BCD methods have also been used in (Shalit & Chechik, 2014; Massart & Abrol, 2022) for solving optimization problems with orthogonal group constraints. However, their column-wise BCD methods are Limited to solving smooth minimization problems with $k = 2$ and $r = n$ (refer to Section 4.2 in (Shalit & Chechik, 2014)). Our row-wise BCD methods can solve coordinate-wise nonsmooth problems with $k \geq 2$ and $r \leq n$. The work of (Gao et al., 2019) proposes a parallelizable column-wise BCD scheme for solving the subproblems of their proximal linearized augmented Lagrangian algorithm. Impressive parallel scalability in a parallel environment of their algorithm is demonstrated. We stress that our **row-wise** BCD methods differ from the two **column-wise** counterparts.

▶ **Summary.** Existing methods typically suffer from one or more of the following limitations: *(i)* they rely on full gradient information, incurring high computational costs per iteration; *(ii)* they do not accommodate coordinate-wise nonsmooth composite objectives; *(iii)* they lack true descent properties and are often infeasible methods that only attain feasibility only in the limit; *(iv)* they often lack rigorous last-iterate convergence guarantees; *(v)* they provide only weak optimality results at critical points. ★ In contrast, our methods overcome these limitations by using a tailored block coordinate descent framework for efficient composite optimization on the Stiefel manifold, with strong optimality and convergence guarantees.

## 1.3 CONTRIBUTIONS AND NOTATIONS

This paper makes the following contributions. *(i)* Algorithmically: We propose a Block Coordinate Descent (BCD) algorithm tailored for nonsmooth composite optimization under orthogonality constraints (Section 2). *(ii)* Theoretically: We provide comprehensive optimality and convergence analyses of our methods (Sections 3 and 4). *(iii)* Empirically: Extensive experiments demonstrate that our methods surpass existing solutions in terms of accuracy and/or efficiency (Section 5).

We define $[n] \triangleq \{1, 2, ..., n\}$, and denote the Stiefel manifold as $\mathcal{M} \triangleq \mathrm{St}(n, r)$. Matlab-style colon notation is used to describe submatrices. For a matrix $\mathbf{X} \in \mathbb{R}^{n \times r}$, let $\mathrm{vec}(\mathbf{X}) \in \mathbb{R}^{nr \times 1}$ denote the vector formed by stacking its columns, and let $\mathrm{mat}(\mathbf{x}) \in \mathbb{R}^{n \times r}$ denote the inverse operator, such that $\mathrm{mat}(\mathrm{vec}(\mathbf{X})) = \mathbf{X}$. We denote $\mathbf{X} \otimes \mathbf{Y}$ as the Kronecker product of the matrices $\mathbf{X}$ and $\mathbf{Y}$. We

use $\mathbb{A} + \mathbb{B}$ and $\mathbb{A} - \mathbb{B}$ to denote standard Minkowski addition and subtraction between sets $\mathbb{A}$ and $\mathbb{B}$, and $\mathbb{A} \oplus \mathbb{B}$ and $\mathbb{A} \ominus \mathbb{B}$ to denote element-wise addition and subtraction, respectively. Additional notations are summarized in Appendix A.1.

## 2 THE PROPOSED **OBCD** ALGORITHM

In this section, we introduce **OBCD**, a Block Coordinate Descent algorithm for solving coordinate-wise nonsmooth composite problems under Orthogonality constraints, as defined in Problem (1).

We start by presenting a new update scheme designed to maintain the orthogonality constraint.

▶ **A New Constraint-Preserving Update Scheme**. For any partition of the index vector $[1, 2, ..., n]$ into $[\text{B}, \text{B}^c]$ with $\text{B} \in \mathbb{N}^k$, $\text{B}^c \in \mathbb{N}^{n-k}$, we define $\text{U}_\text{B} \in \mathbb{R}^{n \times k}$ and $\text{U}_{\text{B}^c} \in \mathbb{R}^{n \times (n-k)}$ as: $(\text{U}_\text{B})_{ji} = \left\{ \begin{array}{ll} 1, & \text{B}_i = j; \\ 0, & \text{else.} \end{array} \right.$, $(\text{U}_{\text{B}^c})_{ji} = \left\{ \begin{array}{ll} 1, & \text{B}_i^c = j; \\ 0, & \text{else.} \end{array} \right.$. Therefore, we have the following variable splitting for any $\mathbf{X} \in \mathbb{R}^{n \times r}$: $\mathbf{X} = \mathbf{I}_n \mathbf{X} = (\text{U}_\text{B} \text{U}_\text{B}^\mathsf{T} + \text{U}_{\text{B}^c} \text{U}_{\text{B}^c}^\mathsf{T}) \mathbf{X} = \text{U}_\text{B} \mathbf{X}(\text{B}, :) + \text{U}_{\text{B}^c} \mathbf{X}(\text{B}^c, :)$, where $\mathbf{X}(\text{B}, :) = \text{U}_\text{B}^\mathsf{T} \mathbf{X} \in \mathbb{R}^{k \times r}$ and $\mathbf{X}(\text{B}^c, :) = \text{U}_{\text{B}^c}^\mathsf{T} \mathbf{X} \in \mathbb{R}^{(n-k) \times r}$.

In each iteration $t$, the indices $\{1, 2, ..., n\}$ of the rows of decision variable $\mathbf{X} \in \text{St}(n, r)$ are separated to two sets $\text{B}$ and $\text{B}^c$, where $\text{B}$ is the working set with $|\text{B}| = k$ and $\text{B}^c = \{1, 2, ..., n\} \setminus \text{B}$. To simplify notation, we use $\text{B}$ instead of $\text{B}^t$, as $t$ can be inferred from the context. We only update $k$ rows of the variable $\mathbf{X}$ via $\mathbf{X}^{t+1}(\text{B}, :) \Leftarrow \mathbf{V} \mathbf{X}^t(\text{B}, :)$ for some appropriate matrix $\mathbf{V} \in \mathbb{R}^{k \times k}$. The following equivalent expressions hold:

$$\mathbf{X}^{t+1}(\text{B}, :) = \mathbf{V} \mathbf{X}^t(\text{B}, :) \quad \Leftrightarrow \quad \mathbf{X}^{t+1} = (\text{U}_\text{B} \mathbf{V} \text{U}_\text{B}^\mathsf{T} + \text{U}_{\text{B}^c} \text{U}_{\text{B}^c}^\mathsf{T}) \mathbf{X}^t \tag{4}$$

$$\Leftrightarrow \quad \mathbf{X}^{t+1} = \mathbf{X}^t + \text{U}_\text{B}(\mathbf{V} - \mathbf{I}_k) \text{U}_\text{B}^\mathsf{T} \mathbf{X}^t. \tag{5}$$

We consider the following minimization procedure to iteratively solve Problem (1):

$$\min_{\mathbf{V}} F(\mathcal{X}_\text{B}^t(\mathbf{V})), \ s.t. \mathcal{X}_\text{B}^t(\mathbf{V}) \in \text{St}(n, r), \text{ where } \mathcal{X}_\text{B}^t(\mathbf{V}) \triangleq \mathbf{X}^t + \text{U}_\text{B}(\mathbf{V} - \mathbf{I}_k) \text{U}_\text{B}^\mathsf{T} \mathbf{X}^t. \tag{6}$$

The following lemma shows that the orthogonality constraint for $\mathbf{X}^+ = \mathbf{X} + \text{U}_\text{B}(\mathbf{V} - \mathbf{I}_k) \text{U}_\text{B}^\mathsf{T} \mathbf{X}$ can be preserved by choosing suitable $\mathbf{V}$ and $\mathbf{X}$.

**Lemma 2.1.** *(Proof in Appendix D.1) We let $\text{B} \in \{\mathcal{B}_i\}_{i=1}^{\text{C}_n^k}$, where the set $\{\mathcal{B}_1, \mathcal{B}_2, ..., \mathcal{B}_{\text{C}_n^k}\}$ denotes all possible combinations of the index vectors choosing $k$ items from $n$ without repetition. We let $\mathbf{V} \in \text{St}(k, k)$. We define $\mathbf{X}^+ \triangleq \mathcal{X}_\text{B}(\mathbf{V}) \triangleq \mathbf{X} + \text{U}_\text{B}(\mathbf{V} - \mathbf{I}_k) \text{U}_\text{B}^\mathsf{T} \mathbf{X}$. (a) For any $\mathbf{X} \in \mathbb{R}^{n \times r}$, we have $[\mathbf{X}^+]^\mathsf{T} \mathbf{X}^+ = \mathbf{X}^\mathsf{T} \mathbf{X}$. (b) If $\mathbf{X} \in \text{St}(n, r)$, then $\mathbf{X}^+ \in \text{St}(n, r)$.*

Thanks to Lemma 2.1, we can now explore the following alternative formulation for Problem (6).

$$\bar{\mathbf{V}}^t \in \arg\min_{\mathbf{V}} F(\mathcal{X}_\text{B}^t(\mathbf{V})), \ s.t. \mathbf{V} \in \text{St}(k, k). \tag{7}$$

Then the solution matrix is updated via: $\mathbf{X}^{t+1} = \mathcal{X}_\text{B}^t(\bar{\mathbf{V}}^t)$.

The following lemma offers important properties for the update rule $\mathbf{X}^+ = \mathbf{X} + \text{U}_\text{B}(\mathbf{V} - \mathbf{I}_k) \text{U}_\text{B}^\mathsf{T} \mathbf{X}$.

**Lemma 2.2.** *(Proof in Appendix D.2) We define $\mathbf{X}^+ = \mathbf{X} + \text{U}_\text{B}(\mathbf{V} - \mathbf{I}_k) \text{U}_\text{B}^\mathsf{T} \mathbf{X}$. For any $\mathbf{X} \in \text{St}(n, r)$, $\mathbf{V} \in \text{St}(k, k)$, $\text{B} \in \{\mathcal{B}_i\}_{i=1}^{\text{C}_n^k}$, and symmetric matrix $\mathbf{H} \in \mathbb{R}^{nr \times nr}$, we have:*

*(a)* $\|\mathbf{X}^+ - \mathbf{X}\|_\mathbf{H}^2 = \|\mathbf{V} - \mathbf{I}_k\|_{\underline{\mathbf{Q}}}^2$, *where $\underline{\mathbf{Q}} \triangleq (\mathbf{Z}^\mathsf{T} \otimes \text{U}_\text{B})^\mathsf{T} \mathbf{H} (\mathbf{Z}^\mathsf{T} \otimes \text{U}_\text{B})$, and $\mathbf{Z} \triangleq \text{U}_\text{B}^\mathsf{T} \mathbf{X} \in \mathbb{R}^{k \times r}$.*

*(b)* $\|\mathbf{X}^+ - \mathbf{X}\|_\mathsf{F}^2 = 2\langle \mathbf{I}_k - \mathbf{V}, \text{U}_\text{B}^\mathsf{T} \mathbf{X} \mathbf{X}^\mathsf{T} \text{U}_\text{B} \rangle$.

*(c)* $\|\mathbf{X}^+ - \mathbf{X}\|_\mathsf{F}^2 \leq \|\mathbf{V} - \mathbf{I}_k\|_\mathsf{F}^2 = 2\langle \mathbf{I}_k, \mathbf{I}_k - \mathbf{V} \rangle$.

▶ **The Main Algorithm**. The proposed algorithm **OBCD** is an iterative procedure that sequentially minimizes the objective function along block coordinate directions within a sub-manifold of $\mathcal{M}$.

Starting with an initial feasible solution, **OBCD** iteratively determines a working set $\text{B}^t$ using specific strategies. It then solves the small-sized subproblem in Problem (7) through successive Majorization Minimization (MM). This method iteratively constructs a surrogate function that majorizes the objective function, driving it to decrease as expected (Mairal, 2013; Razaviyayn et al., 2013; Sun et al., 2016; Breloy et al., 2021), and it has proven effective for minimizing complex functions.

We now demonstrate how to derive the majorization function for $F(\mathcal{X}_{\mathtt{B}}^t(\mathbf{V}))$ in Problem (7). Initially, for any $\mathbf{X}^t \in \mathrm{St}(n, r)$ and $\mathbf{V} \in \mathrm{St}(k, k)$, we establish following inequalities: $f(\mathcal{X}_{\mathtt{B}}^t(\mathbf{V})) - f(\mathbf{X}^t) \overset{\textcircled{1}}{\leq}$ $\langle \mathcal{X}_{\mathtt{B}}^t(\mathbf{V}) - \mathbf{X}^t, \nabla f(\mathbf{X}^t) \rangle + \frac{1}{2} \|\mathcal{X}_{\mathtt{B}}^t(\mathbf{V}) - \mathbf{X}^t\|_{\mathbf{H}}^2 \overset{\textcircled{2}}{=} \langle \mathrm{U}_{\mathtt{B}}(\mathbf{V} - \mathbf{I}_k)\mathrm{U}_{\mathtt{B}}^{\mathsf{T}}\mathbf{X}^t, \nabla f(\mathbf{X}^t) \rangle + \frac{1}{2} \|\mathbf{V} - \mathbf{I}_k\|_{\underline{\mathbf{Q}}}^2 \overset{\textcircled{3}}{\leq}$ $\langle \mathbf{V} - \mathbf{I}_k, [\nabla f(\mathbf{X}^t)(\mathbf{X}^t)^{\mathsf{T}}]_{\mathtt{BB}} \rangle + \frac{1}{2} \|\mathbf{V} - \mathbf{I}_k\|_{\mathbf{Q}+\alpha\mathbf{I}}^2$, where step $\textcircled{1}$ uses Inequality (2); step $\textcircled{2}$ uses Lemma 2.2(a); step $\textcircled{3}$ uses $\alpha > 0$ and $\underline{\mathbf{Q}} \preceq \mathbf{Q}$, which can be ensured by choosing $\mathbf{Q}$ using one of the following methods:

$$\mathbf{Q} = \underline{\mathbf{Q}} \triangleq (\mathbf{Z}^{\mathsf{T}} \otimes \mathrm{U}_{\mathtt{B}})^{\mathsf{T}} \mathbf{H}(\mathbf{Z}^{\mathsf{T}} \otimes \mathrm{U}_{\mathtt{B}}), \tag{8}$$

$$\mathbf{Q} = \varsigma\mathbf{I}, \text{ with } \|\underline{\mathbf{Q}}\|_{\mathsf{sp}} \leq \varsigma \leq L_f. \tag{9}$$

where $\mathbf{Z} \triangleq \mathrm{U}_{\mathtt{B}}^{\mathsf{T}}\mathbf{X}^t$. Then, we apply the MM technique to the smooth function $f(\mathbf{X})$, while keeping the nonsmooth component $h(\mathbf{X})$ unchanged, leading to a function $\mathcal{K}(\mathbf{V}; \mathbf{X}^t, \mathtt{B})$ that majorizes $F(\mathcal{X}_{\mathtt{B}}^t(\mathbf{V})) = f(\mathcal{X}_{\mathtt{B}}^t(\mathbf{V})) + h(\mathcal{X}_{\mathtt{B}}^t(\mathbf{V}))$:

$$F(\mathcal{X}_{\mathtt{B}}^t(\mathbf{V})) \leq f(\mathbf{X}^t) + \langle \mathbf{V} - \mathbf{I}_k, [\nabla f(\mathbf{X}^t)(\mathbf{X}^t)^{\mathsf{T}}]_{\mathtt{BB}} \rangle + \frac{1}{2} \|\mathbf{V} - \mathbf{I}_k\|_{\mathbf{Q}+\alpha\mathbf{I}}^2 + h(\mathbf{V}\mathrm{U}_{\mathtt{B}}^{\mathsf{T}}\mathbf{X}^t)$$
$$\leq \underbrace{\frac{1}{2} \|\mathbf{V} - \mathbf{I}_k\|_{\mathbf{Q}+\alpha\mathbf{I}}^2 + \langle \mathbf{V}, [\nabla f(\mathbf{X}^t)(\mathbf{X}^t)^{\mathsf{T}}]_{\mathtt{BB}} \rangle + h(\mathbf{V}\mathrm{U}_{\mathtt{B}}^{\mathsf{T}}\mathbf{X}^t) + \ddot{c}}_{\triangleq \mathcal{K}(\mathbf{V}; \mathbf{X}^t, \mathtt{B})}, \tag{10}$$

where $\ddot{c} = f(\mathbf{X}^t) + h(\mathrm{U}_{\mathtt{B}^c}^{\mathsf{T}}\mathbf{X}^t) - \langle \mathbf{I}_k, [\nabla f(\mathbf{X}^t)(\mathbf{X}^t)^{\mathsf{T}}]_{\mathtt{BB}} \rangle$ is a constant. Here, we use the coordinate-wise separable property of $h(\cdot)$ as follows: $h(\mathcal{X}_{\mathtt{B}}^t(\mathbf{V})) = h(\mathrm{U}_{\mathtt{B}^c}\mathrm{U}_{\mathtt{B}^c}^{\mathsf{T}}\mathbf{X}^t + \mathrm{U}_{\mathtt{B}}\mathbf{V}\mathrm{U}_{\mathtt{B}}^{\mathsf{T}}\mathbf{X}^t) = h(\mathrm{U}_{\mathtt{B}^c}^{\mathsf{T}}\mathbf{X}^t) + h(\mathbf{V}\mathrm{U}_{\mathtt{B}}^{\mathsf{T}}\mathbf{X}^t)$. We minimize the upper bound of the right-hand side of Inequality (10), resulting in the minimization problem that $\bar{\mathbf{V}}^t \in \arg\min_{\mathbf{V}\in\mathrm{St}(k,k)} \mathcal{K}(\mathbf{V}; \mathbf{X}^t, \mathtt{B})$, which can be efficiently and exactly solved due to our assumption.

Two simple strategies to find the working set $\mathtt{B}$ with $|\mathtt{B}| = k$ can be considered. *(i)* Random strategy: $\mathtt{B}$ is randomly selected from $\{\mathcal{B}_1, \mathcal{B}_2, ..., \mathcal{B}_{\mathrm{C}_n^k}\}$ with equal probability $1/\mathrm{C}_n^k$. *(ii)* Cyclic strategy: $\mathtt{B}^t$ takes all possible combinations in cyclic order, such as $\mathcal{B}_1 \to \mathcal{B}_2 \to ... \to \mathcal{B}_{\mathrm{C}_n^k} \to \mathcal{B}_1 \to ....$

The proposed **OBCD** algorithm is summarized in Algorithm 1. Importantly, **OBCD** is a partial gradient method with low iterative computational complexity as it only assesses $k$ rows of the Euclidean gradient of $\nabla f(\mathbf{X}^t)$ and the solution $\mathbf{X}^t$ to compute the linear term $\langle [\nabla f(\mathbf{X}^t)(\mathbf{X}^t)^{\mathsf{T}}]_{\mathtt{BB}}, \mathbf{V} \rangle = \langle [\nabla f(\mathbf{X}^t)]_{\mathtt{B},:}^{\mathsf{T}}[\mathbf{X}^t]_{\mathtt{B},:}, \mathbf{V} \rangle$, as shown in Equation (10). Appendix C.3 details the complexity comparison between **OBCD** and full gradient methods for some quadratic function $f(\mathbf{X})$.

---

**Algorithm 1 OBCD**: Block Coordinate Descent for Problem (1)

---

1: **Input:** proximal parameter $\alpha > 0$, initial feasible point $\mathbf{X}^0$, block size $k \geq 2$, $t = 0$.
2: **for** $t = 0$ to $T$ **do**
3:     **(S1)** Select a working set $\mathtt{B}^t \in \{1, \ldots, n\}^k$. Denote $\mathtt{B} = \mathtt{B}^t$ for simplicity.
4:     **(S2)** Construct the matrix $\mathbf{Q} \in \mathbb{R}^{k^2 \times k^2}$ using (8) or (9).
5:     **(S3)** Define $\mathcal{K}(\cdot, ; \cdot, \cdot)$ as in Equation (10), and compute $\bar{\mathbf{V}}^t$ as the global minimizer:

$$\bar{\mathbf{V}}^t \in \arg\min_{\mathbf{V}\in\mathrm{St}(k,k)} \mathcal{K}(\mathbf{V}; \mathbf{X}^t, \mathtt{B}). \tag{11}$$

    (Alternatively, find a local solution $\bar{\mathbf{V}}^t$ such that $\mathcal{K}(\bar{\mathbf{V}}^t; \mathbf{X}^t, \mathtt{B}) \leq \mathcal{K}(\mathbf{I}_k; \mathbf{X}^t, \mathtt{B})$.)
6:     **(S4)** $\mathbf{X}^{t+1}(\mathtt{B}, :) \leftarrow \bar{\mathbf{V}}^t \mathbf{X}^t(\mathtt{B}, :)$
7: **end for**

---

▶ **Solving the General OBCD Subproblems**. The following lemma outlines key properties of the **OBCD** subproblems.

**Lemma 2.3.** *(Proof in Appendix D.3) We define* $\mathbf{Z} = \mathrm{U}_{\mathtt{B}}^{\mathsf{T}}\mathbf{X}^t$ *and* $\mathbf{P} \triangleq [\nabla f(\mathbf{X}^t)(\mathbf{X}^t)^{\mathsf{T}}]_{\mathtt{BB}} - \mathrm{mat}(\mathbf{Q}\mathrm{vec}(\mathbf{I}_k)) - \alpha\mathbf{I}_k$. *We have:*

*(a)* *The subproblem in Equation (11) reduces to Problem (3) with* $\tilde{\mathbf{Q}} = \mathbf{Q} + \alpha\mathbf{I}$.
*(b)* *Assume that Formula (9) is used to choose* $\mathbf{Q}$. *Problem (3) further reduces to the following problem:* $\bar{\mathbf{V}}^t \in \arg\min_{\mathbf{V}\in\mathrm{St}(k,k)} \mathcal{P}(\mathbf{V}) \triangleq \langle \mathbf{V}, \mathbf{P} \rangle + h(\mathbf{V}\mathbf{Z})$. *In particular, when* $h(\mathbf{X}) \triangleq 0$, *we obtain:* $\bar{\mathbf{V}}^t = -\mathbb{P}_{\mathcal{M}}(\mathbf{P})$. *Here,* $\mathbb{P}_{\mathcal{M}}(\mathbf{P})$ *is the nearest orthogonality matrix to* $\mathbf{P}$.

**Remark 2.4.** *(a) By Lemma 2.3(b), when $k > 2$, $h(\mathbf{X}) = 0$, and $\mathbf{Q}$ is chosen to be a diagonal matrix as in Equation (9), the subproblem $\bar{\mathbf{V}}^t \in \arg\min_{\mathbf{V} \in \mathrm{St}(k,k)} \mathcal{K}(\mathbf{V}; \mathbf{X}^t, \mathrm{B})$ in Algorithm 1 can be solved exactly and efficiently due to our assumption, see Remark 2.6. (b) For general $k$ and $h(\cdot)$, the subproblem may not admit a global solution. However, if a **local** stationary solution $\bar{\mathbf{V}}^t$ satisfying $\mathcal{K}(\bar{\mathbf{V}}^t; \mathbf{X}^t, \mathrm{B}) \leq \mathcal{K}(\mathbf{I}_k; \mathbf{X}^t, \mathrm{B})$ can be found, then the sufficient descent condition remains valid, and convergence to a weaker optimality condition for the final solution $\mathbf{X}^\infty$ is still achievable (see Inequalities (42), (44)).*

▶ **Smallest Possible Subproblems When** $k = 2$. We now discuss how to solve the subproblems exactly when $k = 2$. The following lemma reveals an equivalent expression for any $\mathbf{V} \in \mathrm{St}(2,2)$.

**Lemma 2.5.** *(Proof in Appendix D.4) Any orthogonal matrix $\mathbf{V} \in \mathrm{St}(2,2)$ can be expressed as $\mathbf{V} = \mathbf{V}_\theta^{\mathrm{rot}}$ or $\mathbf{V} = \mathbf{V}_\theta^{\mathrm{ref}}$ for some $\theta \in \mathbb{R}$, where $\mathbf{V}_\theta^{\mathrm{rot}} \triangleq \left( \begin{smallmatrix} \cos(\theta) & \sin(\theta) \\ -\sin(\theta) & \cos(\theta) \end{smallmatrix} \right)$, $\mathbf{V}_\theta^{\mathrm{ref}} \triangleq \left( \begin{smallmatrix} -\cos(\theta) & \sin(\theta) \\ \sin(\theta) & \cos(\theta) \end{smallmatrix} \right)$. We have $\det(\mathbf{V}_\theta^{\mathrm{rot}}) = 1$ and $\det(\mathbf{V}_\theta^{\mathrm{ref}}) = -1$ for any $\theta$.*

Using Lemma 2.5, we can reformulate Problem (3) as the following one-dimensional problem:

$$\bar{\theta} \in \arg\min_\theta \mathcal{P}(\mathbf{V}), \; s.t. \, \mathbf{V} \in \{\mathbf{V}_\theta^{\mathrm{rot}}, \mathbf{V}_\theta^{\mathrm{ref}}\}.$$

The optimal solution $\bar{\theta}$ can be identified even if $h(\cdot) \neq 0$ using a novel breakpoint searching method, which is discussed later in Section B in the Appendix.

**Remark 2.6.** *(i) $\mathbf{V}_\theta^{\mathrm{rot}}$ and $\mathbf{V}_\theta^{\mathrm{ref}}$ are called Givens rotation matrix and Jacobi reflection matrix respectively in the literature (Sun & Bischof, 1995). Previous research only considered $\{\mathbf{V}_\theta^{\mathrm{rot}}\}$ for solving symmetric linear eigenvalue problems (Golub & Van Loan, 2013) and sparse PCA problems (Shalit & Chechik, 2014), while we use $\{\mathbf{V}_\theta^{\mathrm{ref}}, \mathbf{V}_\theta^{\mathrm{rot}}\}$ for solving Problem (1). (ii) We show the necessity of using $\{\mathbf{V}_\theta^{\mathrm{ref}}, \mathbf{V}_\theta^{\mathrm{rot}}\}$ in the following two examples of $2 \times 2$ optimization problems with orthogonality constraints: $\min_{\mathbf{V} \in \mathrm{St}(2,2)} F(\mathbf{V}) \triangleq \|\mathbf{V} - \mathbf{A}\|_\mathsf{F}^2$, and $\min_{\mathbf{V} \in \mathrm{St}(2,2)} F(\mathbf{V}) \triangleq \|\mathbf{V} - \mathbf{B}\|_\mathsf{F}^2 + 5\|\mathbf{V}\|_1$, where $\mathbf{A} = \left( \begin{smallmatrix} 1 & 0 \\ -1 & -1 \end{smallmatrix} \right)$ and $\mathbf{B} = \left( \begin{smallmatrix} 1 & 0 \\ 1 & 2 \end{smallmatrix} \right)$. The use of the reflection matrix $\mathbf{V}_\theta^{\mathrm{ref}}$ is essential in these examples because it results in lower objective values. See Section C.1 in the Appendix for more details.*

## 3 Optimality Analysis

This section provides the optimality analysis for **OBCD**. First, we establish the completeness of the proposed update scheme, showing that **OBCD** can reach any feasible point from an arbitrary initialization. Second, we analyze the optimality conditions of both Problem (1) and the associated subproblems of **OBCD**. Finally, by comparing these two sets of conditions, we derive a hierarchy of optimality, illustrating how the algorithm's stationarity relates to that of Problem (1).

▶ **Basis Representation of Orthogonal Matrices**. The following theorem shows that any orthogonal matrix $\mathbf{D} \in \mathrm{St}(n, n)$ and any point $\mathbf{X} \in \mathrm{St}(n, r)$ can be generated by composing simple 2-dimensional updates.

**Theorem 3.1** (Basis Representation of Orthogonal Matrices). *(Proof in Appendix E.1) Assume $k = 2$. For all $i \in [\mathrm{C}_n^k]$, define $\mathcal{W}_i \triangleq \mathbf{I}_n + \mathbf{U}_{\mathcal{B}_i}(\mathcal{V}_i - \mathbf{I}_k)\mathbf{U}_{\mathcal{B}_i}^\mathsf{T} = \mathbf{U}_{\mathcal{B}_i}\mathcal{V}_i\mathbf{U}_{\mathcal{B}_i}^\mathsf{T} + \mathbf{U}_{\mathcal{B}_i^c}\mathbf{U}_{\mathcal{B}_i^c}^\mathsf{T}$, where $\mathcal{V}_i \in \mathrm{St}(k, k)$. Then:*

*(a) Any matrix $\mathbf{D} \in \mathrm{St}(n, n)$ can be expressed as $\mathbf{D} = \mathcal{W}_{\mathrm{C}_n^k}...\mathcal{W}_2\mathcal{W}_1$ for suitable choice of $\mathcal{W}_i$ (equivalently, of $\mathcal{V}_i$). Furthermore, if $\forall i, \mathcal{V}_i = \mathbf{I}_k$, then $\mathbf{D} = \mathbf{I}_n$.*

*(b) For any fixed reference point $\mathbf{X}^0 \in \mathrm{St}(n, r)$, every $\mathbf{X} \in \mathrm{St}(n, r)$ can be expressed as $\mathbf{X} = \mathcal{W}_{\mathrm{C}_n^k} \cdots \mathcal{W}_2\mathcal{W}_1\mathbf{X}^0$ for suitable $\mathcal{W}_i$.*

The above representation for $k = 2$ can in fact be extended to any block size $k \geq 2$, as stated next.

**Corollary 3.2.** *(Proof in Appendix E.2) The conclusion of Theorem 3.1 extends to all $k \geq 2$.*

**Remark 3.3.** *(i) We use both Givens rotation and Jacobi reflection matrices to compute $\mathbf{D} \in \mathrm{St}(n, n)$. This is necessary since a reflection matrix cannot be represented through a sequence of rotations. (ii) The result of Corollary 3.2 indicates that the proposed update scheme $\mathbf{X}^+ \Leftarrow \mathbf{X} + \mathbf{U}_\mathrm{B}(\mathbf{V} - \mathbf{I}_k)\mathbf{U}_\mathrm{B}^\mathsf{T}\mathbf{X}$ with $\mathbf{V} \in \mathrm{St}(k, k)$ as shown in Formula (5) can reach any orthogonal matrix $\mathbf{X} \in \mathrm{St}(n, r)$ for any starting solution $\mathbf{X}^0 \in \mathrm{St}(n, r)$.*

▶ **First-Order Optimality Conditions for Problem (1)**. We provide the first-order optimality condition of Problem (1) (Wen & Yin, 2013; Chen et al., 2020). We use $\partial F(\mathbf{X})$ to denote the limiting subdifferential of $F(\mathbf{X})$ (Mordukhovich, 2006; Rockafellar & Wets., 2009), which is always nonempty since $F(\mathbf{X})$ is closed, proper, and lower semicontinuous. Given $f(\mathbf{X})$ is differentiable, we have $\partial F(\mathbf{X}) = \partial(f + h)(\mathbf{X}) = \nabla f(\mathbf{X}) + \partial h(\mathbf{X})$ (Rockafellar & Wets., 2009). We extend the definition of *limiting subdifferential* to introduce $\partial_{\mathcal{M}} F(\mathbf{X})$ as the *Riemannian limiting subdifferential* of $F(\mathbf{X})$ at $\mathbf{X}$, defined as $\partial_{\mathcal{M}} F(\mathbf{X}) \triangleq \partial F(\mathbf{X}) \ominus (\mathbf{X}[\partial F(\mathbf{X})]^\mathsf{T} \mathbf{X})$, where $\ominus$ is the element-wise subtraction between sets.

Introducing a Lagrangian multiplier matrix $\boldsymbol{\Lambda} \in \mathbb{R}^{r \times r}$ for the orthogonality constraint, we define the following Lagrangian function of Problem (1): $\mathcal{L}(\mathbf{X}, \boldsymbol{\Lambda}) = F(\mathbf{X}) + \frac{1}{2}\langle \mathbf{I}_r - \mathbf{X}^\mathsf{T}\mathbf{X}, \boldsymbol{\Lambda}\rangle$. Notably, the matrix $\boldsymbol{\Lambda}$ is symmetric, as $\mathbf{X}^\mathsf{T}\mathbf{X}$ is symmetric. We state the following definition of first-order optimality condition.

**Definition 3.4.** *Critical Point (Wen & Yin, 2013; Chen et al., 2020). A solution $\check{\mathbf{X}} \in \mathrm{St}(n, r)$ is a critical point of Problem (1) if: $\mathbf{0} \in \partial_{\mathcal{M}} F(\check{\mathbf{X}}) \triangleq \partial F(\check{\mathbf{X}}) \ominus (\check{\mathbf{X}}[\partial F(\check{\mathbf{X}})]^\mathsf{T}\check{\mathbf{X}})$, where $(\partial F(\check{\mathbf{X}}) \ominus \check{\mathbf{X}}[\partial F(\check{\mathbf{X}})]^\mathsf{T}\check{\mathbf{X}}) \triangleq \{\mathbf{G} - \check{\mathbf{X}}\mathbf{G}^\mathsf{T}\check{\mathbf{X}} \mid \mathbf{G} \in \partial F(\check{\mathbf{X}})\}$. Moreover, the corresponding multiplier satisfies $\boldsymbol{\Lambda} \in [\partial F(\check{\mathbf{X}})]^\mathsf{T}\check{\mathbf{X}}$.*

**Remark 3.5.** *The critical point condition in Lemma 3.4 can be equivalently expressed as (Absil et al., 2008; Jiang & Dai, 2015; Liu et al., 2016): $\mathbf{0} \in \mathbb{P}_{\mathrm{T}_\mathbf{X}\mathcal{M}}(\partial F(\mathbf{X}))$. Here, $\mathrm{T}_\mathbf{X}\mathcal{M}$ is the tangent space to $\mathcal{M}$ at $\mathbf{X} \in \mathcal{M}$ with $\mathrm{T}_\mathbf{X}\mathcal{M} = \{\mathbf{Y} \in \mathbb{R}^{n \times r} \mid \mathbf{X}^\mathsf{T}\mathbf{Y} + \mathbf{Y}^\mathsf{T}\mathbf{X} = \mathbf{0}\}$.*

▶ **Optimality Conditions for the Subproblems**. The Euclidean subdifferential of $\mathcal{K}(\mathbf{V}; \mathbf{X}^t, \mathtt{B}^t)$ *w.r.t.* $\mathbf{V}$ is given by $\ddot{\mathbf{G}}(\mathbf{V}) \triangleq \ddot{\boldsymbol{\Delta}}(\mathbf{V}) + \mathrm{U}_\mathtt{B}^\mathsf{T}[\nabla f(\mathbf{X}^t) + \partial h(\mathbf{X}^{t+1})](\mathbf{X}^t)^\mathsf{T}\mathrm{U}_\mathtt{B}$, where $\ddot{\boldsymbol{\Delta}}(\mathbf{V}) = \mathrm{mat}((\mathbf{Q} + \alpha\mathbf{I}_k)\mathrm{vec}(\mathbf{V} - \mathbf{I}_k))$ and $\mathbf{X}^{t+1} = \mathbf{X}^t + \mathrm{U}_\mathtt{B}(\mathbf{V} - \mathbf{I}_k)\mathrm{U}_\mathtt{B}^\mathsf{T}\mathbf{X}^t$. Using Lemma 3.4, we set the Riemannian subdifferential of $\mathcal{K}(\mathbf{V}; \mathbf{X}^t, \mathtt{B}^t)$ *w.r.t.* $\mathbf{V}$ to zero and obtain the following first-order optimality condition for $\bar{\mathbf{V}}^t$: $\mathbf{0} \in \partial_{\mathcal{M}}\mathcal{K}(\bar{\mathbf{V}}^t; \mathbf{X}^t, \mathtt{B}^t) \triangleq \ddot{\mathbf{G}}(\bar{\mathbf{V}}^t) \ominus \bar{\mathbf{V}}^t\ddot{\mathbf{G}}(\bar{\mathbf{V}}^t)^\mathsf{T}\bar{\mathbf{V}}^t$. This inclusion is a key ingredient in establishing the optimality hierarchy in Theorem 3.8(a) and the Riemannian subgradient lower bound in Lemma 4.4(a).

▶ **Optimality Conditions and Their Hierarchy**. We introduce the following new optimality condition of block-$k$ stationary points.

**Definition 3.6.** *(Global) Block-k Stationary Point, abbreviated as $\mathrm{BS}_k$-point. Let $\alpha > 0$ and $k \geq 2$. A solution $\ddot{\mathbf{X}} \in \mathrm{St}(n, r)$ is called a block-k stationary point if: $\forall \mathtt{B} \in \{\mathcal{B}_i\}_{i=1}^{\mathrm{C}_n^k}$, $\mathbf{I}_k \in \arg\min_{\mathbf{V} \in \mathrm{St}(k,k)} \mathcal{K}(\mathbf{V}; \ddot{\mathbf{X}}, \mathtt{B})$, where $\mathcal{K}(\cdot; \cdot, \cdot)$ is defined in Equation (10).*

**Remark 3.7.** *$\mathrm{BS}_k$-point states that if we globally minimize the majorization function $\mathcal{K}(\mathbf{V}; \ddot{\mathbf{X}}, \mathtt{B})$, there is no possibility of improving the objective function value for $\mathcal{K}(\mathbf{V}; \ddot{\mathbf{X}}, \mathtt{B})$ across all $\mathtt{B} \in \{\mathcal{B}_i\}_{i=1}^{\mathrm{C}_n^k}$.*

The following theorem establishes the relation between $\mathrm{BS}_k$-points, standard critical points, and global optimal points.

**Theorem 3.8.** *(Proof in Appendix E.3) We establish the following relationships:*

*(a)* $\{\text{critical points } \check{\mathbf{X}}\} \supseteq \{\mathrm{BS}_2\text{-points } \ddot{\mathbf{X}}\}$.

*(b)* $\{\mathrm{BS}_k\text{-points } \ddot{\mathbf{X}}\} \supseteq \{\text{global optimal points } \bar{\mathbf{X}}\}$, *where $k \in \{2, 3, \ldots, n\}$.*

*(c)* $\{\mathrm{BS}_k\text{-points } \ddot{\mathbf{X}}\} \supseteq \{\mathrm{BS}_{k+1}\text{-points } \ddot{\mathbf{X}}\}$, *where $k \in \{2, 3, \ldots, n-1\}$.*

*(d)* *The reverse of the above three inclusions may not always hold true.*

**Remark 3.9.** *(i) The optimality of $\mathrm{BS}_2$-points is stronger than that of standard critical points (Wen & Yin, 2013; Chen et al., 2020; Absil et al., 2008). (ii) Testing whether a solution $\mathbf{X}$ is a $\mathrm{BS}_k$-points deterministically requires solving all $\mathrm{C}_n^k$ subproblems. However, by randomly selecting the working set $\mathtt{B}$ from the $\mathrm{C}_n^k$ possible combinations $\{\mathcal{B}_i\}_{i=1}^{\mathrm{C}_n^k}$, one can test whether $\mathbf{X}$ is a $\mathrm{BS}_k$-point in expectation.*

## 4 CONVERGENCE ANALYSIS

This section establishes the iteration complexity and non-ergodic (last-iterate) convergence rates of the proposed **OBCD** algorithm. We first prove a sufficient descent property, followed by an ergodic convergence rate typical in nonconvex optimization. We then analyze iteration complexity under the Riemannian subgradient condition, commonly used in nonsmooth manifold settings. Finally, we derive a last-iterate convergence rate based on the KL inequality.

Throughout this section, we assume that the working set is determined by a random strategy and that the global minimizer $\bar{\mathbf{V}}^t \in \arg\min_{\mathbf{V} \in \mathrm{St}(k,k)} \mathcal{K}(\mathbf{V}; \mathbf{X}^t, \mathtt{B}^t)$ can be computed. The algorithm **OBCD** then generates a random output $(\bar{\mathbf{V}}^t, \mathbf{X}^{t+1})$ for $t = 0, 1, \ldots, \infty$, depending on the realization of the random variable $\xi^t \triangleq (\mathtt{B}^1, \mathtt{B}^2, \ldots, \mathtt{B}^t)$. We denote $\mathbf{X}^\infty$ as an arbitrary limit point of **OBCD**.

### 4.1 ITERATION COMPLEXITY

Initially, we introduce the notation of $\epsilon$-BS$_k$-*point* as follows.

**Definition 4.1.** *($\epsilon$-BS$_k$-point)* Given any constant $\epsilon > 0$, a point $\ddot{\mathbf{X}}$ is called an $\epsilon$-BS$_k$-*point* if: $\frac{1}{\mathrm{C}_n^k} \sum_{i=1}^{\mathrm{C}_n^k} \mathrm{dist}(\mathbf{I}_k, \arg\min_{\mathbf{V}} \mathcal{K}(\mathbf{V}; \ddot{\mathbf{X}}, \mathcal{B}_i))^2 \le \epsilon$, where $\mathcal{K}(\cdot; \cdot, \cdot)$ is defined in Equation (10). Here, the set $\{\mathcal{B}_1, \mathcal{B}_2, ..., \mathcal{B}_{\mathrm{C}_n^k}\}$ denotes all possible combinations of the index vectors choosing $k$ items from $n$ without repetition, and $\mathrm{dist}(\Xi, \Xi')$ denotes the distance between two sets $\Xi$ and $\Xi'$.

Using the optimality measure from Definition 4.1, we establish the iteration complexity of **OBCD**.

**Theorem 4.2.** *(Proof in Appendix F.1) We define $\tilde{c} \triangleq \frac{2}{\alpha} \cdot (F(\mathbf{X}^0) - F(\mathbf{X}^\infty)) \ge 0$. We have:*

(a) *The following sufficient decrease condition holds for all $t \ge 0$:*
$$\tfrac{\alpha}{2}\|\mathbf{X}^{t+1} - \mathbf{X}^t\|_{\mathsf{F}}^2 \le \tfrac{\alpha}{2}\|\bar{\mathbf{V}}^t - \mathbf{I}_k\|_{\mathsf{F}}^2 \le F(\mathbf{X}^t) - F(\mathbf{X}^{t+1}).$$

(b) *If the $\mathtt{B}^t$ is selected from $\{\mathcal{B}_i\}_{i=1}^{\mathrm{C}_n^k}$ randomly and uniformly, **OBCD** finds an $\epsilon$-BS$_k$-point of Problem (1) in at most $T$ iterations in the sense of expectation, where $T \ge \lceil \frac{\tilde{c}}{\epsilon} \rceil$.*

**Remark 4.3.** *Theorem 4.2 shows that **OBCD** converges to $\epsilon$-block-$k$ stationary points with an iteration complexity of $\mathcal{O}(1/\epsilon)$, which is typical for general nonconvex optimization.*

Apart from Definition 4.1, another common optimality measure relies on the Riemannian subgradient. At the point $\mathbf{V} = \mathbf{I}_k$, the Riemannian subdifferential of $\mathcal{K}(\mathbf{V}; \mathbf{X}^t, \mathtt{B}^t)$ is $\partial_{\mathcal{M}} \mathcal{K}(\mathbf{I}_k; \mathbf{X}^t, \mathtt{B}^t) = \mathrm{U}_{\mathtt{B}^t}^{\mathsf{T}} (\mathbb{D} \ominus \mathbb{D}^{\mathsf{T}}) \mathrm{U}_{\mathtt{B}^t}$, where $\mathbb{D} = [\nabla f(\mathbf{X}^t) + \partial h(\mathbf{X}^t)][\mathbf{X}^t]^{\mathsf{T}}$. We next derive a Riemannian subgradient lower bound in terms of the iterate gap.

**Lemma 4.4.** *(Proof in Appendix F.2, **Riemannian Subgradient Lower Bound for the Iterates Gap**) Assume that $F(\cdot)$ is $C_F$-Lipschitz continuous on $\mathrm{St}(n,r)$, i.e., $\|\mathbf{G}\|_{\mathsf{F}} \le C_F$ for all $\mathbf{X} \in \mathrm{St}(n,r)$ and all $\mathbf{G} \in \partial F(\mathbf{X})$. We have:*

(a) $\mathbb{E}_{\xi^{t+1}}[\mathrm{dist}^2(\mathbf{0}, \partial_{\mathcal{M}} \mathcal{K}(\mathbf{I}_k; \mathbf{X}^{t+1}, \mathtt{B}^{t+1}))] \le \phi \cdot \mathbb{E}_{\xi^t}[\|\bar{\mathbf{V}}^t - \mathbf{I}_k\|_{\mathsf{F}}^2]$, *where $\phi \triangleq 72(C_F^2 + \alpha^2 + L_f^2)$.*

(b) $\mathbb{E}_{\xi^t}[\mathrm{dist}^2(\mathbf{0}, \partial_{\mathcal{M}} F(\mathbf{X}^t))] \le \gamma \cdot \mathbb{E}_{\xi^t}[\mathrm{dist}^2(\mathbf{0}, \partial_{\mathcal{M}} \mathcal{K}(\mathbf{I}_k; \mathbf{X}^t, \mathtt{B}^t))]$, *where $\gamma \triangleq \mathrm{C}_n^k / \mathrm{C}_{n-2}^{k-2}$.*

**Remark 4.5.** *The important class of nonsmooth $\ell_1$ norm function $h(\mathbf{X}) = \|\mathbf{X}\|_1$ (Chen et al., 2020; 2024) satisfies the assumption made in Lemma 4.4.*

We establish the iteration complexity of **OBCD** using the optimality measure of Riemannian subgradient (Chen et al., 2020; Cheung et al., 2024; Li et al., 2024b).

**Theorem 4.6.** *(Proof in Appendix F.3) We define $\tilde{c}$ as in Theorem 4.2 and $\{\phi, \gamma\}$ as in Lemma 4.4.* **OBCD** *finds an $\epsilon$-critical point of Problem (1), i.e., $\mathbb{E}_{\xi^{\bar{t}}}[\mathrm{dist}^2(\mathbf{0}, \partial_{\mathcal{M}} F(\mathbf{X}^{\bar{t}+1}))] \le \epsilon$, in at most $T + 1$ iterations in expectation, where $\bar{t} \in [T]$ and $T \ge \lceil \frac{\gamma\phi\tilde{c}}{\epsilon} \rceil$.*

### 4.2 CONVERGENCE RATE UNDER KL INEQUALITY

We establish the non-ergodic convergence rate of **OBCD** using the Kurdyka-Łojasiewicz inequality, a key tool in non-convex analysis (Attouch et al., 2010; Bolte et al., 2014; Liu et al., 2016).

Initially, we make the following additional assumption.

**Assumption 4.7.** *The function $F_\iota(\mathbf{X}) = F(\mathbf{X}) + \iota_{\mathcal{M}}(\mathbf{X})$ is a Kurdyka-Łojasiewicz (KL) function.*

**Remark 4.8.** *Semi-algebraic functions constitute a broad class of KL functions, including real polynomials, norm functions $\|\mathbf{x}\|_p$ with $p \geq 0$, rank functions, and indicator functions of sets such as the Stiefel manifold and the positive semidefinite cone (Attouch et al., 2010).*

We present the following useful proposition regrading to the KL function.

**Proposition 4.9.** *(**Kurdyka-Łojasiewicz Property**, see, e.g.,(Attouch et al., 2010; Bolte et al., 2014)). Let $F_\iota : \mathbb{R}^{m \times n} \to (-\infty, +\infty]$ be a KL function and $\mathbf{X}^\infty \in \operatorname{dom} F_\iota$. Then there exist $\sigma \in [0, 1)$, $\eta \in (0, +\infty]$, a neighborhood $\Upsilon$ of $\mathbf{X}^\infty$, and a concave continuous function $\varphi(t) = ct^{1-\sigma}$ with $c > 0$ and $t \in [0, \eta)$ such that for all $\mathbf{X}' \in \Upsilon$ satisfying $F_\iota(\mathbf{X}') \in (F_\iota(\mathbf{X}^\infty), F_\iota(\mathbf{X}^\infty) + \eta)$, it holds that $\operatorname{dist}(\mathbf{0}, \partial F_\iota(\mathbf{X}')) \, \varphi'\big(F_\iota(\mathbf{X}') - F_\iota(\mathbf{X}^\infty)\big) \geq 1$.*

Utilizing the Kurdyka-Łojasiewicz property, one can establish a finite-length property of **OBCD**, a result considerably stronger than that of Theorem 4.2.

**Theorem 4.10.** *(Proof in Appendix F.4, **A Finite Length Property**). We define $E_{t+1} \triangleq \big(\mathbb{E}_{\xi^t}[\|\bar{\mathbf{V}}^t - \mathbf{I}_k\|_{\mathsf{F}}]\big)^{1/2}$, and $D_i = \sum_{j=i}^\infty E_{j+1}$. Under the continuity assumption in Lemma 4.4, there exists a sufficiently large $t_\star$ such that, for all $t \geq t_\star$, we have*

(a) *It holds that $(E_{t+1})^2 \leq \kappa E_t(\varphi_t - \varphi_{t+1})$, where $\varphi_t \triangleq \varphi(F(\mathbf{X}^t) - F(\mathbf{X}^\infty))$, $\kappa \triangleq \frac{2\sqrt{\gamma\phi}}{\alpha}$ is a positive constant, $\gamma$ and $\phi$ are defined in Lemma 4.4, and $\varphi(\cdot)$ is the desingularization function defined in Proposition 4.9.*

(b) *It holds that $\sum_{j=t}^\infty E_{j+1} \leq E_t + 2\kappa\varphi_t$. The sequence $\{E_t\}_{t=1}^\infty$ has the finite length property that $D_t \triangleq \sum_{j=t}^\infty E_{j+1}$ is always upper-bounded by a certain constant for all $t \geq t_\star$.*

Finally, we establish the last-iterate convergence rate for **OBCD**.

**Theorem 4.11.** *(Proof in Appendix F.5). Based on the continuity assumption made in Lemma 4.4, for all $t \geq t_\star$, we have:*

(a) *If $\sigma = 0$, then the sequence $\mathbf{X}^t$ converges in a finite number of steps in expectation.*

(b) *If $\sigma \in (0, \frac{1}{2}]$, then there exist $\dot{c} > 0$ and $\dot{\tau} \in [0, 1)$ such that $\mathbb{E}_{\xi^{t-1}}[\|\mathbf{X}^t - \mathbf{X}^\infty\|_{\mathsf{F}}] \leq \dot{c}\dot{\tau}^t$.*

(c) *If $\sigma \in (\frac{1}{2}, 1)$, then there exist $\dot{c} > 0$ such that $\mathbb{E}_{\xi^{t-1}}[\|\mathbf{X}^t - \mathbf{X}^\infty\|_{\mathsf{F}}] \leq \frac{\dot{c}}{t^{\dot{\tau}}}$, where $\dot{\tau} \triangleq \frac{1-\sigma}{2\sigma-1} > 0$.*

**Remark 4.12.** *When $F(\mathbf{X})$ is a semi-algebraic function and the desingularising function is $\varphi(t) = ct^{1-\sigma}$ for some $c > 0$ and $\sigma \in [0, 1)$, Theorem 4.11 shows that **OBCD** converges in finite iterations when $\sigma = 0$, with linear convergence when $\sigma \in (0, \frac{1}{2}]$, and sublinear convergence when $\sigma \in (\frac{1}{2}, 1)$ for the gap $\|\mathbf{X}^t - \mathbf{X}^\infty\|_{\mathsf{F}}$ in expectation. These results are consistent with those in (Attouch et al., 2010).*

## 5 EXPERIMENTS

This section presents numerical comparisons between **OBCD** and state-of-the-art methods on both real-world and synthetic data. We describe the application of $L_0$-regularized SPCA in the sequel, while additional applications for $L_1$-regularized SPCA and nonnegative PCA can be found in Appendix Section G.2.

▶ **Compared Methods on $L_0$-Regularized SPCA**. We compare against three operator splitting methods: Linearized ADMM (LADMM) (Lai & Osher, 2014; He & Yuan, 2012), Riemannian AD-MM (RADMM) (Li et al., 2024a), and the Penalty-based Splitting Method (PSM) (Yuan, 2024; Chen, 2012). Each method is initialized with either a random or identity matrix, yielding six variants: LADMM(id), RADMM(id), SPM(id), LADMM(rnd), RADMM (rnd), and PSM(rnd). For **OBCD**, we adopt a random working set strategy with identity initialization, denoted as **OBCD-R**(id).

▶ **Implementations**. All methods are implemented in MATLAB on an Intel 2.6 GHz CPU with 32 GB RAM. However, our breakpoint searching procedure is developed in C++ and integrated into the MATLAB environment [3], as it requires inefficient element-wise loops in native MATLAB. The

---

[3]Although we prioritize accuracy over speed, the comparisons remain fair, as the other methods based on matrix multiplication and SVD rely on highly optimized BLAS and LAPACK libraries.

| data-m-n | LADMM (id) | RADMM (id) | SPM (id) | LADMM (rnd) | RADMM (rnd) | SPM (rnd) | OBCD-R (id) |
|---|---|---|---|---|---|---|---|
| | | | $r = 20, \lambda = 10$, time limit=40 | | | | |
| w1a-2477-300 | 199.897 | 219.698 | 199.897 | 259.825 | 239.717 | 259.672 | 199.667 |
| TDT2-500-1000 | 199.997 | 359.382 | 199.997 | 389.376 | 269.292 | 389.260 | 199.258 |
| 20News-8000-1000 | 199.995 | 219.673 | 199.995 | 239.301 | 219.243 | 349.228 | 199.222 |
| sector-6412-1000 | 199.980 | 349.793 | 199.980 | 749.996 | 249.813 | 369.651 | 199.649 |
| E2006-2000-1000 | 199.999 | 239.115 | 199.999 | 269.128 | 219.084 | 709.095 | 199.077 |
| MNIST-60000-784 | 199.985 | 379.893 | 199.985 | 289.917 | 339.910 | 1339.774 | 199.896 |
| Gisette-3000-1000 | 199.980 | 339.979 | 199.980 | 539.979 | 369.981 | 1639.952 | 199.979 |
| CnnCal-3000-1000 | 199.981 | 429.979 | 199.981 | 689.970 | 379.979 | 909.931 | 199.946 |
| Cifar-1000-1000 | 199.979 | 479.979 | 199.979 | 1449.982 | 429.975 | 2169.934 | 199.974 |
| randn-500-1000 | 199.980 | 469.980 | 199.980 | 409.980 | 389.980 | 1349.975 | 199.977 |

| data-m-n | LADMM (id) | RADMM (id) | SPM (id) | LADMM (rnd) | RADMM (rnd) | SPM (rnd) | OBCD-R (id) |
|---|---|---|---|---|---|---|---|
| | | | $r = 20, \lambda = 50$, time limit=40 | | | | |
| w1a-2477-300 | 999.891 | 1099.730 | 1099.889 | 1249.723 | 1049.707 | 1649.675 | 999.667 |
| TDT2-500-1000 | 1049.997 | 1099.288 | 999.460 | 1049.282 | 1249.280 | 2149.271 | 999.257 |
| 20News-8000-1000 | 1149.995 | 1149.501 | 999.549 | 3649.247 | 1049.326 | 1799.228 | 999.222 |
| sector-6412-1000 | 2449.886 | 1799.904 | 999.816 | 1549.998 | 1749.952 | 1399.651 | 999.649 |
| E2006-2000-1000 | 999.283 | 1249.109 | 999.284 | 1849.115 | 1349.085 | 2549.136 | 999.077 |
| MNIST-60000-784 | 999.985 | 1699.913 | 2849.852 | 1399.921 | 1649.905 | 4349.781 | 999.896 |
| Gisette-3000-1000 | 999.980 | 1649.980 | 999.980 | 10399.983 | 2249.976 | 6899.967 | 999.979 |
| CnnCal-3000-1000 | 999.981 | 2499.981 | 1049.969 | 4599.973 | 2649.981 | 3499.938 | 999.946 |
| Cifar-1000-1000 | 999.979 | 1449.978 | 999.979 | 2149.979 | 3149.974 | 4349.972 | 999.974 |
| randn-500-1000 | 1349.980 | 2449.980 | 3949.977 | 1299.981 | 1749.980 | 4249.976 | 999.977 |

| data-m-n | LADMM (id) | RADMM (id) | SPM (id) | LADMM (rnd) | RADMM (rnd) | SPM (rnd) | OBCD-R (id) |
|---|---|---|---|---|---|---|---|
| | | | $r = 20, \lambda = 100$, time limit=40 | | | | |
| w1a-2477-300 | 2499.912 | 2799.713 | 2199.819 | 2399.723 | 2499.708 | 3299.662 | 1999.667 |
| TDT2-500-1000 | 2199.515 | 2199.302 | 1999.432 | 8799.310 | 2699.278 | 2499.257 | 1999.258 |
| 20News-8000-1000 | 2699.480 | 2199.262 | 1999.440 | 2099.242 | 1999.230 | 3999.224 | 1999.222 |
| sector-6412-1000 | 7799.995 | 4599.977 | 2099.716 | 3099.999 | 4399.973 | 2199.651 | 1999.649 |
| E2006-2000-1000 | 2099.207 | 3199.083 | 1999.284 | 2599.106 | 2299.085 | 4399.081 | 1999.077 |
| MNIST-60000-784 | 1999.984 | 3199.904 | 11799.715 | 3199.922 | 3599.907 | 8299.829 | 1999.896 |
| Gisette-3000-1000 | 2199.980 | 4299.979 | 1999.980 | 2499.982 | 2799.981 | 11499.971 | 1999.979 |
| CnnCal-3000-1000 | 2499.981 | 4399.982 | 11499.907 | 4399.975 | 3899.983 | 6799.938 | 1999.946 |
| Cifar-1000-1000 | 1999.979 | 4999.979 | 1999.979 | 5199.979 | 4399.978 | 8799.969 | 1999.974 |
| randn-500-1000 | 6699.980 | 4099.980 | 7899.977 | 2599.980 | 3299.980 | 9099.976 | 1999.977 |

| data-m-n | LADMM (id) | RADMM (id) | SPM (id) | LADMM (rnd) | RADMM (rnd) | SPM (rnd) | OBCD-R (id) |
|---|---|---|---|---|---|---|---|
| | | | $r = 20, \lambda = 500$, time limit=40 | | | | |
| w1a-2477-300 | 11999.706 | 10999.702 | 16499.714 | 10499.702 | 9999.711 | 14499.667 | 9999.667 |
| TDT2-500-1000 | 10499.273 | 15999.294 | 10999.395 | 10499.368 | 15499.281 | 12499.256 | 9999.258 |
| 20News-8000-1000 | 9999.347 | 11499.281 | 11499.328 | 10999.454 | 10499.258 | 14499.232 | 9999.222 |
| sector-6412-1000 | 13999.997 | 16999.992 | 12999.660 | 22999.999 | 18999.986 | 13499.649 | 9999.649 |
| E2006-2000-1000 | 9999.918 | 14499.080 | 9999.284 | 26499.095 | 10499.082 | 21499.081 | 9999.077 |
| MNIST-60000-784 | 19499.965 | 20499.886 | 39499.844 | 11999.911 | 16999.905 | 47999.705 | 9999.896 |
| Gisette-3000-1000 | 14999.980 | 16499.979 | 9999.980 | 15499.980 | 16999.978 | 36499.977 | 9999.979 |
| CnnCal-3000-1000 | 12499.980 | 33999.979 | 28999.936 | 15499.974 | 52999.977 | 26999.936 | 9999.946 |
| Cifar-1000-1000 | 9999.979 | 31499.980 | 9999.979 | 37999.979 | 21499.977 | 42999.953 | 9999.974 |
| randn-500-1000 | 19499.981 | 33499.981 | 19999.979 | 19999.980 | 44999.981 | 17999.978 | 9999.977 |

Table 1: Comparisons of objective values for $L_0$-regularized SPCA. The $1^{st}$, $2^{nd}$, and $3^{rd}$ best results are colored with red, green and blue, respectively.

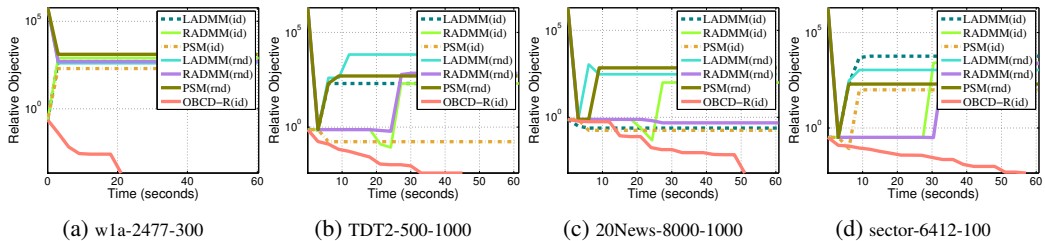

(a) w1a-2477-300    (b) TDT2-500-1000    (c) 20News-8000-1000    (d) sector-6412-100

Figure 1: The convergence curve for solving $L_0$-regularized SPCA with $\lambda = 100$. No matter how long the algorithms run, the other methods remain trapped in poor local minima.

code for all three applications used to reproduce the experiments can be found in the **supplemental material**.

▶ **Experiment Settings**. We compare objective values $F(\mathbf{X})$ for different methods after running for 30 seconds. For numerical stability in reporting the objectives, we use the count of elements with absolute values greater than a threshold of $10^{-6}$ instead of the original $\ell_0$ norm function $\|\mathbf{X}\|_0$. We set $\alpha = 10^{-5}$ for **OBCD**. Full-gradient methods have higher per-iteration complexity but require fewer iterations, while **OBCD**, as a partial-gradient method, has lower per-iteration costs but needs more iterations. Thus, we compare based on CPU time rather than iteration count.

▶ **Experiment Results**. Table 1 and Figure 1 display accuracy and computational efficiency results for $L_0$-regularized PCA, yielding the following observations: *(i)* **OBCD-R** delivers the best performance. *(ii)* Unlike other methods where objectives fluctuate during iterations, **OBCD-R** monotonically decreases the objective function while maintaining the orthogonality constraint. This is because **OBCD** is a greedy descent method for this problem class. *(iii)* While other methods often get stuck in poor local minima, **OBCD-R** escapes from such minima and generally finds lower objectives, aligning with our theory that our methods locate *stronger stationary points*.

## 6 CONCLUSIONS

In this paper, we introduced **OBCD**, a new block coordinate descent method for nonsmooth composite optimization under orthogonality constraints. **OBCD** operates on $k$ rows of the solution matrix, offering lower computational complexity per iteration for $k \geq 2$. We also provide a novel optimality analysis, showing how **OBCD** exploits problem structure to escape bad local minima and find better stationary points than methods focused on critical points. Under the Kurdyka-Lojasiewicz (KL) inequality, we establish strong limit-point convergence. Additionally, we show how novel breakpoint search methods can be used to solve the subproblem when $k = 2$. Extensive experiments demonstrate that **OBCD** outperforms existing methods.

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

## LLM USAGE

A large language model (LLM) was used to assist in refining the writing of this paper.

## ACKNOWLEDGMENTS

This work was supported by Guangdong Natural Science Funds for Distinguished Young Scholar (2018B030306025).

# Appendix

The appendix section is organized as follows.

Section A covers notations, technical preliminaries, and relevant lemmas.

Section B shows how to solve the subproblem when $k = 2$.

Section C offers further discussions on the proposed algorithm.

Section D contains proofs from Section 2.

Section E contains proofs from Section 3.

Section F contains proofs from Section 4.

Section G presents additional experiment details and results.

## A  NOTATIONS, TECHNICAL PRELIMINARIES, AND RELEVANT LEMMAS

### A.1  NOTATIONS

Throughout this paper, $\mathcal{M} \triangleq \mathrm{St}(n, r)$ denotes the Stiefel manifold, which is an embedded submanifold of the Euclidean space $\mathbb{R}^{n \times r}$. Boldfaced lowercase letters denote vectors and uppercase letters denote real-valued matrices. We adopt the Matlab colon notation to denote indices that describe submatrices. For given natural numbers $n$ and $k$, we use $\{\mathcal{B}_1, \mathcal{B}_2, ..., \mathcal{B}_{\mathrm{C}_n^k}\}$ to denote all the possible combinations of the index vectors choosing $k$ items from $n$ without repetition, where $\mathrm{C}_n^k$ is the total number of such combinations and $\mathcal{B}_i \in \mathbb{N}^k$, $\forall i \in [\mathrm{C}_n^k]$. For any one-dimensional function $p(t) : \mathbb{R} \mapsto \mathbb{R}$, we define: $p(\pm x \mp y) \triangleq \min\{p(x - y), p(-x + y)\}$. We use the following notations in this paper.

- $[n]$: $\{1, 2, ..., n\}$
- $\|\mathbf{x}\|$: Euclidean norm: $\|\mathbf{x}\| = \|\mathbf{x}\|_2 = \sqrt{\langle \mathbf{x}, \mathbf{x} \rangle}$
- $\mathbf{x}_i$: the $i$-th element of vector $\mathbf{x}$
- $\mathbf{X}_{i,j}$ or $\mathbf{X}_{ij}$ : the ($i^{\mathrm{th}}$, $j^{\mathrm{th}}$) element of matrix $\mathbf{X}$
- $\mathrm{vec}(\mathbf{X})$ : $\mathrm{vec}(\mathbf{X}) \in \mathbb{R}^{nr \times 1}$, the vector formed by stacking the column vectors of $\mathbf{X}$
- $\mathrm{mat}(\mathbf{x})$ : $\mathrm{mat}(\mathbf{x}) \in \mathbb{R}^{n \times r}$, Convert $\mathbf{x} \in \mathbb{R}^{nr \times 1}$ into a matrix with $\mathrm{mat}(\mathrm{vec}(\mathbf{X})) = \mathbf{X}$
- $\mathbf{X}^\mathsf{T}$ : the transpose of the matrix $\mathbf{X}$
- $\mathrm{sign}(t)$ : the signum function, $\mathrm{sign}(t) = 1$ if $t \geq 0$ and $\mathrm{sign}(t) = -1$ otherwise
- $\det(\mathbf{D})$ : Determinant of a square matrix $\mathbf{D} \in \mathbb{R}^{n \times n}$
- $\mathrm{C}_n^k$ : the number of possible combinations choosing $k$ items from $n$ without repetition
- $\mathbf{0}_{n,r}$ : A zero matrix of size $n \times r$; the subscript is omitted sometimes
- $\mathbf{I}_r$ : $\mathbf{I}_r \in \mathbb{R}^{r \times r}$, Identity matrix
- $\mathbf{X} \succeq \mathbf{0} (\text{or} \succ \mathbf{0})$ : the Matrix $\mathbf{X}$ is symmetric positive semidefinite (or definite)
- $\mathrm{tr}(\mathbf{A})$ : Sum of the elements on the main diagonal $\mathbf{X}$: $\mathrm{tr}(\mathbf{A}) = \sum_i \mathbf{A}_{i,i}$
- $\langle \mathbf{X}, \mathbf{Y} \rangle$ : Euclidean inner product, i.e., $\langle \mathbf{X}, \mathbf{Y} \rangle = \sum_{ij} \mathbf{X}_{ij} \mathbf{Y}_{ij}$
- $\mathbf{X} \otimes \mathbf{Y}$ : Kronecker product of $\mathbf{X}$ and $\mathbf{Y}$
- $\|\mathbf{X}\|_\mathsf{F}$ : Frobenius norm: $(\sum_{ij} \mathbf{X}_{ij}^2)^{1/2}$
- $\|\mathbf{X}\|_\mathsf{sp}$ : Operator/Spectral norm: the largest singular value of $\mathbf{X}$
- $\|\mathbf{X}\|_0$: the number of non-zero elements in the matrix $\mathbf{X}$
- $\|\mathbf{X}\|_1$: the absolute sum of the elements in the matrix $\mathbf{X}$ with $\|\mathbf{X}\|_1 = \sum_{i,j} |\mathbf{X}_{i,j}|$
- $\|\max(|\mathbf{X}|, \tau)\|_1$: the capped-$\ell_1$ norm of $\mathbf{X}$ with $\|\max(|\mathbf{X}|, \tau)\|_1 = \sum_{i,j} \max(|\mathbf{X}_{i,j}|, \tau)$
- $\nabla f(\mathbf{X})$ : Euclidean gradient of $f(\mathbf{X})$ at $\mathbf{X}$
- $\nabla_\mathcal{M} f(\mathbf{X})$ : Riemannian gradient of $f(\mathbf{X})$ at $\mathbf{X}$

- $\partial F(\mathbf{X})$ : limiting Euclidean subdifferential of $F(\mathbf{X})$ at $\mathbf{X}$
- $\partial_{\mathcal{M}} F(\mathbf{X})$ : limiting Riemannian subdifferential of $F(\mathbf{X})$ at $\mathbf{X}$
- $\iota_{\Xi}(\mathbf{X})$ : the indicator function of a set $\Xi$ with $\iota_{\Xi}(\mathbf{X}) = 0$ if $\mathbf{X} \in \Xi$ and otherwise $+\infty$
- $\iota_{\geq 0}(\mathbf{X})$: indicator function of non-negativity constraint with $\iota_{\geq 0}(\mathbf{X}) = \left\{ \begin{smallmatrix} 0, & \mathbf{X} \geq \mathbf{0}; \\ \infty, & \text{else.} \end{smallmatrix} \right\}$
- $\mathbb{P}_{\Xi}(\mathbf{Z})$ : Orthogonal projection of $\mathbf{Z}$ with $\mathbb{P}_{\Xi}(\mathbf{Z}) = \arg\min_{\mathbf{X} \in \Xi} \|\mathbf{Z} - \mathbf{X}\|_{\mathsf{F}}^2$
- $\mathbb{P}_{\mathcal{M}}(\mathbf{Y})$ : Nearest orthogonal matrix of $\mathbf{Y}$ with $\mathbb{P}_{\mathcal{M}}(\mathbf{Y}) = \arg\min_{\mathbf{X}^\intercal \mathbf{X} = \mathbf{I}_r} \|\mathbf{X} - \mathbf{Y}\|_{\mathsf{F}}^2$
- $\text{dist}(\Xi, \Xi')$ : the distance between two sets with $\text{dist}(\Xi, \Xi') \triangleq \inf_{\mathbf{X} \in \Xi, \mathbf{X}' \in \Xi'} \|\mathbf{X} - \mathbf{X}'\|_{\mathsf{F}}$
- $\mathbb{A} + \mathbb{B}, \mathbb{A} - \mathbb{B}$: standard Minkowski addition and subtraction between sets $\mathbb{A}$ and $\mathbb{B}$
- $\mathbb{A} \oplus \mathbb{B}, \mathbb{A} \ominus \mathbb{B}$: element-wise addition and subtraction between sets $\mathbb{A}$ and $\mathbb{B}$
- $\|\partial F(\mathbf{X})\|_{\mathsf{F}}$: the distance from the origin to $\partial F(\mathbf{X})$ with $\|\partial F(\mathbf{X})\|_{\mathsf{F}} = \inf_{\mathbf{Y} \in \partial F(\mathbf{X})} \|\mathbf{Y}\|_{\mathsf{F}}$

## A.2 Technical Preliminaries

As the function $F(\cdot)$ can be non-convex and non-smooth, we introduce some tools in non-smooth analysis (Mordukhovich, 2006; Rockafellar & Wets, 2009). The domain of any extended real-valued function $F : \mathbb{R}^{n \times r} \to (-\infty, +\infty]$ is defined as $\text{dom}(F) \triangleq \{\mathbf{X} \in \mathbb{R}^{n \times r} : |F(\mathbf{X})| < +\infty\}$. The Fréchet subdifferential of $F$ at $\mathbf{X} \in \text{dom}(F)$ is defined as

$$\hat{\partial} F(\mathbf{X}) \triangleq \left\{ \boldsymbol{\xi} \in \mathbb{R}^{n \times r} : \lim_{\mathbf{Z} \to \mathbf{X}} \inf_{\mathbf{Z} \neq \mathbf{X}} \frac{F(\mathbf{Z}) - F(\mathbf{X}) - \langle \boldsymbol{\xi}, \mathbf{Z} - \mathbf{X} \rangle}{\|\mathbf{Z} - \mathbf{X}\|_{\mathsf{F}}} \geq 0 \right\},$$

while the limiting subdifferential of $F(\mathbf{X})$ at $\mathbf{X} \in \text{dom}(F)$ is denoted as

$$\partial F(\mathbf{X}) \triangleq \left\{ \boldsymbol{\xi} \in \mathbb{R}^n : \exists \mathbf{X}^t \to \mathbf{X}, F(\mathbf{X}^t) \to F(\mathbf{X}), \boldsymbol{\xi}^t \in \hat{\partial} F(\mathbf{X}^t) \to \boldsymbol{\xi}, \forall t \right\}.$$

We denote $\nabla F(\mathbf{X})$ as the gradient of $F(\cdot)$ at $\mathbf{X}$ in the Euclidean space. We have the following relation between $\hat{\partial} F(\mathbf{X})$, $\partial F(\mathbf{X})$, and $\nabla F(\mathbf{X})$. (i) It holds that $\hat{\partial} F(\mathbf{X}) \subseteq \partial F(\mathbf{X})$. (ii) If the function $F(\cdot)$ is convex, $\partial F(\mathbf{X})$ and $\hat{\partial} F(\mathbf{X})$ essentially the classical subdifferential for convex functions, i.e.,

$$\partial F(\mathbf{X}) = \hat{\partial} F(\mathbf{X}) = \left\{ \boldsymbol{\xi} \in \mathbb{R}^{n \times r} : F(\mathbf{Z}) \geq F(\mathbf{X}) + \langle \boldsymbol{\xi}, \mathbf{Z} - \mathbf{X} \rangle, \forall \mathbf{Z} \in \mathbb{R}^{n \times r} \right\}.$$

(iii) If the function $F(\cdot)$ is differentiable, then $\hat{\partial} F(\mathbf{X}) = \partial F(\mathbf{X}) = \{\nabla F(\mathbf{X})\}$.

We need some prerequisite knowledge in optimization with orthogonality constraints (Absil et al., 2008). The nearest orthogonality matrix to an arbitrary matrix $\mathbf{Y} \in \mathbb{R}^{n \times r}$ is given by $\mathbb{P}_{\mathcal{M}}(\mathbf{Y}) = \hat{\mathbf{U}} \hat{\mathbf{V}}^\intercal$, where $\mathbf{Y} = \hat{\mathbf{U}} \text{Diag}(\mathbf{s}) \hat{\mathbf{V}}^\intercal$ is the singular value decomposition of $\mathbf{Y}$. We use $\mathcal{N}_{\mathcal{M}}(\mathbf{X})$ to denote the limiting normal cone to $\mathcal{M}$ at $\mathbf{X}$, leading to

$$\mathcal{N}_{\mathcal{M}}(\mathbf{X}) = \partial \iota_{\mathcal{M}}(\mathbf{X}) = \{\mathbf{Z} \in \mathbb{R}^{n \times r} : \langle \mathbf{Z}, \mathbf{X} \rangle \geq \langle \mathbf{Z}, \mathbf{Y} \rangle, \forall \mathbf{Y} \in \mathcal{M}\}.$$

The tangent and norm space to $\mathcal{M}$ at $\mathbf{X} \in \mathcal{M}$ are denoted as $T_{\mathbf{X}} \mathcal{M}$ and $N_{\mathbf{X}} \mathcal{M}$, respectively. For a given $\mathbf{X} \in \mathcal{M}$, we let $\mathcal{A}_{\mathbf{X}}(\mathbf{Y}) \triangleq \mathbf{X}^\intercal \mathbf{Y} + \mathbf{Y}^\intercal \mathbf{X}$ for $\mathbf{Y} \in \mathbb{R}^{n \times r}$, and we have $T_{\mathbf{X}} \mathcal{M} = \{\mathbf{Y} \in \mathbb{R}^{n \times r} | \mathcal{A}_X(\mathbf{Y}) = \mathbf{0}\}$ and $N_{\mathbf{X}} \mathcal{M} = \{2\mathbf{X}\boldsymbol{\Lambda} | \boldsymbol{\Lambda} = \boldsymbol{\Lambda}^\intercal, \boldsymbol{\Lambda} \in \mathbb{R}^{r \times r}\}$. For any non-convex and non-smooth function $F(\mathbf{X})$, we use $\partial_{\mathcal{M}} F(\mathbf{X})$ to denote the limiting Riemannian gradient of $F(\mathbf{X})$ at $\mathbf{X}$, and obtain $\partial_{\mathcal{M}} F(\mathbf{X}) = \mathbb{P}_{T_{\mathbf{X}} \mathcal{M}}(\partial F(\mathbf{X}))$. We denote $\partial F(\mathbf{X}) \ominus \mathbf{X}[\partial F(\mathbf{X})]^\intercal \mathbf{X} \triangleq \{\mathbb{E} | \mathbb{E} = \mathbf{G} - \mathbf{X}\mathbf{G}^\intercal \mathbf{X}, \mathbf{G} \in \partial F(\mathbf{X})\}$.

## A.3 Relevant Lemmas

We offer a set of useful lemmas, each of which stands independently of context and specific methodology.

**Lemma A.1.** *Let $k \geq 2$ and $\mathbf{W} \in \mathbb{R}^{n \times n}$. If $\mathbf{0}_{k,k} = \mathrm{U}_{\mathbb{B}}^\intercal \mathbf{W} \mathrm{U}_{\mathbb{B}}$ for all $\mathbb{B} \in \{\mathcal{B}_i\}_{i=1}^{\mathrm{C}_n^k}$, then $\mathbf{W} = \mathbf{0}$. Here, the set $\{\mathcal{B}_1, \mathcal{B}_2, ..., \mathcal{B}_{\mathrm{C}_n^k}\}$ represents all possible combinations of the index vectors choosing $k$ items from $n$ without repetition.*

*Proof.* The proof is straightforward and relies on elementary reasoning.

Notably, the conclusion of this lemma does not necessarily hold if $|\mathsf{B}| = k = 1$. This is because any matrix $\mathbf{W} \in \mathbb{R}^{n \times n}$ with $\mathbf{W}_{ii} = 0$ for all $i \in [n]$ satisfies the condition of this lemma but is not necessary a zero matrix. $\square$

**Lemma A.2.** *For any matrices $\mathbf{A} \in \mathbb{R}^{k \times k}$ and $\mathbf{C} \in \mathbb{R}^{k \times k}$, we have:*

$$\|\mathbf{A} - \mathbf{A}^\mathsf{T}\|_\mathsf{F} \le 2\|\mathbf{A} - \mathbf{C}\|_\mathsf{F} + \|\mathbf{C} - \mathbf{C}^\mathsf{T}\|_\mathsf{F}.$$

*Proof.* We derive: $\|\mathbf{A} - \mathbf{A}^\mathsf{T}\|_\mathsf{F} = \|(\mathbf{A} - \mathbf{C}) + (\mathbf{C} - \mathbf{C}^\mathsf{T}) + (\mathbf{C}^\mathsf{T} - \mathbf{A}^\mathsf{T})\|_\mathsf{F} \overset{①}{\le} \|\mathbf{A} - \mathbf{C}\|_\mathsf{F} + \|\mathbf{C} - \mathbf{C}^\mathsf{T}\|_\mathsf{F} + \|\mathbf{C}^\mathsf{T} - \mathbf{A}^\mathsf{T}\|_\mathsf{F} = 2\|\mathbf{A} - \mathbf{C}\|_\mathsf{F} + \|\mathbf{C} - \mathbf{C}^\mathsf{T}\|_\mathsf{F}$, where step ① uses the triangle inequality.

$\square$

**Lemma A.3.** *Let $\tau \in \mathbb{R}$, and $\mathbf{A} \in \mathbb{R}^{2 \times 2}$ be any skey-symmetric matrix with $\mathbf{A}^\mathsf{T} = -\mathbf{A}$. We have:*

$$\det\left((\mathbf{I}_2 + \tfrac{\tau}{2}\mathbf{A})^{-1}(\mathbf{I}_k - \tfrac{\tau}{2}\mathbf{A})\right) = 1.$$

*Proof.* Since $\mathbf{A}$ is a two-dimensional matrix, it can be expressed in the form: $\mathbf{A} = \left(\begin{smallmatrix} 0 & a \\ -a & 0 \end{smallmatrix}\right)$ for some $a \in \mathbb{R}$. Letting $b = \tfrac{\tau}{2}a$, we derive:

$$\mathbf{Q} \triangleq (\mathbf{I}_2 + \tfrac{\tau}{2}\mathbf{A})^{-1}(\mathbf{I}_k - \tfrac{\tau}{2}\mathbf{A}) \overset{①}{=} \left(\begin{smallmatrix} 1 & b \\ -b & 1 \end{smallmatrix}\right)^{-1}\left(\begin{smallmatrix} 1 & -b \\ b & 1 \end{smallmatrix}\right) \overset{②}{=} \tfrac{1}{1+b^2}\left(\begin{smallmatrix} 1 & -b \\ b & 1 \end{smallmatrix}\right)\left(\begin{smallmatrix} 1 & -b \\ b & 1 \end{smallmatrix}\right) = \tfrac{1}{1+b^2}\left(\begin{smallmatrix} 1-b^2 & -2b \\ 2b & 1-b^2 \end{smallmatrix}\right),$$

where step ① uses $\tfrac{\tau}{2}\mathbf{A} = \left(\begin{smallmatrix} 0 & b \\ -b & 0 \end{smallmatrix}\right)$; step ② uses the fact that $\left(\begin{smallmatrix} a & b \\ c & d \end{smallmatrix}\right)^{-1} = \tfrac{1}{ad-bc}\left(\begin{smallmatrix} d & -b \\ -c & a \end{smallmatrix}\right)^{-1}$ for all $a, b, c, d \in \mathbb{R}$. We further obtain: $\det(\mathbf{Q}) \overset{①}{=} \tfrac{1-b^2}{1+b^2} \cdot \tfrac{1-b^2}{1+b^2} - \tfrac{2b}{1+b^2} \cdot \tfrac{-2b}{1+b^2} = \tfrac{(1-b^2)^2 + 4b^2}{(1+b^2)^2} = \tfrac{(1+b^2)^2}{(1+b^2)^2} = 1$, where step ① uses the fact that $\det\left(\begin{smallmatrix} a & b \\ c & d \end{smallmatrix}\right) = ad - bc$ for all $a, b, c, d \in \mathbb{R}$.

$\square$

**Lemma A.4.** *For any $\mathbf{W} \in \mathbb{R}^{n \times n}$, we have*

$$\sum_{i=1}^{\mathrm{C}_n^k} \|\mathbf{W}(\mathcal{B}_i, \mathcal{B}_i)\|_\mathsf{F}^2 = \mathrm{C}_{n-2}^{k-2} \sum_i \sum_{j, j \ne i} \mathbf{W}_{ij}^2 + \tfrac{k}{n}\mathrm{C}_n^k \sum_i \mathbf{W}_{ii}^2.$$

*Here, the set $\{\mathcal{B}_1, \mathcal{B}_2, ..., \mathcal{B}_{\mathrm{C}_n^k}\}$ represents all possible combinations of the index vectors choosing $k$ items from $n$ without repetition.*

*Proof.* For any matrix $\mathbf{W} \in \mathbb{R}^{n \times n}$, we define: $\mathbf{w} \triangleq \mathrm{diag}(\mathbf{W}) \in \mathbb{R}^n$, and $\mathbf{W}' \triangleq \mathbf{W} - \mathrm{Diag}(\mathbf{w})$.

We have: $\mathbf{W} = \mathrm{Diag}(\mathbf{w}) + \mathbf{W}'$, this leads to the following decomposition:

$$
\begin{aligned}
\sum_{i=1}^{\mathrm{C}_n^k} \|\mathbf{U}_{\mathcal{B}_i}^\mathsf{T} \mathbf{W} \mathbf{U}_{\mathcal{B}_i}\|_\mathsf{F}^2 &= \sum_{i=1}^{\mathrm{C}_n^k} \|\mathbf{U}_{\mathcal{B}_i}^\mathsf{T}(\mathrm{Diag}(\mathbf{w}) + \mathbf{W}')\mathbf{U}_{\mathcal{B}_i}\|_\mathsf{F}^2 \\
&= \underbrace{\sum_{i=1}^{\mathrm{C}_n^k} \|\mathbf{U}_{\mathcal{B}_i}^\mathsf{T}\mathrm{Diag}(\mathbf{w})\mathbf{U}_{\mathcal{B}_i}\|_\mathsf{F}^2}_{\Gamma_1} + \underbrace{\sum_{i=1}^{\mathrm{C}_n^k} \|\mathbf{U}_{\mathcal{B}_i}^\mathsf{T}\mathbf{W}'\mathbf{U}_{\mathcal{B}_i}\|_\mathsf{F}^2}_{\Gamma_2}. \quad (12)
\end{aligned}
$$

We first focus on the term $\Gamma_1$. We have:

$$\Gamma_1 = \sum_{i=1}^{\mathrm{C}_n^k} \|\mathbf{U}_{\mathcal{B}_i}^\mathsf{T}\mathrm{Diag}(\mathbf{w})\mathbf{U}_{\mathcal{B}_i}\|_\mathsf{F}^2 \overset{①}{=} \sum_{i=1}^{\mathrm{C}_n^k} \|\mathbf{w}_{\mathcal{B}_i}\|_2^2 \overset{②}{=} \mathrm{C}_n^k \cdot \tfrac{k}{n} \cdot \|\mathbf{w}\|_2^2 = \mathrm{C}_n^k \cdot \tfrac{k}{n} \cdot \sum_i \mathbf{W}_{ii}^2, \quad (13)$$

where step ① uses the fact that $\|\mathsf{B}^\mathsf{T}\mathrm{Diag}(\mathbf{w})\mathsf{B}\|_\mathsf{F}^2 = \|[\mathrm{Diag}(\mathbf{w})]_{\mathsf{BB}}\|_\mathsf{F}^2 = \|\mathbf{w}_\mathsf{B}\|_2^2$ for any $\mathsf{B} \in \{\mathcal{B}_i\}_{i=1}^{\mathrm{C}_n^k}$; step ② uses the observation that $\mathbf{w}_i$ appears in the term $\sum_{i=1}^{\mathrm{C}_n^k} \|\mathbf{w}_{\mathcal{B}_i}\|_2^2$ a total of $(\mathrm{C}_n^k \cdot \tfrac{k}{n})$ times, which can be deduced using basic induction.

We now focus on the term $\Gamma_2$. Noticing that $\mathbf{W}'_{ii} = 0$ for all $i \in [n]$, we have:

$$\Gamma_2 = \sum_{i=1}^{\mathrm{C}_n^k} \|\mathbf{U}_{\mathcal{B}_i}^\mathsf{T}\mathbf{W}'\mathbf{U}_{\mathcal{B}_i}\|_\mathsf{F}^2 \overset{①}{=} \sum_i \sum_{j \ne i}[\mathrm{C}_{n-2}^{k-2}(\mathbf{W}'_{ij})^2] \overset{②}{=} \mathrm{C}_{n-2}^{k-2} \sum_i \sum_{j, j \ne i}(\mathbf{W}_{ij})^2, \quad (14)$$

where step ① uses the fact that the term $\sum_{i=1}^{\mathrm{C}_n^k} \|\mathbf{U}_{\mathcal{B}_i}^\mathsf{T}\mathbf{W}'\mathbf{U}_{\mathcal{B}_i}\|_\mathsf{F}^2$ comprises $\mathrm{C}_{n-2}^{k-2}$ distinct patterns, each including $\{i, j\}$ with $i \ne j$; step ② uses $\sum_{i, j \ne i}(\mathbf{W}_{ij})^2 = \sum_{i, j \ne i}(\mathbf{W}'_{ij})^2$.

In view of Equalities (12), (13), and (14), we complete the proof of this lemma. $\square$

**Lemma A.5.** *Assume* $\mathbf{Q}\mathbf{R} = \mathbf{X} \in \mathbb{R}^{n \times n}$*, where* $\mathbf{Q} \in \mathrm{St}(n, n)$ *and* $\mathbf{R}$ *is a lower triangular matrix with* $\mathbf{R}_{i,j} = 0$ *for all* $i < j$*. If* $\mathbf{X} \in \mathrm{St}(n, n)$*, then* $\mathbf{R}$ *is a diagonal matrix with* $\mathbf{R}_{i,i} \in \{-1, +1\}$ *for all* $i \in [n]$*.*

*Proof.* We derive: $\mathbf{R}\mathbf{R}^{\mathsf{T}} \overset{\text{①}}{=} (\mathbf{Q}\mathbf{X})(\mathbf{Q}\mathbf{X})^{\mathsf{T}} = \mathbf{Q}\mathbf{X}\mathbf{X}^{\mathsf{T}}\mathbf{Q}^{\mathsf{T}} \overset{\text{②}}{=} \mathbf{I}$, where step ① uses $\mathbf{R} = \mathbf{Q}^{\mathsf{T}}\mathbf{X}$; step ② uses $\mathbf{X} \in \mathrm{St}(n, n)$ and $\mathbf{Q} \in \mathrm{St}(n, n)$. First, given $\|\mathbf{R}(1, :)\| = 1$ and $\mathbf{R}(1, 2 : n) = 0$, we have $\mathbf{R}_{1,1} \in \{-1, +1\}$. Second, we have $\|\mathbf{R}(2, :)\| = 1$ and $\mathbf{R}(1, :)^{\mathsf{T}}\mathbf{R}(:, 2) = 0$, leading to $\mathbf{R}_{1,2} = 0$ and $\mathbf{R}_{2,2} \in \{-1, +1\}$. Finally, using similar recursive strategy, we conclude that $\mathbf{R}$ is a diagonal matrix with $\mathbf{R}_{i,i} \in \{-1, +1\}$ for all $i \in [n]$. $\qquad\square$

**Lemma A.6.** *We define* $\mathrm{T}_{\mathbf{X}}\mathcal{M} \triangleq \{\mathbf{Y} \in \mathbb{R}^{n \times r} \,|\, \mathcal{A}_X(\mathbf{Y}) = 0\}$ *and* $\mathcal{A}_{\mathbf{X}}(\mathbf{Y}) \triangleq \mathbf{X}^{\mathsf{T}}\mathbf{Y} + \mathbf{Y}^{\mathsf{T}}\mathbf{X}$*. For any* $\mathbf{G} \in \mathbb{R}^{n \times r}$ *and* $\mathbf{X} \in \mathrm{St}(n, k)$*, we have:*

$$(\mathbf{G} - \tfrac{1}{2}\mathbf{X}\mathcal{A}_{\mathbf{X}}(\mathbf{G})) = \arg \min_{\mathbf{Y} \in \mathrm{T}_{\mathbf{X}}\mathcal{M}} \|\mathbf{Y} - \mathbf{G}\|_{\mathsf{F}}^2.$$

*Proof.* The conclusion of this lemma can be found in (Absil et al., 2008). For completeness, we present a short proof.

Consider the convex problem: $\bar{\mathbf{Y}} = \arg \min_{\mathbf{Y}} \|\mathbf{Y} - \mathbf{G}\|_{\mathsf{F}}^2$, $s.t.\, \mathbf{X}^{\mathsf{T}}\mathbf{Y} + \mathbf{Y}^{\mathsf{T}}\mathbf{X} = 0$. Introducing a multiplier $\mathbf{\Lambda} \in \mathbb{R}^{r \times r}$ for the linear constraints leads to the following Lagrangian function: $\tilde{\mathcal{L}}(\mathbf{Y}; \mathbf{\Lambda}) = \|\mathbf{Y} - \mathbf{G}\|_{\mathsf{F}}^2 + \langle \mathbf{X}^{\mathsf{T}}\mathbf{Y} + \mathbf{Y}^{\mathsf{T}}\mathbf{X}, \mathbf{\Lambda} \rangle$. We derive the subsequent first-order optimality condition: $2(\mathbf{Y} - \mathbf{G}) + \mathbf{X}(\mathbf{\Lambda} + \mathbf{\Lambda}^{\mathsf{T}}) = 0$, and $\mathbf{X}^{\mathsf{T}}\mathbf{Y} + \mathbf{Y}^{\mathsf{T}}\mathbf{X} = 0$. Given $\mathbf{\Lambda}$ is symmetric, we have $\mathbf{Y} = \mathbf{G} - \mathbf{X}\mathbf{\Lambda}$. Incorporating this result into $\mathbf{X}^{\mathsf{T}}\mathbf{Y} + \mathbf{Y}^{\mathsf{T}}\mathbf{X} = 0$, we obtain: $\mathbf{X}^{\mathsf{T}}(\mathbf{G} - \mathbf{X}\mathbf{\Lambda}) + (\mathbf{G} - \mathbf{X}\mathbf{\Lambda})^{\mathsf{T}}\mathbf{X} = 0$. Given $\mathbf{X} \in \mathrm{St}(n, r)$, we have $\mathbf{X}^{\mathsf{T}}\mathbf{G} - \mathbf{\Lambda} + \mathbf{G}^{\mathsf{T}}\mathbf{X} - \mathbf{\Lambda}^{\mathsf{T}} = 0$, leading to: $\mathbf{\Lambda} = \tfrac{1}{2}(\mathbf{X}^{\mathsf{T}}\mathbf{G} + \mathbf{G}^{\mathsf{T}}\mathbf{X})$. Therefore, the optimal solution $\bar{\mathbf{Y}}$ can be computed as $\bar{\mathbf{Y}} = \mathbf{G} - \mathbf{X}\mathbf{\Lambda} = \mathbf{G} - \tfrac{1}{2}\mathbf{X}(\mathbf{X}^{\mathsf{T}}\mathbf{G} + \mathbf{G}^{\mathsf{T}}\mathbf{X})$.

$\qquad\square$

**Lemma A.7.** *Consider the following problem:* $\min_{\mathbf{X}} F_{\iota}(\mathbf{X}) \triangleq F(\mathbf{X}) + \iota_{\mathcal{M}}(\mathbf{X})$*, where* $F(\mathbf{X})$ *is defined in Equation (1). For any* $\mathbf{X} \in \mathrm{St}(n, r)$*, it holds that*

$$\mathrm{dist}(\mathbf{0}, \partial F_{\iota}(\mathbf{X})) \le \mathrm{dist}(\mathbf{0}, \partial_{\mathcal{M}} F(\mathbf{X})).$$

*Proof.* We let $\mathbf{G} \in \partial F(\mathbf{X})$ and define $\mathcal{A}_{\mathbf{X}}(\mathbf{G}) \triangleq \mathbf{X}^{\mathsf{T}}\mathbf{G} + \mathbf{G}^{\mathsf{T}}\mathbf{X}$.

Recall that the following first-order optimality conditions are equivalent for all $\mathbf{X} \in \mathrm{St}(n, r)$: $(\mathbf{0} \in \partial F_{\iota}(\mathbf{X})) \Leftrightarrow (\mathbf{0} \in \mathbb{P}_{\mathrm{T}_{\mathbf{X}}\mathcal{M}}(\partial F(\mathbf{X})))$. Therefore, we derive:

$$
\begin{aligned}
\mathrm{dist}(\mathbf{0}, \partial F_{\iota}(\mathbf{X})) &= \inf_{\mathbf{Y} \in \partial F_{\iota}(\mathbf{X})} \|\mathbf{Y}\|_{\mathsf{F}} = \inf_{\mathbf{Y} \in \mathbb{P}_{(\mathrm{T}_{\mathbf{X}}\mathcal{M})}(\partial F(\mathbf{X}))} \|\mathbf{Y}\|_{\mathsf{F}} \\
&\overset{\text{①}}{=} \|\mathbb{P}_{(\mathrm{T}_{\mathbf{X}}\mathcal{M})}(\mathbf{G})\|_{\mathsf{F}} \\
&\overset{\text{②}}{=} \|\mathbf{G} - \tfrac{1}{2}\mathbf{X}\mathcal{A}_{\mathbf{X}}(\mathbf{G})\|_{\mathsf{F}} \\
&\overset{\text{③}}{=} \|\mathbf{G} - \tfrac{1}{2}\mathbf{X}(\mathbf{X}^{\mathsf{T}}\mathbf{G} + \mathbf{G}^{\mathsf{T}}\mathbf{X})\|_{\mathsf{F}} \\
&\overset{\text{④}}{=} \|(\mathbf{I} - \tfrac{1}{2}\mathbf{X}\mathbf{X}^{\mathsf{T}})(\mathbf{G} - \mathbf{X}\mathbf{G}^{\mathsf{T}}\mathbf{X})\|_{\mathsf{F}} \\
&\overset{\text{⑤}}{\le} \|\mathbf{G} - \mathbf{X}\mathbf{G}^{\mathsf{T}}\mathbf{X}\|_{\mathsf{F}},
\end{aligned}
$$

where step ① uses $\mathbf{G} \in \partial F(\mathbf{X})$; step ② uses Lemma A.6; step ③ uses the definition of $\mathcal{A}_{\mathbf{X}}(\mathbf{G})$; step ④ uses the identity that $\mathbf{G} - \tfrac{1}{2}\mathbf{X}(\mathbf{X}^{\mathsf{T}}\mathbf{G} + \mathbf{G}^{\mathsf{T}}\mathbf{X}) = (\mathbf{I} - \tfrac{1}{2}\mathbf{X}\mathbf{X}^{\mathsf{T}})(\mathbf{G} - \mathbf{X}\mathbf{G}^{\mathsf{T}}\mathbf{X})$; step ⑤ uses the norm inequality and fact that the matrix $\mathbf{I} - \tfrac{1}{2}\mathbf{X}\mathbf{X}^{\mathsf{T}}$ only contains eigenvalues that are $\tfrac{1}{2}$ or 1.

$\qquad\square$

**Lemma A.8.** *Assume* $\cos(\theta) \ne 0$*. Any pair of trigonometric functions* $(\cos(\theta), \sin(\theta))$ *can be represented as follows:*

*a)* $\cos(\theta) = \frac{1}{\sqrt{1+\tan^2(\theta)}}$*, and* $\sin(\theta) = \frac{\tan(\theta)}{\sqrt{1+\tan^2(\theta)}}$*.*

**b)** $\cos(\theta) = \frac{-1}{\sqrt{1+\tan^2(\theta)}}$, *and* $\sin(\theta) = \frac{-\tan(\theta)}{\sqrt{1+\tan^2(\theta)}}$.

*Proof.* For all values of $\theta$ where $\cos(\theta) \neq 0$, the trigonometric functions $\{\sin(\theta), \cos(\theta), \tan(\theta)\}$ are well-defined. Utilizing the identity $\sin^2(\theta) + \cos^2(\theta) = 1$ and $\tan(\theta)\cos(\theta) = \sin(\theta)$, we derive: $(\tan(\theta) \cdot \cos(\theta))^2 + \cos^2(\theta) = 1$. Consequently, we find: $\cos(\theta) = \frac{\pm 1}{\sqrt{\tan^2(\theta)+1}}$. Finally, we can express $\sin(\theta)$ as $\sin(\theta) = \tan(\theta) \cdot \cos(\theta) = \frac{\tan(\theta)}{\sqrt{\tan^2(\theta)+1}}$.

$\square$

**Lemma A.9.** *Assume* $(E_{t+1})^2 \leq E_t(p_t - p_{t+1})$ *and* $p_t \geq p_{t+1}$, *where* $\{E_t, p_t\}_{t=0}^\infty$ *are two non-negative sequences. For all* $i \geq 1$, *we have:* $\sum_{t=i}^\infty E_{t+1} \leq E_i + 2p_i$.

*Proof.* We define $w_t \triangleq p_t - p_{t+1}$. We let $1 \leq i < T$.

First, for any $i \geq 1$, we have:

$$\sum_{t=i}^T w_t = \sum_{t=i}^T (p_t - p_{t+1}) = p_i - p_{T+1} \overset{\text{①}}{\leq} p_i, \tag{15}$$

where step ① uses $p_i \geq 0$ for all $i$.

Second, we obtain:

$$
\begin{aligned}
E_{t+1} &\overset{\text{①}}{\leq} \sqrt{E_t w_t} \\
&\overset{\text{②}}{\leq} \sqrt{\tfrac{\alpha}{2}(E_t)^2 + (w_t)^2/(2\alpha)}, \forall \alpha > 0 \\
&\overset{\text{③}}{\leq} \sqrt{\tfrac{\alpha}{2}} \cdot E_t + w_t\sqrt{1/(2\alpha)}, \forall \alpha > 0.
\end{aligned}
\tag{16}
$$

Here, step ① uses $(E_{t+1})^2 \leq E_t(p_t - p_{t+1})$ and $w_t \triangleq p_t - p_{t+1}$; step ② uses the fact that $ab \leq \frac{\alpha}{2}a^2 + \frac{1}{2\alpha}b^2$ for all $\alpha > 0$; step ③ uses the fact that $\sqrt{a+b} \leq \sqrt{a} + \sqrt{b}$ for all $a, b \geq 0$.

Assume $1 - \sqrt{\frac{\alpha}{2}} > 0$. Telescoping Inequality (16) over $t$ from $i$ to $T$, we have:

$$
\begin{aligned}
&\sum_{t=i}^T w_t \sqrt{1/(2\alpha)} \\
&\geq \{\sum_{t=i}^T E_{t+1}\} - \sqrt{\tfrac{\alpha}{2}}\{\sum_{t=i}^T E_t\} \\
&= \{E_{T+1} + \sum_{t=i}^{T-1} E_{t+1}\} - \sqrt{\tfrac{\alpha}{2}}\{E_i + \sum_{t=i}^{T-1} E_{t+1}\} \\
&= E_{T+1} - \sqrt{\tfrac{\alpha}{2}}E_i + (1 - \sqrt{\tfrac{\alpha}{2}})\sum_{t=i}^{T-1} E_{t+1} \\
&\overset{\text{①}}{\geq} -\sqrt{\tfrac{\alpha}{2}}E_i + (1 - \sqrt{\tfrac{\alpha}{2}})\sum_{t=i}^{T-1} E_{t+1},
\end{aligned}
$$

where step ① uses $E_{T+1} \geq 0$ and $1 - \sqrt{\frac{\alpha}{2}} > 0$. This leads to:

$$
\begin{aligned}
\sum_{t=i}^{T-1} E_{t+1} &\leq (1 - \sqrt{\tfrac{\alpha}{2}})^{-1} \cdot \{\sqrt{\tfrac{\alpha}{2}}E_i + \sqrt{\tfrac{1}{2\alpha}}\sum_{t=i}^T w_t\} \\
&\overset{\text{①}}{=} E_i + 2\sum_{t=i}^T w_t \\
&\overset{\text{②}}{\leq} E_i + 2p_i,
\end{aligned}
$$

step ① uses the fact that $(1 - \sqrt{\frac{\alpha}{2}})^{-1} \cdot \sqrt{\frac{\alpha}{2}} = 1$ and $(1 - \sqrt{\frac{\alpha}{2}})^{-1} \cdot \sqrt{\frac{1}{2\alpha}} = 2$ when $\alpha = \frac{1}{2}$; step ② uses Inequalities (15). Letting $T \to \infty$, we conclude this lemma.

$\square$

**Lemma A.10.** *Assume that* $[D_t]^{\tau+1} \leq a(D_{t-1} - D_t)$, *where* $\tau, a > 0$, *and* $\{D_t\}_{t=0}^\infty$ *is a nonnegative sequence. We have:* $D_T \leq \mathcal{O}(T^{-1/\tau})$.

*Proof.* We let $\kappa > 1$ be any constant. We define $h(s) = s^{-\tau-1}$, where $\tau > 0$.

We consider two cases for $r^t \triangleq h(D_t)/h(D_{t-1})$.

**Case (1)**. $r^t \leq \kappa$. We define $\breve{h}(s) \triangleq -\frac{1}{\tau} \cdot s^{-\tau}$. We derive:

$$
\begin{aligned}
1 \quad &\overset{①}{\leq} \quad a(D_{t-1} - D_t) \cdot h(D_t) \\
&\overset{②}{\leq} \quad a(D_{t-1} - D_t) \cdot \kappa h(D_{t-1}) \\
&\overset{③}{\leq} \quad a\kappa \int_{D_t}^{D_{t-1}} h(s)ds \\
&\overset{④}{=} \quad a\kappa \cdot (\breve{h}(D_{t-1}) - \breve{h}(D_t)) \\
&\overset{⑤}{=} \quad a\kappa \cdot \frac{1}{\tau} \cdot ([D_t]^{-\tau} - [D_{t-1}]^{-\tau}),
\end{aligned}
$$

where step ① uses $[D_t]^{\tau+1} \leq a(D_{t-1} - D_t)$; step ② uses $h(D_t) \leq \kappa h(D_{t-1})$; step ③ uses the fact that $h(s)$ is a nonnegative and increasing function that $(a-b)h(a) \leq \int_b^a h(s)ds$ for all $a, b \in [0, \infty)$; step ④ uses the fact that $\nabla \breve{h}(s) = h(s)$; step ⑤ uses the definition of $\breve{h}(\cdot)$. This leads to:

$$[D_t]^{-\tau} - [D_{t-1}]^{-\tau} \geq \tfrac{\tau}{\kappa a}. \tag{17}$$

**Case (2)**. $r^t > \kappa$. We have:

$$
\begin{aligned}
h(D_t) > \kappa h(D_{t-1}) \quad &\overset{①}{\Rightarrow} \quad [D_t]^{-(\tau+1)} > \kappa \cdot [D_{t-1}]^{-(\tau+1)} \\
&\overset{②}{\Rightarrow} \quad ([D_t]^{-(\tau+1)})^{\frac{\tau}{\tau+1}} > \kappa^{\frac{\tau}{\tau+1}} \cdot ([D_{t-1}]^{-(\tau+1)})^{\frac{\tau}{\tau+1}} \\
&\Rightarrow \quad [D_t]^{-\tau} > \kappa^{\frac{\tau}{\tau+1}} \cdot [D_{t-1}]^{-\tau},
\end{aligned} \tag{18}
$$

where step ① uses the definition of $h(\cdot)$; step ② uses the fact that if $a > b > 0$, then $a^{\dot\tau} > b^{\dot\tau}$ for any exponent $\dot\tau \triangleq \frac{\tau}{\tau+1} \in (0, 1)$. For any $t \geq 1$, we derive:

$$
\begin{aligned}
[D_t]^{-\tau} - [D_{t-1}]^{-\tau} \quad &\overset{①}{\geq} \quad (\kappa^{\frac{\tau}{\tau+1}} - 1) \cdot [D_{t-1}]^{-\tau} \\
&\overset{②}{\geq} \quad (\kappa^{\frac{\tau}{\tau+1}} - 1) \cdot [D_0]^{-\tau},
\end{aligned} \tag{19}
$$

where step ① uses Inequality (18); step ② uses $\tau > 0$ and $D_{t-1} \leq D_0$ for all $t \geq 1$.

In view of Inequalities (17) and (19), we have:

$$[D_t]^{-\tau} - [D_{t-1}]^{-\tau} \geq \underbrace{\min(\tfrac{\tau}{\kappa a}, (\kappa^{\frac{\tau}{\tau+1}} - 1) \cdot [D_0]^{-\tau})}_{\triangleq \ddot{c}}. \tag{20}$$

Telescoping Inequality (20) over $t$ from 1 to $T$, we have:

$$[D_T]^{-\tau} - [D_0]^{-\tau} \geq T\ddot{c}.$$

This leads to:

$$D_T = ([D_T]^{-\tau})^{-1/\tau} \leq \mathcal{O}(T^{-1/\tau}).$$

$\square$

## B   SOLVING THE SUBPROBLEM WHEN $k = 2$

This section presents a novel Breakpoint Searching Method (**BSM**) to find the *global optimal solution* of Problem (3) when $k = 2$.

Initially, Problem (3) boils down to the following one-dimensional subproblem:

$$\min_{\theta} \ \tfrac{1}{2}\|\mathbf{V}\|_{\mathbf{Q}}^2 + \langle \mathbf{V}, \mathbf{P} \rangle + h(\mathbf{VZ}), \ s.t. \ \mathbf{V} \in \{\mathbf{V}_\theta^{\mathrm{rot}}, \mathbf{V}_\theta^{\mathrm{ref}}\},$$

which can be further rewritten as:

$$\bar{\theta} \in \arg\min_{\theta} \tfrac{1}{2}\mathrm{vec}(\mathbf{V})^{\mathsf{T}}\mathbf{Q}\mathrm{vec}(\mathbf{V}) + \langle \mathbf{V}, \mathbf{P}\rangle + h(\mathbf{VZ}), \; s.t. \; \mathbf{V} \triangleq \left(\begin{smallmatrix}\pm\cos(\theta) & \sin(\theta) \\ \mp\sin(\theta) & \cos(\theta)\end{smallmatrix}\right),$$

where $\mathbf{Q} \in \mathbb{R}^{4\times4}$, $\mathbf{P} \in \mathbb{R}^{2\times2}$, and $\mathbf{Z} \in \mathbb{R}^{2\times r}$. Given $h(\cdot)$ is coordinate-wise separable, we have the following equivalent optimization problem:

$$\min_{\theta} \quad h\left(\cos(\theta)\mathbf{x} + \sin(\theta)\mathbf{y}\right) + a\cos(\theta) + b\sin(\theta)$$
$$+ c\cos^2(\theta) + d\cos(\theta)\sin(\theta) + e\sin^2(\theta), \tag{21}$$

where $a = \mathbf{P}_{22} \pm \mathbf{P}_{11}$, $b = \mathbf{P}_{12} \mp \mathbf{P}_{21}$, $c = 0.5(\mathbf{Q}_{11} + \mathbf{Q}_{44}) \pm \mathbf{Q}_{14}$, $d = -\mathbf{Q}_{12} \pm \mathbf{Q}_{13} \mp \mathbf{Q}_{24} + \mathbf{Q}_{34}$, $e = 0.5(\mathbf{Q}_{22} + \mathbf{Q}_{33}) \mp \mathbf{Q}_{23}$, $\mathbf{r} = \pm\mathbf{Z}(1, :)$, $\mathbf{s} = \mathbf{Z}(2, :)$, $\mathbf{p} = \mathbf{Z}(2, :)$, $\mathbf{u} = \mp\mathbf{Z}(1, :)$, $\mathbf{x} \triangleq [\mathbf{r}; \mathbf{p}] \in \mathbb{R}^{2r\times1}$, and $\mathbf{y} \triangleq [\mathbf{s}; \mathbf{u}] \in \mathbb{R}^{2r\times1}$.

Our key strategy is to perform a variable substitution to convert Problem (21) into an equivalent problem that depends on the variable $\tan(\theta) \triangleq t$. The substitution is based on the trigonometric identities that $\cos(\theta) = \pm1/\sqrt{1 + \tan^2(\theta)}$ and $\sin(\theta) = \pm\tan(\theta)/\sqrt{1 + \tan^2(\theta)}$.

The following lemma provides a characterization of the global optimal solution for Problem (21).

**Lemma B.1.** *We define* $\breve{F}(\tilde{c}, \tilde{s}) \triangleq a\tilde{c} + b\tilde{s} + c\tilde{c}^2 + d\tilde{c}\tilde{s} + e\tilde{s}^2 + h(\tilde{c}\mathbf{x} + \tilde{s}\mathbf{y})$, *and* $w \triangleq c - e$. *The optimal solution* $\bar{\theta}$ *to (21) can be computed as:*

$$[\cos(\bar{\theta}), \sin(\bar{\theta})] \in \arg\min_{[c,s]} \breve{F}(c, s), \; s.t. \; [c, s] \in \left\{[c_1, s_1], [c_2, s_2], [0, 1], [0, -1]\right\},$$

*where* $c_1 \triangleq \frac{1}{\sqrt{1+(\bar{t}_+)^2}}$, $s_1 = \frac{\bar{t}_+}{\sqrt{1+(\bar{t}_+)^2}}$, $c_2 \triangleq \frac{-1}{\sqrt{1+(\bar{t}_-)^2}}$, *and* $s_2 \triangleq \frac{-\bar{t}_-}{\sqrt{1+(\bar{t}_-)^2}}$. *Furthermore,* $\bar{t}_+$ *and* $\bar{t}_-$ *are respectively defined as:*

$$\bar{t}_+ \in \arg\min_t p(t) \triangleq \tfrac{a+bt}{\sqrt{1+t^2}} + \tfrac{w+dt}{1+t^2} + h(\tfrac{\mathbf{x}+t\mathbf{y}}{\sqrt{1+t^2}}), \tag{22}$$
$$\bar{t}_- \in \arg\min_t \tilde{p}(t) \triangleq \tfrac{-a-bt}{\sqrt{1+t^2}} + \tfrac{w+dt}{1+t^2} + h(\tfrac{-\mathbf{x}-t\mathbf{y}}{\sqrt{1+t^2}}). \tag{23}$$

*Proof.* We define $w \triangleq c - e$, and $\breve{F}(\tilde{c}, \tilde{s}) \triangleq a\tilde{c} + b\tilde{s} + c\tilde{c}^2 + d\tilde{c}\tilde{s} + e\tilde{s}^2 + h(\tilde{c}\mathbf{x} + \tilde{s}\mathbf{y})$.

With the identity $\sin^2(\theta) = 1 - \cos^2(\theta)$, Problem (21) can be equivalently written as:

$$\bar{\theta} \in \arg\min_{\theta} \quad h(\cos(\theta)\mathbf{x} + \sin(\theta)\mathbf{y}) + a\cos(\theta) + b\sin(\theta)$$
$$+ w\cos^2(\theta) + d\cos(\theta)\sin(\theta) + e. \tag{24}$$

We first consider the case $\cos(\theta) \neq 0$. By Lemma A.8, there are two possible parameterizations for $(\cos(\theta), \sin(\theta))$ in Problem (24).

**Case a).** $\cos(\theta) = \frac{1}{\sqrt{1+\tan^2(\theta)}}$ and $\sin(\theta) = \frac{\tan(\theta)}{\sqrt{1+\tan^2(\theta)}}$. Then Problem (21) becomes:

$$\bar{\theta}_+ \in \arg\min_{\theta} \tfrac{a+\tan(\theta)b}{\sqrt{1+\tan^2(\theta)}} + \tfrac{w+\tan(\theta)d}{1+\tan^2(\theta)} + h(\tfrac{\mathbf{x}+\tan(\theta)\mathbf{y}}{\sqrt{1+\tan^2(\theta)}}).$$

Setting $t = \tan(\theta)$, we have the equivalent problem:

$$\bar{t}_+ \in \arg\min_t \tfrac{a+bt}{\sqrt{1+t^2}} + \tfrac{w+dt}{1+t^2} + h(\tfrac{\mathbf{x}+\mathbf{y}t}{\sqrt{1+t^2}}).$$

Hence the corresponding optimal trigonometric pair is

$$\cos(\bar{\theta}_+) = \tfrac{1}{\sqrt{1+(\bar{t}_+)^2}}, \; \sin(\bar{\theta}_+) = \tfrac{\bar{t}_+}{\sqrt{1+(\bar{t}_+)^2}}. \tag{25}$$

**Case b).** $\cos(\theta) = \frac{-1}{\sqrt{1+\tan(\theta)^2}}$ and $\sin(\theta) = \frac{-\tan(\theta)}{\sqrt{1+\tan(\theta)^2}}$. In this case, Problem (21) reduces to

$$\bar{\theta}_- \in \arg\min_{\theta} \tfrac{-a-\tan(\theta)b}{\sqrt{1+\tan(\theta)^2}} + \tfrac{w+\tan(\theta)d}{1+\tan(\theta)^2} + h(\tfrac{-\mathbf{x}-\tan(\theta)\mathbf{y}}{\sqrt{1+\tan(\theta)^2}}).$$

Again letting $t = \tan \theta$, we obtain

$$\bar{t}_- \in \arg\min_t \frac{-a-bt}{\sqrt{1+t^2}} + \frac{w+dt}{1+t^2} + h\left(\frac{-\mathbf{x}-\mathbf{y}t}{\sqrt{1+t^2}}\right).$$

Thus, the corresponding optimal trigonometric pair is

$$\cos(\bar{\theta}_-) = \frac{-1}{\sqrt{1+(\bar{t}_-)^2}}, \quad \sin(\bar{\theta}_-) = \frac{-\bar{t}_-}{\sqrt{1+(\bar{t}_-)^2}} \tag{26}$$

Combining (25) and (26), when $\cos(\theta) \neq 0$ the optimal solution $\bar{\theta}$ to (24) satisfies $[\cos(\bar{\theta}), \sin(\bar{\theta})] \in \arg\min_{c,s} \check{F}(c,s)$, $s.t.\ [c,s] \in \{[\cos(\bar{\theta}_+), \sin(\bar{\theta}_+)], [\cos(\bar{\theta}_-), \sin(\bar{\theta}_-)]\}$. Including the case $\cos(\theta) = 0$, that is, $[c,s] \in \{[0,1], [0,-1]\}$, the final selection rule for the optimal pair is

$$[\cos(\bar{\theta}), \sin(\bar{\theta})] \in \arg\min_{c,s} \check{F}(c,s),$$
$$s.t.\ [c,s] \in \big\{[\cos(\bar{\theta}_+), \sin(\bar{\theta}_+)], [\cos(\bar{\theta}_-), \sin(\bar{\theta}_-)], [0,1], [0,-1]\big\}.$$

Note that $\{\cos(\bar{\theta}), \sin(\bar{\theta})\}$ uniquely determines $\bar{\theta}$, and the objective in Problem (21) depends only on $\{\cos(\theta), \sin(\theta)\}$ for some $\theta$. Thus, it is not necessary to explicitly recover the angles $\bar{\theta}_+$ and $\bar{\theta}_-$; it suffices to work with their cosine-sine representations.

$\square$

We describe our **BSM** to solve Problem (22); our approach can be naturally extended to tackle Problem (23). **BSM** first identifies all the possible breakpoints / critical points $\Theta$, and then picks the solution that leads to the lowest value as the optimal solution $\bar{t}$, i.e., $\bar{t} \in \arg\min_t p(t),\ s.t.\ t \in \Theta$.

We assume $\mathbf{y}_i \neq 0$. If this is not true and there exists $\mathbf{y}_i = 0$ for some $i$, then $\{\mathbf{x}_i, \mathbf{y}_i\}$ can be removed since it does not affect the minimizer of the problem.

▶ **Finding the Breakpoint Set for** $h(\mathbf{x}) \triangleq \lambda\|\mathbf{x}\|_0$

Since the function $h(\mathbf{x}) \triangleq \lambda\|\mathbf{x}\|_0$ is scale-invariant and symmetric with $\|\pm t\mathbf{x}\|_0 = \|\mathbf{x}\|_0$ for all $t > 0$, Problem (22) reduces to the following problem:

$$\min_t p(t) \triangleq \frac{a+bt}{\sqrt{1+t^2}} + \frac{w+dt}{1+t^2} + \lambda\|\mathbf{x} + t\mathbf{y}\|_0. \tag{27}$$

Given the limiting subdifferential of the $\ell_0$ norm function can be computed as $\partial\|t\|_0 \in \{ \begin{smallmatrix} \mathbb{R}, & t = 0; \\ \{0\}, & \text{else.} \end{smallmatrix} \}$ (see Appendix C.5), we consider the following two cases. **(i)** We assume $(\mathbf{x} + t\mathbf{y})_i = 0$ for some $i$. Then the solution $\bar{t}$ can be determined using $\bar{t} = \frac{\mathbf{x}_i}{\mathbf{y}_i}$. There are $2r$ breakpoints $\{\frac{\mathbf{x}_1}{\mathbf{y}_1}, \frac{\mathbf{x}_2}{\mathbf{y}_2}, ..., \frac{\mathbf{x}_{2r}}{\mathbf{y}_{2r}}\}$ for this case. **(ii)** We now assume $(\mathbf{x} + t\mathbf{y})_i \neq 0$ for all $i$. Then $\lambda\|\mathbf{x} + t\mathbf{y}\|_0 = 2r\lambda$ becomes a constant. Setting the subgradient of $p(t)$ to zero yields: $0 = \nabla p(t) = [b(1+t^2) - (a+bt)t] \cdot \sqrt{1+t^2} \cdot t^\circ + [d(1+t^2) - (w+dt)(2t)] \cdot t^\circ$, where $t^\circ = (1+t^2)^{-2}$. Since $t^\circ > 0$, we obtain: $d(1+t^2) - (w+dt)2t = -(b-at) \cdot \sqrt{1+t^2}$. Squaring both sides, we obtain the following quartic equation: $c_4 t^4 + c_3 t^3 + c_2 t^2 + c_1 t + c_0 = 0$ for some suitable $c_4, c_3, c_2, c_1$ and $c_0$. Solving this equation analytically using Lodovico Ferrari's method (WikiContributors), we obtain all its real roots $\{\bar{t}_1, \bar{t}_2, ..., \bar{t}_j\}$ with $1 \leq j \leq 4$. There are at most 4 breakpoints for this case. Therefore, Problem (27) contains at most $2r + 4$ breakpoints $\Theta = \{\frac{\mathbf{x}_1}{\mathbf{y}_1}, \frac{\mathbf{x}_2}{\mathbf{y}_2}, ..., \frac{\mathbf{x}_{2r}}{\mathbf{y}_{2r}}, \bar{t}_1, \bar{t}_2, ..., \bar{t}_j\}$.

▶ **Finding the Breakpoint Set for** $h(\mathbf{x}) \triangleq \lambda\|\mathbf{x}\|_1$

Since the function $h(\mathbf{x}) \triangleq \lambda\|\mathbf{x}\|_1$ is symmetric, Problem (22) reduces to the following problem:

$$\bar{t} \in \arg\min_t p(t) \triangleq \frac{a+bt}{\sqrt{1+t^2}} + \frac{w+dt}{1+t^2} + \frac{\lambda\|\mathbf{x} + t\mathbf{y}\|_1}{\sqrt{1+t^2}}. \tag{28}$$

Setting the subgradient of $p(\cdot)$ to zero yields: $0 \in \partial p(t) = t^\circ[d(1+t^2) - (w+dt)2t + (b-at) \cdot \sqrt{1+t^2}] + t^\circ \lambda \cdot \sqrt{1+t^2} \cdot [\langle\text{sign}(\mathbf{x} + t\mathbf{y}), \mathbf{y}\rangle(1+t^2) - \|\mathbf{x} + t\mathbf{y}\|_1 t]$, where $t^\circ = (1+t^2)^{-2}$. We consider the following two cases. **(i)** We assume $(\mathbf{x} + t\mathbf{y})_i = 0$ for some $i$. Then the solution $\bar{t}$ can be determined using $\bar{t} = \frac{\mathbf{x}_i}{\mathbf{y}_i}$. There are $2r$ breakpoints $\{\frac{\mathbf{x}_1}{\mathbf{y}_1}, \frac{\mathbf{x}_2}{\mathbf{y}_2}, ..., \frac{\mathbf{x}_{2r}}{\mathbf{y}_{2r}}\}$ for this case. **(ii)** We

now assume $(\mathbf{x} + t\mathbf{y})_i \neq 0$ for all $i$. We define $\mathbf{z} \triangleq \{+\frac{\mathbf{x}_1}{\mathbf{y}_1}, -\frac{\mathbf{x}_1}{\mathbf{y}_1}, +\frac{\mathbf{x}_2}{\mathbf{y}_2}, -\frac{\mathbf{x}_2}{\mathbf{y}_2}, ..., +\frac{\mathbf{x}_{2r}}{\mathbf{y}_{2r}}, -\frac{\mathbf{x}_{2r}}{\mathbf{y}_{2r}}\} \in \mathbb{R}^{4r \times 1}$, and sort $\mathbf{z}$ in non-descending order. Given $\bar{t} \neq \mathbf{z}_i$ for all $i$ in this case, the domain $p(t)$ can be divided into $(4r + 1)$ non-overlapping intervals: $(-\infty, \mathbf{z}_1), (\mathbf{z}_1, \mathbf{z}_2), ..., (\mathbf{z}_{4r}, +\infty)$. In each interval, $\text{sign}(\mathbf{x} + t\mathbf{y}) \triangleq \mathbf{o}$ can be determined. Combining with the fact that $t^\circ > 0$ and $\|\mathbf{x} + t\mathbf{y}\|_1 = \langle \mathbf{o}, \mathbf{x} + t\mathbf{y} \rangle$, the first-order optimality condition reduces to: $0 = [d(1 + t^2) - (w + dt)2t + (b - at) \cdot \sqrt{1 + t^2}] + \lambda \cdot \sqrt{1 + t^2} \cdot [\langle \mathbf{o}, \mathbf{y} \rangle(1 + t^2) - \langle \mathbf{o}, \mathbf{x} + t\mathbf{y} \rangle t]$, which can be simplified as: $(at - b) \cdot \sqrt{1 + t^2} - \lambda \cdot \sqrt{1 + t^2} \cdot [\langle \mathbf{o}, \mathbf{y} - t\mathbf{x} \rangle] = [d(1 + t^2) - (w + dt)2t]$. We square both sides and then solve the quartic equation. We obtain obtain all its real roots $\{\bar{t}_1, \bar{t}_2, ..., \bar{t}_j\}$ with $1 \leq j \leq 4$. Therefore, Problem (28) contains at most $2r + (4r + 1) \times 4$ breakpoints.

▶ **Finding the Breakpoint Set for** $h(\mathbf{x}) \triangleq I_{\geq 0}(\mathbf{x})$

Since the function $h(\mathbf{x}) \triangleq \iota_{\geq 0}(\mathbf{x})$ is scale-invariant with $h(t\mathbf{x}) = h(\mathbf{x})$ forall $t \geq 0$, Problem (22) reduces to the following problem:

$$\bar{t} \in \arg\min_t p(t) \triangleq \frac{a + bt}{\sqrt{1 + t^2}} + \frac{w + dt}{1 + t^2}, \ s.t. \ \mathbf{x} + t\mathbf{y} \geq \mathbf{0}. \tag{29}$$

We define $I \triangleq \{i | \mathbf{y}_i > 0\}$ and $J \triangleq \{i | \mathbf{y}_i < 0\}$. It is not difficult to verity that $\{x + t\mathbf{y} \geq 0\} \Leftrightarrow \{-\frac{\mathbf{x}_I}{\mathbf{y}_I} \leq t, t \leq -\frac{\mathbf{x}_J}{\mathbf{y}_J}\} \Leftrightarrow \{lb \triangleq \max(-\frac{\mathbf{x}_I}{\mathbf{y}_I}) \leq t \leq \min(-\frac{\mathbf{x}_J}{\mathbf{y}_J}) \triangleq ub\}$. When $lb > ub$, we can directly conclude that the problem has no solution for this case. Now we assume $ub \geq lb$ and define $P(t) \triangleq \min(ub, \max(t, lb))$. We omit the bound constraint and set the gradient of $p(t)$ to zero, which yields: $0 = \nabla p(t) = [b(1 + t^2) - (a + bt)t] \cdot \sqrt{1 + t^2} \cdot t^\circ + [d(1 + t^2) - (w + dt)(2t)] \cdot t^\circ$, where $t^\circ = (1 + t^2)^{-2}$. We obtain all its real roots $\{\bar{t}_1, \bar{t}_2, ..., \bar{t}_j\}$ with $1 \leq j \leq 4$ after squaring both sides and solving the quartic equation. Combining with the bound constraints, we conclude that Problem (29) contains at most $(4 + 2)$ breakpoints $\{P(\bar{t}_1), P(\bar{t}_2), ..., P(\bar{t}_j), lb, ub\}$ with $1 \leq j \leq 4$.

## C  ADDITIONAL DISCUSSIONS

This section encompasses various discussions, covering topics such as: (*i*) simple examples for the optimality hierarchy, (*ii*) computation of the matrix $\mathbf{Q}$, (*iii*) complexity comparison between **OBCD** and full gradient methods, (*iv*) generalization to multiple row updates, and (*v*) the subdifferential of the cardinality function.

### C.1  SIMPLE EXAMPLES FOR THE OPTIMALITY HIERARCHY

To demonstrate the strong optimality of $BS_2$-points and the advantages of the proposed method, we examine the following simple examples of $2 \times 2$ optimization problems mentioned in the paper:

$$\min_{\mathbf{V} \in \text{St}(2,2)} F(\mathbf{V}) \triangleq \|\mathbf{V} - \mathbf{A}\|_\mathsf{F}^2, \ \text{with } \mathbf{A} = \left(\begin{smallmatrix} 1 & 0 \\ -1 & -1 \end{smallmatrix}\right). \tag{30}$$

$$\min_{\mathbf{V} \in \text{St}(2,2)} F(\mathbf{V}) \triangleq \|\mathbf{V} - \mathbf{B}\|_\mathsf{F}^2 + 5\|\mathbf{V}\|_1, \ \text{with } \mathbf{B} = \left(\begin{smallmatrix} 1 & 0 \\ 1 & 2 \end{smallmatrix}\right). \tag{31}$$

Figure 2 shows the geometric visualizations of Problems (30) and (31) using the relation $\min_\theta \min(F(\mathbf{V}_\theta^{\text{rot}}), F(\mathbf{V}_\theta^{\text{ref}})) = \min_{\mathbf{V} \in \text{St}(2,2)} F(\mathbf{V})$. The two objective functions exhibit periodicity with a period of $2\pi$. Within the interval $[0, 2\pi)$, each of them contains one unique $BS_2$-*point*, while the two respective examples contain 4 and 8 critical points. Therefore, the optimality condition of $BS_2$-points might be much stronger than that of critical points.

**$BS_2$-points vs. Critical Point: Algorithm Instance Study**. We briefly review methods that seek critical points of Problem (30) and demonstrate that they may lead to suboptimal solutions for Problem (30). As a representative example, we consider the well-known feasible method based on the Cayley transform (Wen & Yin, 2013). According to Equation (7) in (Wen & Yin, 2013), the update rule is:

$$\mathbf{X}^{t+1} \Leftarrow \mathbf{Q}\mathbf{X}^t, \ \mathbf{Q} \triangleq [(\mathbf{I}_2 + \tfrac{\tau}{2}\tilde{\mathbf{A}})^{-1}(\mathbf{I}_2 - \tfrac{\tau}{2}\tilde{\mathbf{A}})], \tag{32}$$

where $\tau \in \mathbb{R}$, and $\tilde{\mathbf{A}} \in \mathbb{R}^{2 \times 2}$ is a suitable skew-symmetric matrix. Lemma A.3 shows that the matrix $\mathbf{Q}$ is always a rotation matrix. Consequently, if $\mathbf{X}^0$ is initialized as a rotation matrix with $\det(\mathbf{Q}) = 1$, all iterates $\mathbf{X}^{t+1}$ remain rotation matrices, which in general do not coincide with the optimal solution.

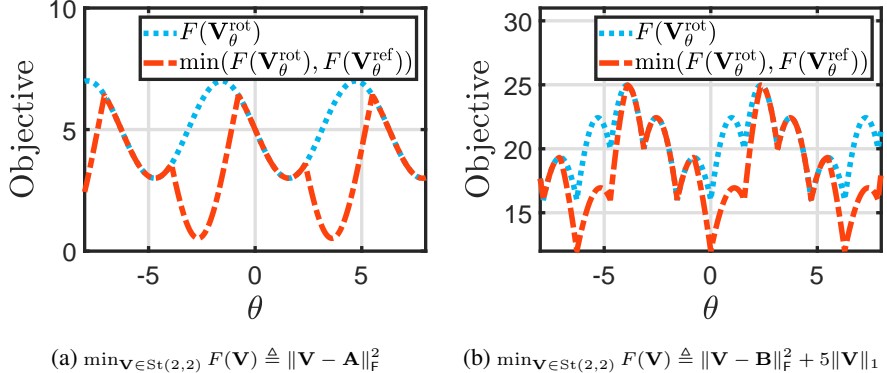

(a) $\min_{\mathbf{V} \in \mathrm{St}(2,2)} F(\mathbf{V}) \triangleq \|\mathbf{V} - \mathbf{A}\|_{\mathsf{F}}^2$  (b) $\min_{\mathbf{V} \in \mathrm{St}(2,2)} F(\mathbf{V}) \triangleq \|\mathbf{V} - \mathbf{B}\|_{\mathsf{F}}^2 + 5\|\mathbf{V}\|_1$

Figure 2: Geometric Visualizations of Two Examples of $2 \times 2$ Optimization Problems with Orthogonality Constraints with $\mathbf{A} = \left( \begin{smallmatrix} 1 & 0 \\ -1 & -1 \end{smallmatrix} \right)$ and $\mathbf{B} = \left( \begin{smallmatrix} 1 & 0 \\ 1 & 2 \end{smallmatrix} \right)$.

## C.2 COMPUTING THE MATRIX $\mathbf{Q}$

Computing the matrix $\mathbf{Q} \in \mathbb{R}^{k^2 \times k^2}$ as in (8) can be a challenging task because it involves the matrix $\mathbf{H} \in \mathbb{R}^{nr \times nr}$. However, in practice, $\mathbf{H}$ often has some special structure that enables fast matrix computation. For example, $\mathbf{H}$ might take a diagonal matrix that is equal to $L\mathbf{I}_{nr}$ for some $L \geq 0$ or has a Kronecker structure where $\mathbf{H} = \mathbf{H}_1 \otimes \mathbf{H}_2$ for some $\mathbf{H}_1 \in \mathbb{R}^{r \times r}$ and $\mathbf{H}_2 \in \mathbb{R}^{n \times n}$. The lemmas provided below demonstrate how to compute the matrix $\mathbf{Q}$.

**Lemma C.1.** *Assume (8) is used to find* $\mathbf{Q}$. *(a) If* $\mathbf{H} = \mathbf{H}_1 \otimes \mathbf{H}_2$, *we have:* $\mathbf{Q} = \mathbf{Q}_1 \otimes \mathbf{Q}_2$, *where* $\mathbf{Q}_1 = \mathbf{Z}\mathbf{H}_1\mathbf{Z}^{\mathsf{T}} \in \mathbb{R}^{k \times k}$ *and* $\mathbf{Q}_2 = \mathrm{U}_{\mathrm{B}}^{\mathsf{T}}\mathbf{H}_2\mathrm{U}_{\mathrm{B}} \in \mathbb{R}^{k \times k}$. *(b) If* $\mathbf{H} = L\mathbf{I}_{nr}$, *we have* $\mathbf{Q} = (L\mathbf{Z}\mathbf{Z}^{\mathsf{T}}) \otimes \mathbf{I}_k$.

*Proof.* Recall that for any matrices $\bar{\mathbf{A}}, \bar{\mathbf{B}}, \bar{\mathbf{C}}, \bar{\mathbf{D}}$ of suitable dimensions, we have the following equality: $(\bar{\mathbf{A}} \otimes \bar{\mathbf{B}})(\bar{\mathbf{C}} \otimes \bar{\mathbf{D}}) = (\bar{\mathbf{A}}\bar{\mathbf{C}}) \otimes (\bar{\mathbf{B}}\bar{\mathbf{D}})$.

**(a)** If $\mathbf{H} = \mathbf{H}_1 \otimes \mathbf{H}_2$, we derive: $\mathbf{Q} \triangleq (\mathbf{Z}^{\mathsf{T}} \otimes \mathrm{U}_{\mathrm{B}})^{\mathsf{T}}\mathbf{H}(\mathbf{Z}^{\mathsf{T}} \otimes \mathrm{U}_{\mathrm{B}}) = (\mathbf{Z}^{\mathsf{T}} \otimes \mathrm{U}_{\mathrm{B}})^{\mathsf{T}}(\mathbf{H}_1 \otimes \mathbf{H}_2)(\mathbf{Z}^{\mathsf{T}} \otimes \mathrm{U}_{\mathrm{B}}) = (\mathbf{Z}^{\mathsf{T}} \otimes \mathrm{U}_{\mathrm{B}})^{\mathsf{T}}[(\mathbf{H}_1\mathbf{Z}^{\mathsf{T}}) \otimes (\mathbf{H}_2\mathrm{U}_{\mathrm{B}})] = (\mathbf{Z}\mathbf{H}_1\mathbf{Z}^{\mathsf{T}}) \otimes (\mathrm{U}_{\mathrm{B}}^{\mathsf{T}}\mathbf{H}_2\mathrm{U}_{\mathrm{B}}) = \mathbf{Q}_1 \otimes \mathbf{Q}_2$.

**(b)** If $\mathbf{H} = L\mathbf{I}_{nr}$, we have: $\mathbf{Q} \triangleq (\mathbf{Z}^{\mathsf{T}} \otimes \mathrm{U}_{\mathrm{B}})^{\mathsf{T}}\mathbf{H}(\mathbf{Z}^{\mathsf{T}} \otimes \mathrm{U}_{\mathrm{B}}) = L(\mathbf{Z}^{\mathsf{T}} \otimes \mathrm{U}_{\mathrm{B}})^{\mathsf{T}}(\mathbf{Z}^{\mathsf{T}} \otimes \mathrm{U}_{\mathrm{B}}) = L(\mathbf{Z}\mathbf{Z}^{\mathsf{T}}) \otimes \mathbf{I}_k$.

$\square$

**Lemma C.2.** *Assume (9) is used to find* $\mathbf{Q}$. *(a) If* $\mathbf{H} = \mathbf{H}_1 \otimes \mathbf{H}_2$, *we have* $\mathbf{Q} = \|\mathbf{Q}_1\|_{\mathsf{sp}} \cdot \|\mathbf{Q}_2\|_{\mathsf{sp}} \cdot \mathbf{I}$, *where* $\mathbf{Q}_1$ *and* $\mathbf{Q}_2$ *are defined in Lemma C.1. (b) If* $\mathbf{H} = L\mathbf{I}_{nr}$, *we have* $\mathbf{Q} = L\|\mathbf{Z}\|_{\mathsf{sp}}^2 \cdot \mathbf{I}$.

*Proof.* **(a)** Using the results Lemma C.1(a), we have: $(\mathbf{Z}^{\mathsf{T}} \otimes \mathrm{U}_{\mathrm{B}})^{\mathsf{T}}\mathbf{H}(\mathbf{Z}^{\mathsf{T}} \otimes \mathrm{U}_{\mathrm{B}}) = \mathbf{Q}_1 \otimes \mathbf{Q}_2 \preceq \|\mathbf{Q}_1\|_{\mathsf{sp}} \cdot \|\mathbf{Q}_2\|_{\mathsf{sp}} \cdot \mathbf{I}$.

**(b)** Using the results in Claim (*b*) of Lemma C.1, we have: $(\mathbf{Z}^{\mathsf{T}} \otimes \mathrm{U}_{\mathrm{B}})^{\mathsf{T}}\mathbf{H}(\mathbf{Z}^{\mathsf{T}} \otimes \mathrm{U}_{\mathrm{B}}) = L\mathbf{Z}\mathbf{Z}^{\mathsf{T}} \otimes \mathbf{I}_k \preceq L\|\mathbf{Z}\|_{\mathsf{sp}}^2 \cdot \mathbf{I}$.

$\square$

## C.3 COMPLEXITY COMPARISON BETWEEN **OBCD** AND FULL GRADIENT METHODS

We present a computational complexity comparison with full gradient methods using the linear eigenvalue problem: $\min_{\mathbf{X}} F(\mathbf{X}) \triangleq \frac{1}{2}\langle \mathbf{X}, \mathbf{C}\mathbf{X}\rangle$, *s.t.* $\mathbf{X}^{\mathsf{T}}\mathbf{X} = \mathbf{I}_r$, where $\mathbf{C} \in \mathbb{R}^{n \times n}$ is given.

We first examine full gradient methods such as the Riemannian gradient method (Jiang & Dai, 2015; Liu et al., 2016). The computation of the Riemannian gradient $\nabla_{\mathcal{M}}F(\mathbf{X}) = \mathbf{C}\mathbf{X} - \mathbf{X}[\mathbf{C}\mathbf{X}]^{\mathsf{T}}\mathbf{X}$ requires $\mathcal{O}(n^2 r)$ time, while the retraction step using SVD, QR, or polar decomposition demands

$\mathcal{O}(nr^2)$. Consequently, the overall complexity for Riemannian gradient method is $N_1 \times \mathcal{O}(n^2r)$, where $N_1$ is the number of iterations required for convergence.

We now consider the proposed **OBCD** method where the matrix $\mathbf{Q}$ is chosen to be a diagonal matrix as in Equality (9). (***i***) We adopt an incremental update strategy for computing the Euclidean gradient $\nabla F(\mathbf{X}) = \mathbf{CX}$, maintaining the relationship $\mathbf{Y}^t = \mathbf{CX}^t$ for all $t$. The initialization $\mathbf{Y}^0 = \mathbf{CX}^0$ occurs only once. When $\mathbf{X}^t$ is updated via a $k$-row change, resulting in $\mathbf{X}^{t+1} = \mathbf{X}^t + \mathrm{U}_{\mathrm{B}}(\mathbf{V} - \mathbf{I})\mathrm{U}_{\mathrm{B}}^\mathsf{T}\mathbf{X}^t$, we efficiently reconstruct $\mathbf{CX}^{t+1}$ by updating $\mathbf{Y}^{t+1} = \mathbf{Y}^t + \mathbf{C}\mathrm{U}_{\mathrm{B}}(\mathbf{V} - \mathbf{I})\mathrm{U}_{\mathrm{B}}^\mathsf{T}\mathbf{X}^t$ in $\mathcal{O}(nr)$ time. (***ii***) Computing the matrix $\mathbf{P}$ as shown in (3) involves matrix multiplication between matrices $[\nabla f(\mathbf{X}^t)]_{\mathrm{B}:} \in \mathbb{R}^{k \times r}$ and $[[\mathbf{X}^t]_{\mathrm{B}:}]^\mathsf{T} \in \mathbb{R}^{r \times k}$, which can be done in $\mathcal{O}(rk^2)$. (***iii***) Solving the subproblem using small-size SVD takes $\mathcal{O}(k^3)$ time. Thus, the total complexity for **OBCD** is $N_2 \times \mathcal{O}(nr + rk^2 + k^3)$, with $N_2$ denoting the number of **OBCD** iterations.

## C.4 GENERALIZATION TO MULTIPLE ROW UPDATES

The proposed **OBCD** algorithm can be generalized to multiple row updates scheme.

Assume that $n$ is an even number, and $k = 2$. As mentioned in Lemma 2.3, when (9) is used to find $\mathbf{Q}$, the subproblem $\bar{\mathbf{V}}^t \in \arg\min_{\mathbf{V} \in \mathrm{St}(k,k)} \mathcal{K}(\mathbf{V}; \mathbf{X}^t, \mathrm{B})$ in Algorithm 1 reduces to:

$$\min_{\mathbf{V} \in \mathrm{St}(2,2)} \langle \mathbf{V}, (\nabla f(\mathbf{X}^t)[\mathbf{X}^t]^\mathsf{T})_{\mathrm{BB}} \rangle + h(\mathbf{V}\mathrm{U}_{\mathrm{B}}\mathbf{X}^t). \tag{33}$$

One can independently solve $(n/2)$ subproblems, each formulated as follows:

$\min_{\mathbf{V} \in \mathrm{St}(2,2)} \langle \mathbf{V}, (\nabla f(\mathbf{X}^t)[\mathbf{X}^t]^\mathsf{T})_{\mathrm{BB}} \rangle + h(\mathbf{V}\mathrm{U}_{\mathrm{B}}\mathbf{X}^t)$ with $\mathrm{B} = [1, 2]$.

$\min_{\mathbf{V} \in \mathrm{St}(2,2)} \langle \mathbf{V}, (\nabla f(\mathbf{X}^t)[\mathbf{X}^t]^\mathsf{T})_{\mathrm{BB}} \rangle + h(\mathbf{V}\mathrm{U}_{\mathrm{B}}\mathbf{X}^t)$ with $\mathrm{B} = [3, 4]$.

$\cdots$

$\min_{\mathbf{V} \in \mathrm{St}(2,2)} \langle \mathbf{V}, (\nabla f(\mathbf{X}^t)[\mathbf{X}^t]^\mathsf{T})_{\mathrm{BB}} \rangle + h(\mathbf{V}\mathrm{U}_{\mathrm{B}}\mathbf{X}^t)$ with $\mathrm{B} = [n - 1, n]$.

This approach, known as the Jacobi update in the literature, allows for the parallel update of $n$ rows of the matrix $\mathbf{X}$.

Notably, one can consider $k \triangleq |\mathrm{B}| > 2$ when $h(\cdot) = 0$, and the associated subproblems can be solved using SVD.

## C.5 LIMITING SUBDIFFERENTIAL OF THE CARDINALITY FUNCTION

We demonstrate how to calculate the limiting subdifferential of the cardinality function $h(\mathbf{X}) = \|\mathbf{X}\|_0$. Given that $h(\mathbf{X}) = \|\mathbf{X}\|_0$ is coordinate-wise separable, we focus only on the scalar function $h(x) = |x|_0$, where $|x|_0 = \left\{ \begin{smallmatrix} 0, & x = 0; \\ 1, & \text{else.} \end{smallmatrix} \right\}$.

The Fréchet subdifferential of the function $h(x) = |x|_0$ at $x \in \mathrm{dom}(h)$ is defined as $\hat{\partial}h(x) \triangleq \{\xi \in \mathbb{R} : \lim_{z \to x} \inf_{z \neq x} \frac{h(z) - h(x) - \langle \xi, z - x \rangle}{|z - x|} \geq 0\}$, while the limiting subdifferential of $h(x)$ at $x \in \mathrm{dom}(h)$ is denoted as $\partial h(x) \triangleq \{\xi \in \mathbb{R} : \exists x^t \to x, h(x^t) \to h(x), \xi^t \in \hat{\partial}h(x^t) \to \xi, \forall t\}$. We consider the following two cases. (***i***) $x \neq 0$. We have: $\hat{\partial}h(x) = \{\xi \in \mathbb{R} : \lim_{z \to x} \inf_{z \neq x} \frac{-\langle \xi, z - x \rangle}{|z - x|} \geq 0\} = \{0\}$. (***ii***) $x = 0$. We have: $\hat{\partial}h(x) = \{\xi \in \mathbb{R} : \lim_{z \to x} \inf_{z \neq x} \frac{|z|_0 - \langle \xi, z - x \rangle}{|z - x|} \geq 0\} = \{\xi \in \mathbb{R} : \lim_{z \to x} \inf_{z \neq x} \frac{1 - \langle \xi, z \rangle}{|z|} \geq 0\} = \mathbb{R}$.

We therefore conclude that $[\partial \|\mathbf{X}\|_0]_{i,j} \in \left\{ \begin{smallmatrix} \mathbb{R}, & \\ \{0\}, & \end{smallmatrix} \begin{smallmatrix} \mathbf{X}_{i,j} = 0; \\ \text{else.} \end{smallmatrix} \right\}$ for all $i \in [n]$ and $j \in [r]$.

# D PROOF FOR SECTION 2

## D.1 PROOF FOR LEMMA 2.1

*Proof.* **Part (a)**. For any $\mathbf{V} \in \mathbb{R}^{k \times k}$ and $\mathrm{B} \in \{\mathcal{B}_i\}_{i=1}^{\mathrm{C}_n^k}$, we have:

$$
[\mathbf{X}^+]^\mathsf{T} \mathbf{X}^+ - \mathbf{X}^\mathsf{T} \mathbf{X}
$$

$$
\overset{\textcircled{1}}{=} [\mathbf{X} + \mathrm{U}_\mathrm{B}(\mathbf{V} - \mathbf{I}_k)\mathrm{U}_\mathrm{B}^\mathsf{T}\mathbf{X}]^\mathsf{T}[\mathbf{X} + \mathrm{U}_\mathrm{B}(\mathbf{V} - \mathbf{I}_k)\mathrm{U}_\mathrm{B}^\mathsf{T}\mathbf{X}] - \mathbf{X}^\mathsf{T}\mathbf{X}
$$

$$
= \mathbf{X}^\mathsf{T}\mathrm{U}_\mathrm{B}(\mathbf{V} - \mathbf{I}_k)\mathrm{U}_\mathrm{B}^\mathsf{T}\mathbf{X} + [\mathrm{U}_\mathrm{B}(\mathbf{V} - \mathbf{I}_k)\mathrm{U}_\mathrm{B}^\mathsf{T}\mathbf{X}]^\mathsf{T}\mathbf{X} + [\mathrm{U}_\mathrm{B}(\mathbf{V} - \mathbf{I}_k)\mathrm{U}_\mathrm{B}^\mathsf{T}\mathbf{X}]^\mathsf{T}[\mathrm{U}_\mathrm{B}(\mathbf{V} - \mathbf{I}_k)\mathrm{U}_\mathrm{B}^\mathsf{T}\mathbf{X}]
$$

$$
= \mathbf{X}^\mathsf{T}\mathrm{U}_\mathrm{B}\left[(\mathbf{V} - \mathbf{I}_k + \mathbf{V}^\mathsf{T} - \mathbf{I}_k) + (\mathbf{V} - \mathbf{I}_k)^\mathsf{T}\mathrm{U}_\mathrm{B}^\mathsf{T}\mathrm{U}_\mathrm{B}(\mathbf{V} - \mathbf{I}_k)\right]\mathrm{U}_\mathrm{B}^\mathsf{T}\mathbf{X}
$$

$$
\overset{\textcircled{2}}{=} \mathbf{X}^\mathsf{T}\mathrm{U}_\mathrm{B}\left[(\mathbf{V} - \mathbf{I}_k + \mathbf{V}^\mathsf{T} - \mathbf{I}_k) + (\mathbf{V} - \mathbf{I}_k)^\mathsf{T}(\mathbf{V} - \mathbf{I}_k)\right]\mathrm{U}_\mathrm{B}^\mathsf{T}\mathbf{X}
$$

$$
= \mathbf{X}^\mathsf{T}\mathrm{U}_\mathrm{B}(\mathbf{V} - \mathbf{I}_k + \mathbf{V}^\mathsf{T} - \mathbf{I}_k + \mathbf{V}^\mathsf{T}\mathbf{V} - \mathbf{V}^\mathsf{T} - \mathbf{V} + \mathbf{I}_k)\mathrm{U}_\mathrm{B}^\mathsf{T}\mathbf{X}
$$

$$
= \mathbf{X}^\mathsf{T}\mathrm{U}_\mathrm{B}(-\mathbf{I}_k + \mathbf{V}^\mathsf{T}\mathbf{V})\mathrm{U}_\mathrm{B}^\mathsf{T}\mathbf{X}
$$

$$
\overset{\textcircled{3}}{=} \mathbf{X}^\mathsf{T}\mathrm{U}_\mathrm{B} \cdot \mathbf{0} \cdot \mathrm{U}_\mathrm{B}^\mathsf{T}\mathbf{X}
$$

$$
= \mathbf{0},
$$

where step ① uses $\mathbf{X}^+ = \mathbf{X} + \mathrm{U}_\mathrm{B}(\mathbf{V} - \mathbf{I}_k)\mathrm{U}_\mathrm{B}^\mathsf{T}\mathbf{X}$; step ② uses $\mathrm{U}_\mathrm{B}^\mathsf{T}\mathrm{U}_\mathrm{B} = \mathbf{I}_k$; step ③ uses $\mathbf{V}^\mathsf{T}\mathbf{V} = \mathbf{I}_k$.

**Part (b)**. Obvious.

$\square$

## D.2 PROOF OF LEMMA 2.2

*Proof.* We define $\mathbf{X}^+ \triangleq \mathbf{X} + \mathrm{U}_\mathrm{B}(\mathbf{V} - \mathbf{I}_k)\mathrm{U}_\mathrm{B}^\mathsf{T}\mathbf{X}$, $\underline{\mathbf{Q}} \triangleq (\mathbf{Z}^\mathsf{T} \otimes \mathrm{U}_\mathrm{B})^\mathsf{T}\mathbf{H}(\mathbf{Z}^\mathsf{T} \otimes \mathrm{U}_\mathrm{B})$, and $\mathbf{Z} \triangleq \mathrm{U}_\mathrm{B}^\mathsf{T}\mathbf{X}$.

**Part (a)**. We derive the following results:

$$
\begin{aligned}
\|\mathbf{X}^+ - \mathbf{X}\|_\mathbf{H}^2 &\overset{\textcircled{1}}{=} \|\mathrm{U}_\mathrm{B}(\mathbf{V} - \mathbf{I}_k)\mathbf{Z}\|_\mathbf{H}^2 \\
&\overset{\textcircled{2}}{=} \mathrm{vec}(\mathrm{U}_\mathrm{B}(\mathbf{V} - \mathbf{I}_k)\mathbf{Z})^\mathsf{T}\mathbf{H}\mathrm{vec}(\mathrm{U}_\mathrm{B}(\mathbf{V} - \mathbf{I}_k)\mathbf{Z}) \\
&\overset{\textcircled{3}}{=} \mathrm{vec}(\mathbf{V} - \mathbf{I}_k)^\mathsf{T}(\mathbf{Z}^\mathsf{T} \otimes \mathrm{U}_\mathrm{B})^\mathsf{T}\mathbf{H}(\mathbf{Z}^\mathsf{T} \otimes \mathrm{U}_\mathrm{B})\mathrm{vec}(\mathbf{V} - \mathbf{I}_k) \\
&\overset{\textcircled{4}}{=} \|\mathbf{V} - \mathbf{I}_k\|_{(\mathbf{Z}^\mathsf{T} \otimes \mathrm{U}_\mathrm{B})^\mathsf{T}\mathbf{H}(\mathbf{Z}^\mathsf{T} \otimes \mathrm{U}_\mathrm{B})}^2 \\
&\overset{\textcircled{5}}{=} \|\mathbf{V} - \mathbf{I}_k\|_{\underline{\mathbf{Q}}}^2,
\end{aligned}
$$

where step ① uses $\mathbf{X}^+ \triangleq \mathbf{X} + \mathrm{U}_\mathrm{B}(\mathbf{V} - \mathbf{I}_k)\mathbf{Z}$; step ② uses $\|\mathbf{X}\|_\mathbf{H}^2 = \mathrm{vec}(\mathbf{X})^\mathsf{T}\mathbf{H}\mathrm{vec}(\mathbf{X})$; step ③ uses $(\mathbf{Z}^\mathsf{T} \otimes \mathbf{R})\mathrm{vec}(\mathbf{U}) = \mathrm{vec}(\mathbf{R}\mathbf{U}\mathbf{Z})$ for all $\mathbf{R}$, $\mathbf{Z}$, and $\mathbf{U}$ of suitable dimensions; step ④ uses $\|\mathbf{X}\|_\mathbf{H}^2 = \mathrm{vec}(\mathbf{X})^\mathsf{T}\mathbf{H}\mathrm{vec}(\mathbf{X})$ again; step ⑤ uses the definition of $\underline{\mathbf{Q}}$.

**Part (b)**. We derive the following equalities:

$$
\begin{aligned}
\|\mathbf{X}^+ - \mathbf{X}\|_\mathsf{F}^2 &\overset{\textcircled{1}}{=} \|\mathrm{U}_\mathrm{B}(\mathbf{V} - \mathbf{I}_k)\mathbf{Z}\|_\mathsf{F}^2 \\
&\overset{\textcircled{2}}{=} \|(\mathbf{V} - \mathbf{I}_k)\mathbf{Z}\|_\mathsf{F}^2 \\
&= \langle(\mathbf{V} - \mathbf{I}_k)^\mathsf{T}(\mathbf{V} - \mathbf{I}_k), \mathbf{Z}\mathbf{Z}^\mathsf{T}\rangle \\
&\overset{\textcircled{3}}{=} 2\langle\mathbf{I}_k - \mathbf{V}, \mathbf{Z}\mathbf{Z}^\mathsf{T}\rangle + \langle\mathbf{V} - \mathbf{V}^\mathsf{T}, \mathbf{Z}\mathbf{Z}^\mathsf{T}\rangle. \\
&\overset{\textcircled{4}}{=} 2\langle\mathbf{I}_k - \mathbf{V}, \mathbf{Z}\mathbf{Z}^\mathsf{T}\rangle + 0.
\end{aligned}
$$

where step ① uses $\mathbf{X}^+ \triangleq \mathbf{X} + \mathrm{U}_\mathrm{B}(\mathbf{V} - \mathbf{I}_k)\mathbf{Z}$; step ② uses the fact that $\|\mathrm{U}_\mathrm{B}\mathbf{V}\|_\mathsf{F}^2 = \|\mathbf{V}\|_\mathsf{F}^2$ for any $\mathbf{V} \in \mathbb{R}^{k \times k}$; step ③ uses

$$
(\mathbf{V} - \mathbf{I}_k)^\mathsf{T}(\mathbf{V} - \mathbf{I}_k) = \mathbf{I}_k - \mathbf{V}^\mathsf{T} - \mathbf{V} + \mathbf{I}_k = 2(\mathbf{I}_k - \mathbf{V}) + (\mathbf{V} - \mathbf{V}^\mathsf{T});
$$

step ④ uses the fact that $\langle\mathbf{V}, \mathbf{Z}\mathbf{Z}^\mathsf{T}\rangle = \langle\mathbf{V}^\mathsf{T}, (\mathbf{Z}\mathbf{Z}^\mathsf{T})^\mathsf{T}\rangle = \langle\mathbf{V}^\mathsf{T}, \mathbf{Z}\mathbf{Z}^\mathsf{T}\rangle$ which holds true as the matrix $\mathbf{Z}\mathbf{Z}^\mathsf{T}$ is symmetric.

**Part (c)**. We have:

$$
\begin{aligned}
\|\mathbf{X}^+ - \mathbf{X}\|_{\mathsf{F}}^2 &= \|\mathrm{U}_{\mathrm{B}}(\mathbf{V} - \mathbf{I}_k)\mathrm{U}_{\mathrm{B}}^{\mathsf{T}}\mathbf{X}\|_{\mathsf{F}}^2 \\
&\overset{①}{\leq} \|\mathrm{U}_{\mathrm{B}}\|_{\mathsf{sp}}^2 \cdot \|(\mathbf{V} - \mathbf{I}_k)\mathrm{U}_{\mathrm{B}}^{\mathsf{T}}\mathbf{X}\|_{\mathsf{F}}^2 \\
&\overset{②}{\leq} \|\mathrm{U}_{\mathrm{B}}\|_{\mathsf{sp}}^2 \cdot \|\mathbf{V} - \mathbf{I}_k\|_{\mathsf{F}}^2 \cdot \|\mathrm{U}_{\mathrm{B}}^{\mathsf{T}}\|_{\mathsf{sp}}^2 \cdot \|\mathbf{X}\|_{\mathsf{sp}}^2 \\
&\overset{③}{=} \|\mathbf{V} - \mathbf{I}_k\|_{\mathsf{F}}^2 \\
&\overset{④}{=} 2\langle \mathbf{I}_k - \mathbf{V}, \mathbf{I}_k \rangle,
\end{aligned}
$$

where step ① and step ② uses the norm inequality that $\|\mathbf{A}\mathbf{X}\|_{\mathsf{F}} \leq \|\mathbf{A}\|_{\mathsf{F}} \cdot \|\mathbf{X}\|_{\mathsf{sp}}$ for any $\mathbf{A}$ and $\mathbf{X}$; step ③ uses $\|\mathrm{U}_{\mathrm{B}}\|_{\mathsf{sp}} = \|\mathrm{U}_{\mathrm{B}}^{\mathsf{T}}\|_{\mathsf{sp}} = \|\mathbf{X}\|_{\mathsf{sp}} = 1$ for any $\mathbf{X} \in \mathrm{St}(n, r)$; step ④ uses the following equalities for any $\mathbf{V} \in \mathrm{St}(k, k)$:

$$
\|\mathbf{V} - \mathbf{I}_k\|_{\mathsf{F}}^2 = \|\mathbf{V}\|_{\mathsf{F}}^2 + \|\mathbf{I}_k\|_{\mathsf{F}}^2 - 2\langle \mathbf{I}_k, \mathbf{V} \rangle = \|\mathbf{I}_k\|_{\mathsf{F}}^2 + \|\mathbf{I}_k\|_{\mathsf{F}}^2 - 2\langle \mathbf{I}_k, \mathbf{V} \rangle = 2\langle \mathbf{I}_k, \mathbf{I}_k - \mathbf{V} \rangle.
$$

$\square$

## D.3  PROOF OF LEMMA 2.3

*Proof.* We define $\mathcal{K}(\mathbf{V}; \mathbf{X}^t, \mathrm{B}) \triangleq \frac{1}{2}\|\mathbf{V} - \mathbf{I}_k\|_{\mathbf{Q}+\alpha\mathbf{I}}^2 + h(\mathbf{V}\mathbf{Z}) + \langle \mathbf{V}, [\nabla f(\mathbf{X}^t)(\mathbf{X}^t)^{\mathsf{T}}]_{\mathrm{BB}} \rangle + \ddot{c}$, where $\mathbf{Z} \triangleq \mathrm{U}_{\mathrm{B}}^{\mathsf{T}}\mathbf{X}^t$, and $\ddot{c} = h(\mathrm{U}_{\mathrm{B}^c}^{\mathsf{T}}\mathbf{X}^t) + f(\mathbf{X}^t) - \langle \mathbf{I}_k, [\nabla f(\mathbf{X}^t)(\mathbf{X}^t)^{\mathsf{T}}]_{\mathrm{BB}} \rangle$ is a constant.

**Part (a)**. Using the definition of $\mathcal{K}(\mathbf{V}; \mathbf{X}^t, \mathrm{B})$, we have the following equalities for all $\mathbf{V} \in \mathrm{St}(k, k)$:

$$
\begin{aligned}
&\mathcal{K}(\mathbf{V}; \mathbf{X}^t, \mathrm{B}) \\
\triangleq\ & \ddot{c} + \tfrac{1}{2}\|\mathbf{V} - \mathbf{I}_k\|_{\mathbf{Q}+\alpha\mathbf{I}_k}^2 + \langle \mathbf{V}, [\nabla f(\mathbf{X}^t)(\mathbf{X}^t)^{\mathsf{T}}]_{\mathrm{BB}} \rangle + h(\mathbf{V}\mathbf{Z}) \\
=\ & \ddot{c} + \tfrac{1}{2}\|\mathbf{V} - \mathbf{I}_k\|_{\mathbf{Q}}^2 + \tfrac{\alpha}{2}\|\mathbf{V} - \mathbf{I}_k\|_{\mathsf{F}}^2 + \langle \mathbf{V}, [\nabla f(\mathbf{X}^t)(\mathbf{X}^t)^{\mathsf{T}}]_{\mathrm{BB}} \rangle + h(\mathbf{V}\mathbf{Z}) \\
\overset{①}{=}\ & \ddot{c} + \tfrac{1}{2}\|\mathbf{V}\|_{\mathbf{Q}}^2 - \langle \mathbf{V}, \mathrm{mat}(\mathbf{Q}\mathrm{vec}(\mathbf{I}_k)) \rangle + \tfrac{1}{2}\|\mathbf{I}_k\|_{\mathbf{Q}}^2 + \alpha\langle \mathbf{I}_k, \mathbf{I}_k - \mathbf{V} \rangle + \langle \mathbf{V}, [\nabla f(\mathbf{X}^t)(\mathbf{X}^t)^{\mathsf{T}}]_{\mathrm{BB}} \rangle + h(\mathbf{V}\mathbf{Z}) \\
\overset{②}{=}\ & \ddot{c} + \tfrac{1}{2}\|\mathbf{V}\|_{\mathbf{Q}}^2 + \langle \mathbf{V}, \underbrace{[\nabla f(\mathbf{X}^t)(\mathbf{X}^t)^{\mathsf{T}}]_{\mathrm{BB}} - \mathrm{mat}(\mathbf{Q}\mathrm{vec}(\mathbf{I}_k)) - \alpha\mathbf{I}_k}_{\triangleq \mathbf{P}} \rangle + h(\mathbf{V}\mathbf{Z}) + \tfrac{1}{2}\|\mathbf{I}_k\|_{\mathbf{Q}}^2,
\end{aligned}
$$

where step ① uses Lemma 2.2(c) that: $\frac{1}{2}\|\mathbf{V} - \mathbf{I}_k\|_{\mathsf{F}}^2 = \langle \mathbf{I}_k, \mathbf{I}_k - \mathbf{V} \rangle$; step ② uses the definition of $\mathbf{P}$.

**Part (b)**. We consider the case that $\mathbf{Q}$ is chosen to be a diagonal matrix that $\mathbf{Q} = \varsigma\mathbf{I}_k$, where $\varsigma$ is defined in Equation (9). Using $\mathbf{V} \in \mathrm{St}(k, k)$, the term $\frac{1}{2}\|\mathbf{V}\|_{\mathbf{Q}}^2$ simplifies to a constant with $\frac{1}{2}\|\mathbf{V}\|_{\mathbf{Q}}^2 = \frac{1}{2}\varsigma k$. We can deduce from (3):

$$
\bar{\mathbf{V}}^t \in \arg\min_{\mathbf{V}\in\mathrm{St}(k,k)} \mathcal{P}(\mathbf{V}) \triangleq \langle \mathbf{V}, \mathbf{P} \rangle + h(\mathbf{X}). \tag{34}
$$

In particular, when $h(\mathbf{X}) = 0$, Problem (34) becomes the nearest orthogonality matrix problem and can be solved analytically, yielding a closed-form solution that:

$$
\bar{\mathbf{V}}^t \in \arg\min_{\mathbf{V}\in\mathrm{St}(k,k)} \tfrac{1}{2}\|\mathbf{V} - (-\mathbf{P})\|_{\mathsf{F}}^2 = \mathbb{P}_{\mathcal{M}}(-\mathbf{P}) = -\mathbb{P}_{\mathcal{M}}(\mathbf{P}) = -\tilde{\mathbf{U}}\tilde{\mathbf{V}}^{\mathsf{T}}.
$$

Here, $\mathbf{P} = \tilde{\mathbf{U}}\mathrm{Diag}(\mathbf{s})\tilde{\mathbf{V}}^{\mathsf{T}}$ is the singular value decomposition of $\mathbf{P}$ with $\tilde{\mathbf{U}}, \tilde{\mathbf{V}} \in \mathrm{St}(k, k)$, $\mathbf{s} \in \mathbb{R}^k$, and $\mathbf{s} \geq \mathbf{0}$.

Notably, the multiplier for the orthogonality constraint $\mathbf{V}^{\mathsf{T}}\mathbf{V} = \mathbf{I}_k$ can be computed as: $\mathbf{\Lambda} = -\mathbf{P}^{\mathsf{T}}\bar{\mathbf{V}}^t \overset{①}{=} -[\tilde{\mathbf{U}}\mathrm{Diag}(\mathbf{s})\tilde{\mathbf{V}}^{\mathsf{T}}]^{\mathsf{T}} \cdot [-\tilde{\mathbf{U}}\tilde{\mathbf{V}}^{\mathsf{T}}] = \tilde{\mathbf{V}}\mathrm{Diag}(\mathbf{s})\tilde{\mathbf{U}}^{\mathsf{T}}\tilde{\mathbf{U}}\tilde{\mathbf{V}}^{\mathsf{T}} \overset{②}{=} \tilde{\mathbf{V}}\mathrm{Diag}(\mathbf{s})\tilde{\mathbf{V}}^{\mathsf{T}} \overset{③}{\succeq} \mathbf{0}$, where step ① uses $\mathbf{P} = \tilde{\mathbf{U}}\mathrm{Diag}(\mathbf{s})\tilde{\mathbf{V}}^{\mathsf{T}}$ and $\bar{\mathbf{V}}^t = -\tilde{\mathbf{U}}\tilde{\mathbf{V}}^{\mathsf{T}}$; step ② uses $\tilde{\mathbf{U}}^{\mathsf{T}}\tilde{\mathbf{U}} = \mathbf{I}_k$; step ③ uses $\mathbf{s} \geq 0$.

$\square$

## D.4  PROOF OF LEMMA 2.5

*Proof.* Any $2 \times 2$ matrix takes the form $\mathbf{V} = \left(\begin{smallmatrix} a & b \\ c & d \end{smallmatrix}\right)$. The orthogonality constraint implies that $\mathbf{V} \in \mathrm{St}(2, 2)$ meets the following three equations: $1 = a^2 + b^2$, $1 = c^2 + d^2$, $0 = ac + bd$.

Without loss of generality, we let $c = \sin(\theta)$ and $d = \cos(\theta)$ with $\theta \in \mathbb{R}$. Then we obtain either *(i)* $a = \cos(\theta), b = -\sin(\theta)$ or *(ii)* $a = -\cos(\theta), b = \sin(\theta)$. Therefore, we have the following Givens rotation matrix $\mathbf{V}_\theta^{\mathrm{rot}}$ and Jacobi reflection matrix $\mathbf{V}_\theta^{\mathrm{ref}}$:

$$\mathbf{V}_\theta^{\mathrm{rot}} \triangleq \begin{bmatrix} \cos(\theta) & -\sin(\theta) \\ \sin(\theta) & \cos(\theta) \end{bmatrix}, \ \mathbf{V}_\theta^{\mathrm{ref}} \triangleq \begin{bmatrix} -\cos(\theta) & \sin(\theta) \\ \sin(\theta) & \cos(\theta) \end{bmatrix}.$$

Note that for any $a, b, c, d \in \mathbb{R}$, we have: $\det(\begin{smallmatrix} a & b \\ c & d \end{smallmatrix}) = ad - bc$. Therefore, we obtain: $\det(\mathbf{V}_\theta^{\mathrm{rot}}) = \cos^2(\theta) + \sin^2(\theta) = 1$ and $\det(\mathbf{V}_\theta^{\mathrm{rot}}) = -\cos^2(\theta) - \sin^2(\theta) = -1$ for any $\theta \in \mathbb{R}$.

$\square$

# E  PROOF FOR SECTION 3

```
function [Q,R] = JacobiGivensQR(X)                                    1
n = size(X,1); Q = eye(n); R = X;                                     2
for j=1:n                                                             3
    for i=n:-1:(j+1)                                                  4
        B = [i-1;i]; V = Givens(R(i-1,j),R(i,j));                     5
        R(B,:) = V'*R(B,:); Q(:,B) = Q(:,B)*V;                        6
        if (i==j+1 && R(j,j)<0)                                       7
            V = [-1 0; 0 -1]; % or V = [-1 0; 0 1];                   8
            R(B,:) = V'*R(B,:); Q(:,B) = Q(:,B)*V;                    9
        end                                                          10
    end                                                              11
end                                                                  12
if(R(n,n)<0)                                                         13
    V = [1 0;0 -1]; R(B,:) = V'*R(B,:); Q(:,B) = Q(:,B)*V;           14
end                                                                  15
                                                                     16
function V = Givens(a,b)                                             17
% Find a Givens rotation that V'*[a;b] = [r;0]                       18
if (b==0)                                                            19
    c = 1; s = 0;                                                    20
else                                                                 21
    if (abs(b) > abs(a))                                             22
        tau = -a/b; s = 1/sqrt(1+tau^2); c = s*tau;                  23
    else                                                             24
        tau = -b/a; c = 1/sqrt(1+tau^2); s = c*tau;                  25
    end                                                              26
end                                                                  27
V = [c s;-s c];                                                      28
```

Listing 1: Matlab implementation for our **Jacobi-Givens-QR** algorithm.

## E.1  PROOF OF THEOREM 3.1

*Proof.* **Part (a)**. First, recall the classical **Givens-QR** algorithm, which is detailed in Section 5.2.5 of (Golub & Van Loan, 2013)). This algorithm can decompose any matrix $\mathbf{X} \in \mathbb{R}^{n \times n}$ (not necessarily orthogonal) into the form $\mathbf{X} = \mathbf{QR}$, where $\mathbf{Q}$ is an orthogonal matrix ($\mathbf{Q} \in \mathrm{St}(n, n)$) and $\mathbf{R}$ is a lower triangular matrix with $\mathbf{R}_{ij} = 0$ for all $i < j$, achieved through $\mathrm{C}_n^2 = \frac{n(n-1)}{2}$ Givens rotation steps.

Combining the result from Lemma A.5, we can conclude that classical **Givens-QR** algorithm can decompose any orthogonal matrix into the form $\mathbf{X} = \mathbf{QR}$, where $\mathbf{Q} \in \mathrm{St}(n, n)$ and $\mathbf{R}$ is diagonal matrix with $\mathbf{R}_{i,i} \in \{-1, +1\}$ for all $i \in [n]$.

We introduce a modification to the **Givens-QR** algorithm, resulting in our **Jacobi-Givens-QR** algorithm as presented in Listing 1. This algorithm can decompose any matrix $\mathbf{X} \in \mathrm{St}(n, n)$ into the form $\mathbf{X} = \mathbf{QR}$, where $\mathbf{Q} = \mathbf{X}$ and $\mathbf{R} = \mathbf{I}_n$, using a sequence of $\mathrm{C}_n^k$ Givens rotation or Jacobi reflection steps.

Please take note of the following four important points in Listing 1.

a) When we remove Lines 7-10 and Lines 13-15 from Listing 1, it essentially reverts to the classical **Givens-QR** algorithm. **Givens-QR** operates by selecting an appropriate Givens rotation matrix $\mathbf{V} = \left[ \begin{smallmatrix} \cos(\theta) & \sin(\theta) \\ -\sin(\theta) & \cos(\theta) \end{smallmatrix} \right]$ with a suitable rotation angle $\theta$ to zero-out the matrix element $\mathbf{R}_{ij}$ systematically from left to right ($j = 1 \to n$) and bottom to top ($i = n \to (j+1)$) within every pair of neighboring columns.

b) Lines 7-10 and Lines 13-15 can be viewed as correction steps to ensure that the entries $\mathbf{R}_{j,j} = 1$ for all $j = n$.

c) Line 7-10 is executed for $(n-2)$ times. In Line 7-10, when **Jacobi-Givens-QR** detects a negative entry $\mathbf{R}_{i-1,i-1}$ with $i = j + 1$, it applies a rotation matrix $\mathbf{V} \triangleq \left( \begin{smallmatrix} -1 & 0 \\ 0 & -1 \end{smallmatrix} \right)$ to the two rows $\mathtt{B} = [i-1, i]$ to ensure that[4] $\mathbf{R}_{i-1,i-1} = 1$.

d) Line 13-15 is executed only once when $\det(\mathbf{X}) = -1$. In such cases, we have $\mathbf{R}_{\mathtt{BB}} = \left( \begin{smallmatrix} 1 & 0 \\ 0 & -1 \end{smallmatrix} \right)$ and $\det(\mathbf{R}_{\mathtt{BB}}) = -1$, where $\mathtt{B} = [n-1, n]$ is the two indices for the final rotation or reflection step. To ensure that the resulting $\mathbf{R}_{\mathtt{BB}}$ is an identify matrix, **Jacobi-Givens-QR** employs a reflection matrix $\mathbf{V} = \left( \begin{smallmatrix} 1 & 0 \\ 0 & -1 \end{smallmatrix} \right)$, leading to $\mathbf{V}^\mathsf{T}\mathbf{R}_{\mathtt{BB}} = \mathbf{I}_2$.

Therefore, we establish the conclusion that any orthogonal matrix $\mathbf{X} \in \mathrm{St}(n,n)$ can be expressed as $\mathbf{D} = \mathcal{W}_{\mathrm{C}_n^k}...\mathcal{W}_2\mathcal{W}_1$, where $\mathcal{W}_i = \mathbf{U}_{\mathcal{B}_i}\mathcal{V}_i\mathbf{U}_{\mathcal{B}_i}^\mathsf{T} + \mathbf{U}_{\mathcal{B}_i^c}\mathbf{U}_{\mathcal{B}_i^c}^\mathsf{T}$, and $\mathcal{V}_i \in \mathrm{St}(2,2)$ is a suitable matrix associated with $\mathcal{B}_i$. Furthermore, if $\forall i, \mathcal{V}_i = \mathbf{I}_2$, we have $\forall i, \mathcal{W}_i = \mathbf{I}_n$, leading to $\mathbf{D} = \mathbf{I}_n$. This concludes the proof of the first part of this theorem.

**Part (b)**. For any given $\mathbf{X} \in \mathrm{St}(n,r)$ and $\mathbf{X}^0 \in \mathrm{St}(n,r)$, we let:

$$\bar{\mathbf{D}} = \mathbb{P}_{\mathrm{St}(n,n)}(\mathbf{X}[\mathbf{X}^0]^\mathsf{T}), \tag{35}$$

where $\mathbb{P}_{\mathrm{St}(n,n)}(\mathbf{Y})$ denotes the nearest orthogonality matrix to the given matrix $\mathbf{Y}$.

Assume that the matrix $\mathbf{X}[\mathbf{X}^0]^\mathsf{T}$ has the following singular value decomposition:

$$\mathbf{X}(\mathbf{X}^0)^\mathsf{T} = \mathbf{U}\mathrm{Diag}(\mathbf{z})\mathbf{V}^\mathsf{T}, \ \mathbf{z} \in \{0,1\}^n, \ \mathbf{U} \in \mathrm{St}(n,n), \ \mathbf{V} \in \mathrm{St}(n,n).$$

Therefore, we have the following equalities:

$$\mathrm{Diag}(\mathbf{z}) = \mathbf{U}^\mathsf{T}\mathbf{X}[\mathbf{X}^0]^\mathsf{T}\mathbf{V}. \tag{36}$$
$$\bar{\mathbf{D}} = \mathbf{U}\mathbf{V}^\mathsf{T} \in \mathrm{St}(n,n). \tag{37}$$

Furthermore, we derive the following results:

$$\begin{aligned}
& \mathbf{z} \in \{0,1\}^n \\
\Rightarrow\ & \mathrm{Diag}(\mathbf{z})^\mathsf{T} = \mathrm{Diag}(\mathbf{z})\mathrm{Diag}(\mathbf{z})^\mathsf{T} \\
\Rightarrow\ & \mathbf{U}[\mathrm{Diag}(\mathbf{z})^\mathsf{T} - \mathrm{Diag}(\mathbf{z})\mathrm{Diag}(\mathbf{z})^\mathsf{T}]\mathbf{U}^\mathsf{T}\mathbf{X} = \mathbf{0} \\
\overset{①}{\Rightarrow}\ & \mathbf{U}[\mathbf{V}^\mathsf{T}\mathbf{X}^0\mathbf{X}^\mathsf{T}\mathbf{U} - \mathbf{U}^\mathsf{T}\mathbf{X}(\mathbf{X}^0)^\mathsf{T}\mathbf{V}\mathbf{V}^\mathsf{T}\mathbf{X}^0\mathbf{X}^\mathsf{T}\mathbf{U}]\mathbf{U}^\mathsf{T}\mathbf{X} = \mathbf{0} \\
\Rightarrow\ & \mathbf{U}\mathbf{V}^\mathsf{T}\mathbf{X}^0\mathbf{X}^\mathsf{T}\mathbf{U}\mathbf{U}^\mathsf{T}\mathbf{X} - \mathbf{U}\mathbf{U}^\mathsf{T}\mathbf{X}(\mathbf{X}^0)^\mathsf{T}\mathbf{V}\mathbf{V}^\mathsf{T}\mathbf{X}^0\mathbf{X}^\mathsf{T}\mathbf{U}\mathbf{U}^\mathsf{T}\mathbf{X} = \mathbf{0} \\
\overset{②}{\Rightarrow}\ & \mathbf{U}\mathbf{V}^\mathsf{T}\mathbf{X}^0 - \mathbf{X} = \mathbf{0} \\
\overset{③}{\Rightarrow}\ & \bar{\mathbf{D}} \cdot \mathbf{X}^0 - \mathbf{X} = \mathbf{0},
\end{aligned}$$

where step ① uses (36); step ② uses $\mathbf{U}\mathbf{U}^\mathsf{T} = \mathbf{I}_n$, $\mathbf{V}\mathbf{V}^\mathsf{T} = \mathbf{I}_n$, $\mathbf{X}^\mathsf{T}\mathbf{X} = \mathbf{I}_r$, and $[\mathbf{X}^0]^\mathsf{T}\mathbf{X}^0 = \mathbf{I}_r$; step ③ uses (37). We conclude that, for any given $\mathbf{X} \in \mathrm{St}(n,r)$ and $\mathbf{X}^0 \in \mathrm{St}(n,r)$, we can always find a matrix $\bar{\mathbf{D}} \in \mathrm{St}(n,n)$ such that $\bar{\mathbf{D}}\mathbf{X}^0 = \mathbf{X}$.

Since the matrix $\bar{\mathbf{D}} \in \mathrm{St}(n,n)$ can be represented as $\bar{\mathbf{D}} = \mathcal{W}_{\mathrm{C}_n^k}...\mathcal{W}_2\mathcal{W}_1$, where $\mathcal{W}_i = \mathbf{U}_{\mathcal{B}_i}\mathcal{V}_i\mathbf{U}_{\mathcal{B}_i}^\mathsf{T} + \mathbf{U}_{\mathcal{B}_i^c}\mathbf{U}_{\mathcal{B}_i^c}^\mathsf{T}$ for some suitable $\mathcal{V}_i \in \mathrm{St}(2,2)$ (as established in the first part of this theorem), we can conclude that any matrix $\mathbf{X} \in \mathrm{St}(n,r)$ can be expressed as $\mathbf{X} = \bar{\mathbf{D}}\mathbf{X}^0 = \mathcal{W}_{\mathrm{C}_n^k}...\mathcal{W}_2\mathcal{W}_1\mathbf{X}^0$.

$\square$

---

[4]Alternatively, one can use the reflection matrix $\mathbf{V} \triangleq \left( \begin{smallmatrix} -1 & 0 \\ 0 & 1 \end{smallmatrix} \right)$ instead of the rotation matrix $\mathbf{V} \triangleq \left( \begin{smallmatrix} -1 & 0 \\ 0 & -1 \end{smallmatrix} \right)$ to ensure that $\mathbf{R}_{i-1,i-1} = 1$.

### E.2 Proof of Corollary 3.2

*Proof.* We denote $e_i$ as the $i$-th canonical basis vector in $\mathbb{R}^n$.

We denote the set $\{\mathcal{B}_1, \mathcal{B}_2, ..., \mathcal{B}_{\mathrm{C}_n^k}\}$ as all possible combinations of the index vectors choosing $k$ items from $n$ without repetition.

**Part (a)**. Fix any $k \geq 2$. By Theorem 3.1(a) for the case $k = 2$, for every $\mathbf{D} \in \mathrm{St}(n,n)$ there exist index pairs $(p_i, q_i)$ and matrices $\mathcal{V}_i^{(2)} \in \mathrm{St}(2,2)$ such that

$$\mathbf{D} = \mathcal{W}_{\mathrm{C}_n^2}^{(2)} \cdots \mathcal{W}_2^{(2)} \mathcal{W}_1^{(2)},$$

where

$$\mathcal{W}_i^{(2)} = \mathbf{I}_n + \mathbf{U}_{\mathcal{B}_i}^{(2)} (\mathcal{V}_j^{(2)} - \mathbf{I}_2) [\mathbf{U}_{\mathcal{B}_i}^{(2)}]^\top, \quad \mathbf{U}_{\mathcal{B}_i}^{(2)} = [e_{p_i}, e_{q_i}] \in \mathbb{R}^{n \times 2}.$$

We let

$$\mathcal{V}_i \triangleq \begin{pmatrix} \mathcal{V}_i^{(2)} & \mathbf{0} \\ \mathbf{0} & \mathbf{I}_{k-2} \end{pmatrix} \in \mathrm{St}(k,k), \qquad \mathcal{W}_i \triangleq \mathbf{I}_n + \mathbf{U}_{\mathcal{B}_i} (\mathcal{V}_i - \mathbf{I}_k) \mathbf{U}_{\mathcal{B}_i}^\top.$$

By construction, $\mathcal{W}_j$ acts as $\mathcal{V}_j^{(2)}$ on the two coordinates $p_j, q_j$ and as the identity on all other coordinates, hence $\mathcal{W}_j = \mathcal{W}_j^{(2)}$ as linear operators on $\mathbb{R}^n$. Therefore

$$\mathbf{D} = \mathcal{W}_{\mathrm{C}_n^2}^{(2)} \cdots \mathcal{W}_1^{(2)} = \mathcal{W}_{\mathrm{C}_n^2} \cdots \mathcal{W}_1,$$

which proves the first part of this corollary for any $k \geq 2$.

**Part (b)**. A similar argument to that used in the proof of Theorem 3.1(b) yields the second part of this corollary.

$\square$

### E.3 Proof for Theorem 3.8

*Proof.* We use $\bar{\mathbf{X}}$, $\ddot{\mathbf{X}}$, and $\check{\mathbf{X}}$ to denote a *global optimal point*, a $\mathrm{BS}_k$-*point*, and a *critical point* of Problem (1), respectively.

Setting the Riemannian subgradient of $\mathcal{K}(\mathbf{V}; \ddot{\mathbf{X}}, \mathtt{B})$ *w.r.t.* $\mathbf{V}$ to zero, we have $\mathbf{0} \in \partial_{\mathcal{M}} \mathcal{K}(\mathbf{V}; \ddot{\mathbf{X}}, \mathtt{B}) = \ddot{\mathbf{G}}(\mathbf{V}) \ominus \mathbf{V}[\ddot{\mathbf{G}}(\mathbf{V})]^\top \mathbf{V}$, where $\ddot{\mathbf{G}}(\mathbf{V}) = \alpha(\mathbf{V} - \mathbf{I}_k) + \mathbf{U}_{\mathtt{B}}^\top [\mathrm{mat}(\mathbf{H} \mathrm{vec}(\mathbf{X}^+ - \ddot{\mathbf{X}})) + \nabla f(\ddot{\mathbf{X}}) + \partial h(\mathbf{X}^+)] \ddot{\mathbf{X}}^\top \mathbf{U}_{\mathtt{B}}$ and $\mathbf{X}^+ = \ddot{\mathbf{X}} + \mathbf{U}_{\mathtt{B}}(\mathbf{V} - \mathbf{I}_k) \mathbf{U}_{\mathtt{B}}^\top \ddot{\mathbf{X}}$. Letting $\mathbf{V} = \mathbf{I}_k$, we have the following **necessary but not sufficient** condition for any $\mathrm{BS}_k$-*point*:

$$\forall \mathtt{B} \in \{\mathcal{B}_i\}_{i=1}^{\mathrm{C}_n^k}, \ \mathbf{0} = \mathbf{U}_{\mathtt{B}}^\top (\mathbf{G} \ddot{\mathbf{X}}^\top - \ddot{\mathbf{X}} \mathbf{G}^\top) \mathbf{U}_{\mathtt{B}}, \text{ with } \mathbf{G} \in \nabla f(\ddot{\mathbf{X}}) + \partial h(\ddot{\mathbf{X}}). \tag{38}$$

**Part (a)**. We now show that $\{\text{critical points } \check{\mathbf{X}}\} \supseteq \{\mathrm{BS}_k\text{-points } \ddot{\mathbf{X}}\}$ for all $k \geq 2$. We let $\mathbf{G} \in \nabla f(\ddot{\mathbf{X}}) + \partial h(\ddot{\mathbf{X}})$. Using Lemma A.1, we have:

$$
\begin{aligned}
\mathbf{0}_{n,n} = \mathbf{G} \ddot{\mathbf{X}}^\top - \ddot{\mathbf{X}} \mathbf{G}^\top \ \Rightarrow \ & (\mathbf{0}_{n,n} \cdot \ddot{\mathbf{X}}) = (\mathbf{G} \ddot{\mathbf{X}}^\top - \ddot{\mathbf{X}} \mathbf{G}^\top) \ddot{\mathbf{X}} \\
\overset{①}{\Rightarrow} \ & \mathbf{0}_{n,r} = \mathbf{G} - \ddot{\mathbf{X}} \mathbf{G}^\top \ddot{\mathbf{X}}, \\
\Rightarrow \ & \ddot{\mathbf{X}}^\top \cdot \mathbf{0}_{n,r} = \ddot{\mathbf{X}}^\top (\mathbf{G} - \ddot{\mathbf{X}} \mathbf{G}^\top \ddot{\mathbf{X}}) \\
\overset{②}{\Rightarrow} \ & \mathbf{0}_{r,r} = \ddot{\mathbf{X}}^\top \mathbf{G} - \mathbf{G}^\top \ddot{\mathbf{X}} \\
\Rightarrow \ & \mathbf{0}_{n,n} = \ddot{\mathbf{X}} (\ddot{\mathbf{X}}^\top \mathbf{G} - \mathbf{G}^\top \ddot{\mathbf{X}}) \ddot{\mathbf{X}}^\top \\
\overset{③}{\Rightarrow} \ & \mathbf{0}_{n,n} = \ddot{\mathbf{X}} \underbrace{\ddot{\mathbf{X}}^\top \mathbf{G} \ddot{\mathbf{X}}^\top}_{\triangleq \mathbf{G}^\top} - \underbrace{\ddot{\mathbf{X}} \mathbf{G}^\top \ddot{\mathbf{X}}}_{\triangleq \mathbf{G}} \ddot{\mathbf{X}}^\top,
\end{aligned}
\tag{39}
$$

where steps ① and ② use $\ddot{\mathbf{X}}^\top \ddot{\mathbf{X}} = \mathbf{I}_r$; step ③ uses Equality (39) that $\mathbf{G} = \ddot{\mathbf{X}} \mathbf{G}^\top \ddot{\mathbf{X}}$. We conclude that the necessary condition in Equation (38) is equivalent to the optimality condition of critical points.

**Part (b)**. We now show that $\{\text{BS}_k\text{-points}\,\ddot{\mathbf{X}}\} \supseteq \{\text{global optimal points}\,\bar{\mathbf{X}}\}$ for all $k \in \{2, 3, \ldots, n\}$. We define $\mathcal{X}_{\text{B}}^{\star}(\mathbf{V}) \triangleq \bar{\mathbf{X}} + \mathrm{U}_{\text{B}}(\mathbf{V} - \mathbf{I}_k)\mathrm{U}_{\text{B}}^{\mathsf{T}}\bar{\mathbf{X}}$, and $\mathcal{K}(\mathbf{V}; \mathbf{X}, \text{B}) \triangleq f(\mathbf{X}) + \langle \mathbf{V} - \mathbf{I}_k, [\nabla f(\mathbf{X})(\mathbf{X})^{\mathsf{T}}]_{\text{BB}} \rangle + \frac{1}{2}\|\mathbf{V} - \mathbf{I}_k\|_{\mathbf{Q}+\alpha\mathbf{I}_k}^2 + h(\mathrm{U}_{\text{B}^c}^{\mathsf{T}}\mathbf{X}) + h(\mathbf{V}\mathrm{U}_{\text{B}}^{\mathsf{T}}\mathbf{X})$. We let $\mathbf{V}_{(i)} \in \text{St}(k,k)$ and $\mathcal{B}_i \in \{\mathcal{B}_i\}_{i=1}^{C_n^k}$. We derive:

$$\mathcal{K}(\mathbf{I}_k; \bar{\mathbf{X}}, \mathcal{B}_i),\ \forall \mathcal{B}_i$$

$$\overset{\text{①}}{=} F(\bar{\mathbf{X}}) = h(\bar{\mathbf{X}}) + f(\bar{\mathbf{X}})$$

$$\overset{\text{②}}{\leq} h(\mathbf{X}) + f(\mathbf{X}), \forall \mathbf{X} \in \text{St}(n,r)$$

$$\overset{\text{③}}{\leq} h(\bar{\mathbf{X}} + \mathrm{U}_{\mathcal{B}_i}(\mathbf{V}_{(i)} - \mathbf{I}_k)\mathrm{U}_{\mathcal{B}_i}^{\mathsf{T}}\bar{\mathbf{X}}) + f(\bar{\mathbf{X}} + \mathrm{U}_{\mathcal{B}_i}(\mathbf{V}_{(i)} - \mathbf{I}_k)\mathrm{U}_{\mathcal{B}_i}^{\mathsf{T}}\bar{\mathbf{X}}),\ \forall \mathbf{V}_{(i)},\ \forall \mathcal{B}_i$$

$$\overset{\text{④}}{=} h(\mathcal{X}_{\mathcal{B}_i}^{\star}(\mathbf{V}_{(i)})) + f(\mathcal{X}_{\mathcal{B}_i}^{\star}(\mathbf{V}_{(i)})),\ \forall \mathbf{V}_{(i)},\ \forall \mathcal{B}_i$$

$$\overset{\text{⑤}}{=} \mathcal{K}(\mathbf{V}_{(i)}; \bar{\mathbf{X}}, \mathcal{B}_i),\ \forall \mathbf{V}_{(i)},\ \forall \mathcal{B}_i$$

$$\leq \min_{\mathbf{V} \in \text{St}(k,k)} \mathcal{K}(\mathbf{V}; \bar{\mathbf{X}}, \mathcal{B}_i),\ \forall \mathcal{B}_i, \tag{40}$$

where step ① uses the definition of $\mathcal{K}(\mathbf{V}; \mathbf{X}, \text{B}) \triangleq f(\mathbf{X}) + \langle \mathbf{V} - \mathbf{I}_k, [\nabla f(\mathbf{X})(\mathbf{X})^{\mathsf{T}}]_{\text{BB}} \rangle + \frac{1}{2}\|\mathbf{V} - \mathbf{I}_k\|_{\mathbf{Q}+\alpha\mathbf{I}_k}^2 + h(\mathrm{U}_{\text{B}^c}^{\mathsf{T}}\mathbf{X}) + h(\mathbf{V}\mathrm{U}_{\text{B}}^{\mathsf{T}}\mathbf{X})$; step ② uses the definition of $\bar{\mathbf{X}}$; step ③ uses the basis representation of orthogonal matrices for all $k \geq 2$, as shown in Corollary 3.2; step ④ uses the definition of $\mathcal{X}_{\text{B}}^{\star}(\mathbf{V})$; step ⑤ uses the same strategy as in deriving Inequality (10). This leads to:

$$\mathbf{I}_k \in \arg\min_{\mathbf{V} \in \text{St}(k,k)} \mathcal{K}(\mathbf{V}; \bar{\mathbf{X}}, \mathcal{B}_i),\ \forall \mathcal{B}_i.$$

The inclusion above implies that $\{\text{BS}_k\text{-points}\,\ddot{\mathbf{X}}\} \supseteq \{\text{global optimal points}\,\bar{\mathbf{X}}\}$.

**Part (c)**. We now show that $\{\text{BS}_k\text{-}points\,\ddot{\mathbf{X}}\} \supseteq \{\text{BS}_{k+1}\text{-}points\,\ddot{\mathbf{X}}\}$. It is evident that the subproblem of finding $\text{BS}_k\text{-}points$ is encompassed within that of finding $\text{BS}_{k+1}\text{-}points$ stationary point. Thus, we conclude that the optimality of the latter is stronger.

**Part (d)**. The inclusion $\{\text{critical points}\,\check{\mathbf{X}}\} \subseteq \{\text{BS}_k\text{-points}\,\ddot{\mathbf{X}}\}$ may not always hold true. This can be illustrated through simple examples of $2 \times 2$ optimization problems under orthogonality constraints (see Appendix Section C.1 for more details). Lastly, it is also evident that the inclusions $\{\text{BS}_2\text{-points}\,\ddot{\mathbf{X}}\} \subseteq \{\text{global optimal points}\,\bar{\mathbf{X}}\}$ and $\{\text{BS}_k\text{-}points\,\ddot{\mathbf{X}}\} \subseteq \{\text{BS}_{k+1}\text{-}points\,\ddot{\mathbf{X}}\}$ may not always hold true.

$\square$

# F  PROOF FOR SECTION 4

## F.1  PROOF FOR THEOREM 4.2

*Proof.* We define $\mathcal{K}(\mathbf{V}; \mathbf{X}^t, \text{B}) \triangleq \frac{1}{2}\|\mathbf{V} - \mathbf{I}_k\|_{\mathbf{Q}+\alpha\mathbf{I}_k}^2 + h(\mathbf{V}\mathbf{Z}) + \langle \mathbf{V}, [\nabla f(\mathbf{X}^t)(\mathbf{X}^t)^{\mathsf{T}}]_{\text{BB}} \rangle + \ddot{c}$, where $\mathbf{Z} \triangleq \mathrm{U}_{\text{B}}^{\mathsf{T}}\mathbf{X}^t$ and $\ddot{c} = h(\mathrm{U}_{\text{B}^c}^{\mathsf{T}}\mathbf{X}^t) + f(\mathbf{X}^t) - \langle \mathbf{I}_k, [\nabla f(\mathbf{X}^t)(\mathbf{X}^t)^{\mathsf{T}}]_{\text{BB}} \rangle$ is a constant.

We define $\tilde{c} \triangleq \frac{2}{\alpha} \cdot (F(\mathbf{X}^0) - F(\mathbf{X}^\infty))$.

**Part (a)**. First, we have the following equalities:

$$h(\mathbf{X}^{t+1}) - h(\mathbf{X}^t) \overset{\text{①}}{=} h(\mathrm{U}_{\text{B}}\bar{\mathbf{V}}^t\mathrm{U}_{\text{B}}^{\mathsf{T}}\mathbf{X}^t + \mathrm{U}_{\text{B}^c}\mathrm{U}_{\text{B}^c}^{\mathsf{T}}\mathbf{X}^t) - h(\mathrm{U}_{\text{B}}\mathrm{U}_{\text{B}}^{\mathsf{T}}\mathbf{X}^t + \mathrm{U}_{\text{B}^c}\mathrm{U}_{\text{B}^c}^{\mathsf{T}}\mathbf{X}^t)$$

$$\overset{\text{②}}{=} h(\mathrm{U}_{\text{B}}\bar{\mathbf{V}}^t\mathrm{U}_{\text{B}}^{\mathsf{T}}\mathbf{X}^t) + h(\mathrm{U}_{\text{B}^c}\mathrm{U}_{\text{B}^c}^{\mathsf{T}}\mathbf{X}^t) - h(\mathrm{U}_{\text{B}}\mathrm{U}_{\text{B}}^{\mathsf{T}}\mathbf{X}^t) - h(\mathrm{U}_{\text{B}^c}\mathrm{U}_{\text{B}^c}^{\mathsf{T}}\mathbf{X}^t)$$

$$\overset{\text{③}}{=} h(\bar{\mathbf{V}}^t\mathrm{U}_{\text{B}}^{\mathsf{T}}\mathbf{X}^t) - h(\mathrm{U}_{\text{B}}^{\mathsf{T}}\mathbf{X}^t), \tag{41}$$

where step ① uses $\mathbf{X}^{t+1} = \mathrm{U}_{\text{B}}\mathbf{V}\mathrm{U}_{\text{B}}^{\mathsf{T}}\mathbf{X}^t + \mathrm{U}_{\text{B}^c}\mathrm{U}_{\text{B}^c}^{\mathsf{T}}\mathbf{X}^t$ as in (4) and $\mathbf{I}_k = \mathrm{U}_{\text{B}}\mathrm{U}_{\text{B}}^{\mathsf{T}} + \mathrm{U}_{\text{B}^c}\mathrm{U}_{\text{B}^c}^{\mathsf{T}}$; step ② and step ③ use the coordinate-wise separable structure of $h(\cdot)$.

Second, since $\bar{\mathbf{V}}^t \in \arg\min_{\mathbf{V} \in \text{St}(k,k)} \mathcal{K}(\mathbf{V}; \mathbf{X}^t, \text{B})$, it follows that $\mathcal{K}(\bar{\mathbf{V}}^t; \mathbf{X}^t, \text{B}) \leq \mathcal{K}(\mathbf{I}_k; \mathbf{X}^t, \text{B})$. This further leads to:

$$h(\bar{\mathbf{V}}^t\mathrm{U}_{\text{B}}^{\mathsf{T}}\mathbf{X}^t) + \frac{1}{2}\|\bar{\mathbf{V}}^t - \mathbf{I}_k\|_{\mathbf{Q}+\alpha\mathbf{I}_k}^2 + \langle \bar{\mathbf{V}}^t - \mathbf{I}_k, [\nabla f(\mathbf{X}^t)(\mathbf{X}^t)^{\mathsf{T}}]_{\text{BB}} \rangle \leq h(\mathrm{U}_{\text{B}}^{\mathsf{T}}\mathbf{X}^t). \tag{42}$$

Third, we denote $\mathbf{X}^{t+1} = \mathcal{X}_{\mathrm{B}}^t(\bar{\mathbf{V}}^t)$ and derive:

$$
\begin{aligned}
f(\mathbf{X}^{t+1}) - f(\mathbf{X}^t) &\overset{①}{\leq} \langle \mathcal{X}_{\mathrm{B}}^t(\bar{\mathbf{V}}^t) - \mathbf{X}^t, \nabla f(\mathbf{X}^t) \rangle + \tfrac{1}{2}\|\mathcal{X}_{\mathrm{B}}^t(\bar{\mathbf{V}}^t) - \mathbf{X}^t\|_{\mathbf{H}}^2 \\
&\overset{②}{=} \langle \mathrm{U}_{\mathrm{B}}(\bar{\mathbf{V}}^t - \mathbf{I}_k)\mathrm{U}_{\mathrm{B}}^\mathsf{T}\mathbf{X}^t, \nabla f(\mathbf{X}^t) \rangle + \tfrac{1}{2}\|\bar{\mathbf{V}}^t - \mathbf{I}_k\|_{\underline{\mathbf{Q}}}^2 \\
&\overset{③}{\leq} \langle \bar{\mathbf{V}}^t - \mathbf{I}_k, [\nabla f(\mathbf{X}^t)(\mathbf{X}^t)^\mathsf{T}]_{\mathrm{BB}} \rangle + \tfrac{1}{2}\|\bar{\mathbf{V}}^t - \mathbf{I}_k\|_{\mathbf{Q}}^2,
\end{aligned}
\tag{43}
$$

where step ① uses Inequality (2); step ② uses Lemma 2.2(a); step ③ uses $\mathbf{Q} \succcurlyeq \underline{\mathbf{Q}}$.

Adding (41), (42), and (43) together, we obtain the following sufficient decrease condition:

$$
F(\mathbf{X}^{t+1}) - F(\mathbf{X}^t) \leq -\tfrac{\alpha}{2}\|\bar{\mathbf{V}}^t - \mathbf{I}_k\|_{\mathsf{F}}^2 \overset{①}{\leq} -\tfrac{\alpha}{2}\|\mathbf{X}^{t+1} - \mathbf{X}^t\|_{\mathsf{F}}^2,
\tag{44}
$$

where step ① uses Lemma 2.2(c).

**Part (b)**. We assume that $\mathrm{B}^t$ is selected from $\{\mathcal{B}_i\}_{i=1}^{\mathrm{C}_n^k}$ randomly and uniformly.

Taking the expectation for Inequality (44), we obtain a lower bound on the expected progress made by each iteration:

$$
\mathbb{E}_{\xi^t}[F(\mathbf{X}^{t+1})] - F(\mathbf{X}^t) \leq -\mathbb{E}_{\xi^t}[\tfrac{\alpha}{2}\|\bar{\mathbf{V}}^t - \mathbf{I}_k\|_{\mathsf{F}}^2].
$$

Telescoping the inequality above over $t = 0, 1, ..., T$, we have:

$$
\mathbb{E}_{\xi^T}[\tfrac{\alpha}{2}\textstyle\sum_{t=0}^T \|\bar{\mathbf{V}}^t - \mathbf{I}_k\|_{\mathsf{F}}^2] \leq \mathbb{E}_{\xi^T}[F(\mathbf{X}^0) - F(\mathbf{X}^{T+1})] \leq \mathbb{E}_{\xi^T}[F(\mathbf{X}^0) - F(\mathbf{X}^\infty)],
$$

where $\mathbf{X}^\infty$ denotes the limit point of Algorithm 1. As a result, there exists an index $\bar{t}$ with $0 \leq \bar{t} \leq T$ such that

$$
\mathbb{E}_{\xi^T}[\|\bar{\mathbf{V}}^{\bar{t}} - \mathbf{I}_k\|_{\mathsf{F}}^2] \leq \tfrac{2}{\alpha(T+1)}[F(\mathbf{X}^0) - F(\mathbf{X}^\infty)] = \tfrac{\tilde{c}}{T+1}.
\tag{45}
$$

Furthermore, for any $t$, $\bar{\mathbf{V}}^t$ is the optimal solution of the following minimization problem at $\mathbf{X}^t$: $\bar{\mathbf{V}}^t \in \arg\min_{\mathbf{V}} \mathcal{K}(\mathbf{V}; \mathbf{X}^t, \mathrm{B}^t)$. Since $\bar{\mathbf{V}}^t$ is a random output matrix that depends on the observed realization of the random variable $\mathrm{B}^t$, we directly obtain the following equality:

$$
\tfrac{1}{\mathrm{C}_n^k}\textstyle\sum_{i=1}^{\mathrm{C}_n^k} \mathrm{dist}(\mathbf{I}_k, \arg\min_{\mathbf{V}} \mathcal{K}(\mathbf{V}; \mathbf{X}^t, \mathcal{B}_i))^2 = \mathbb{E}_{\xi^t}[\|\bar{\mathbf{V}}^t - \mathbf{I}_k\|_{\mathsf{F}}^2].
\tag{46}
$$

Combining (45) and (46), we conclude that there exists an index $\bar{t}$ with $\bar{t} \in [0, T]$ such that the associated solution $\mathbf{X}^{\bar{t}}$ qualifies as an $\epsilon$-BS$_k$-*point* of Problem (1), provided that $T$ is sufficiently large such that $\tfrac{\tilde{c}}{T+1} \leq \epsilon$.

$\square$

### F.2 Proof of Lemma 4.4

*Proof.* We define $\mathbb{A} \ominus \mathbb{B}$ as the element-wise subtraction between sets $\mathbb{A}$ and $\mathbb{B}$.

We let $\mathbb{H}^{t+1} \in \partial h(\mathbf{X}^{t+1})$, and define:

$$
\begin{aligned}
\Omega_0 &\triangleq \mathrm{U}_{\mathrm{B}^t}^\mathsf{T}[\nabla f(\mathbf{X}^{t+1}) + \mathbb{H}^{t+1}][\mathbf{X}^{t+1}]^\mathsf{T}\mathrm{U}_{\mathrm{B}^t} \in \mathbb{R}^{k\times k}, \tag{47} \\
\Omega_1 &\triangleq \mathrm{U}_{\mathrm{B}^t}^\mathsf{T}[\nabla f(\mathbf{X}^{t+1}) + \mathbb{H}^{t+1}][\mathbf{X}^t]^\mathsf{T}\mathrm{U}_{\mathrm{B}^t} \in \mathbb{R}^{k\times k}, \tag{48} \\
\Omega_2 &\triangleq \mathrm{U}_{\mathrm{B}^t}^\mathsf{T}[\nabla f(\mathbf{X}^t) - \nabla f(\mathbf{X}^{t+1})][\mathbf{X}^t]^\mathsf{T}\mathrm{U}_{\mathrm{B}^t} \in \mathbb{R}^{k\times k}. \tag{49}
\end{aligned}
$$

**Part (a)**. First, using the optimality of $\bar{\mathbf{V}}^t$ for the subproblem, we have:

$$
\mathbf{0}_{k,k} = \tilde{\mathbf{G}} - \bar{\mathbf{V}}^t\tilde{\mathbf{G}}^\mathsf{T}\bar{\mathbf{V}}^t
$$
$$
\text{where } \tilde{\mathbf{G}} = \underbrace{\mathrm{mat}((\mathbf{Q} + \alpha\mathbf{I}_k)\mathrm{vec}(\bar{\mathbf{V}}^t - \mathbf{I}_k))}_{\triangleq \Upsilon_1} + \underbrace{\mathrm{U}_{\mathrm{B}^t}^\mathsf{T}[\nabla f(\mathbf{X}^t) + \mathbb{H}^{t+1}](\mathbf{X}^t)^\mathsf{T}\mathrm{U}_{\mathrm{B}^t}}_{\triangleq \Upsilon_2}.
$$

Using the relation that $\tilde{\mathbf{G}} = \Upsilon_1 + \Upsilon_2$, we obtain the following results from the above equality:

$$
\begin{aligned}
\mathbf{0}_{k,k} &= (\Upsilon_1 + \Upsilon_2) - \bar{\mathbf{V}}^t(\Upsilon_1 + \Upsilon_2)^\mathsf{T}\bar{\mathbf{V}}^t \\
&\overset{①}{\Rightarrow} \mathbf{0}_{k,k} = \Upsilon_1 + \Omega_1 + \Omega_2 - \bar{\mathbf{V}}^t(\Upsilon_1 + \Omega_1 + \Omega_2)^\mathsf{T}\bar{\mathbf{V}}^t \\
&\Rightarrow \Omega_1 = \bar{\mathbf{V}}^t(\Upsilon_1 + \Omega_1 + \Omega_2)^\mathsf{T}\bar{\mathbf{V}}^t - \Upsilon_1 - \Omega_2,
\end{aligned}
\tag{50}
$$

where step ① uses $\Upsilon_2 = \Omega_1 + \Omega_2$.

Second, since both $\mathrm{B}^t$ and $\mathrm{B}^{t+1}$ are randomly and dependently selected from $\{\mathcal{B}_i\}_{i=1}^{\mathrm{C}_n^k}$ *with replacement*, each with an equal probability of $\frac{1}{\mathrm{C}_n^k}$, for any $\tilde{\mathbf{A}} \in \mathbb{R}^{n \times n}$, we have:

$$\mathbb{E}_{\mathrm{B}^{t+1}}[\|\mathrm{U}_{\mathrm{B}^{t+1}}^\mathsf{T}\tilde{\mathbf{A}}\mathrm{U}_{\mathrm{B}^{t+1}}\|_\mathsf{F}^2] = \tfrac{1}{\mathrm{C}_n^k}\sum_{i=1}^{\mathrm{C}_n^k}\|\mathbf{U}_{\mathcal{B}_i}^\mathsf{T}\tilde{\mathbf{A}}\mathbf{U}_{\mathcal{B}_i}\|_\mathsf{F}^2 = \mathbb{E}_{\mathrm{B}^t}\|\mathrm{U}_{\mathrm{B}^t}^\mathsf{T}\tilde{\mathbf{A}}\mathrm{U}_{\mathrm{B}^t}\|_\mathsf{F}^2.$$

Using the definition $\xi^t \triangleq (\mathrm{B}^1, \mathrm{B}^2, \ldots, \mathrm{B}^t)$, we have:

$$\mathbb{E}_{\xi^{t+1}}[\|\mathrm{U}_{\mathrm{B}^{t+1}}^\mathsf{T}\tilde{\mathbf{A}}\mathrm{U}_{\mathrm{B}^{t+1}}\|_\mathsf{F}^2] = \mathbb{E}_{\xi^t}\|\mathrm{U}_{\mathrm{B}^t}^\mathsf{T}\tilde{\mathbf{A}}\mathrm{U}_{\mathrm{B}^t}\|_\mathsf{F}^2. \tag{51}$$

Third, we derive the following results:

$$\mathbb{E}_{\xi^{t+1}}[\mathrm{dist}^2(\mathbf{0}, \partial_\mathcal{M}\mathcal{K}(\mathbf{I}_k; \mathbf{X}^{t+1}, \mathrm{B}^{t+1}))] = \mathbb{E}_{\xi^{t+1}}[\|\partial_\mathcal{M}\mathcal{K}(\mathbf{I}_k; \mathbf{X}^{t+1}, \mathrm{B}^{t+1})\|_\mathsf{F}^2]$$

$$\overset{①}{=} \mathbb{E}_{\xi^{t+1}}[\|\mathrm{U}_{\mathrm{B}^{t+1}}^\mathsf{T}\{\partial F(\mathbf{X}^{t+1})[\mathbf{X}^{t+1}]^\mathsf{T} \ominus \mathbf{X}^{t+1}[\partial F(\mathbf{X}^{t+1})]^\mathsf{T}\}\mathrm{U}_{\mathrm{B}^{t+1}}\|_\mathsf{F}^2]$$

$$\overset{②}{=} \mathbb{E}_{\xi^t}[\|\mathrm{U}_{\mathrm{B}^t}^\mathsf{T}\{\partial F(\mathbf{X}^{t+1})[\mathbf{X}^{t+1}]^\mathsf{T} \ominus \mathbf{X}^{t+1}[\partial F(\mathbf{X}^{t+1})]^\mathsf{T}\}\mathrm{U}_{\mathrm{B}^t}\|_\mathsf{F}^2]$$

$$\overset{③}{\leq} \mathbb{E}_{\xi^t}[\|\Omega_0 - \Omega_0^\mathsf{T}\|_\mathsf{F}^2]$$

$$\overset{④}{\leq} 8\mathbb{E}_{\xi^t}[\|\Omega_0 - \Omega_1\|_\mathsf{F}^2] + 2\mathbb{E}_{\xi^t}[\|\Omega_1 - \Omega_1^\mathsf{T}\|_\mathsf{F}^2]$$

$$\overset{⑤}{=} 8\mathbb{E}_{\xi^t}[\|\Omega_0 - \Omega_1\|_\mathsf{F}^2] + 2\mathbb{E}_{\xi^t}[\|\bar{\mathbf{V}}^t(\Upsilon_1 + \Omega_1 + \Omega_2)^\mathsf{T}\bar{\mathbf{V}}^t - \Upsilon_1 - \Omega_2 - \Omega_1^\mathsf{T}\|_\mathsf{F}^2]$$

$$\overset{⑥}{\leq} 8\mathbb{E}_{\xi^t}[\|\Omega_0 - \Omega_1\|_\mathsf{F}^2] + 6\mathbb{E}_{\xi^t}[\|\bar{\mathbf{V}}^t\Upsilon_1^\mathsf{T}\bar{\mathbf{V}}^t - \Upsilon_1\|_\mathsf{F}^2] + 6\mathbb{E}_{\xi^t}[\|\bar{\mathbf{V}}^t\Omega_1^\mathsf{T}\bar{\mathbf{V}}^t - \Omega_1^\mathsf{T}\|_\mathsf{F}^2]$$
$$+ 6\mathbb{E}_{\xi^t}[\|\bar{\mathbf{V}}^t\Omega_2^\mathsf{T}\bar{\mathbf{V}}^t - \Omega_2\|_\mathsf{F}^2], \tag{52}$$

where step ① uses the definition of $\partial_\mathcal{M}\mathcal{K}(\mathbf{V}; \mathbf{X}^{t+1}, \mathrm{B}^{t+1})$ at the point $\mathbf{V} = \mathbf{I}_k$; step ② uses Equality (51) with $\tilde{\mathbf{A}} = \partial F(\mathbf{X}^{t+1})(\mathbf{X}^{t+1})^\mathsf{T} \ominus \mathbf{X}^{t+1}(\partial F(\mathbf{X}^{t+1}))^\mathsf{T}$; step ③ uses the definition of $\Omega_0$ in Equation (47); step ④ uses Lemma A.2 and the fact that $(a + b) \leq 2a^2 + 2b^2$ for all $a, b \in \mathbb{R}$; step ⑤ uses Equality (50); step ⑥ uses the inequality $(a + b + c) \leq 3a^2 + 3b^2 + 3c^2$ for all $a, b, c \in \mathbb{R}$.

We now establish individual bounds for each term in Inequality (52). For the first term $8\mathbb{E}_{\xi^t}[\|\Omega_0 - \Omega_1\|_\mathsf{F}^2]$ in (52), we have:

$$8\mathbb{E}_{\xi^t}[\|\Omega_0 - \Omega_1\|_\mathsf{F}^2]$$

$$\leq 8\mathbb{E}_{\xi^t}[\|\mathrm{U}_{\mathrm{B}^t}^\mathsf{T}[\nabla f(\mathbf{X}^{t+1}) + \mathbb{H}^{t+1}][\mathbf{X}^{t+1} - \mathbf{X}^t]^\mathsf{T}\mathrm{U}_{\mathrm{B}^t}\|_\mathsf{F}^2]$$

$$\overset{①}{=} 8\mathbb{E}_{\xi^t}[\|\mathrm{U}_{\mathrm{B}^t}^\mathsf{T}[\nabla f(\mathbf{X}^{t+1}) + \mathbb{H}^{t+1}][\mathrm{U}_\mathrm{B}(\bar{\mathbf{V}}^t - \mathbf{I}_k)\mathrm{U}_{\mathrm{B}^t}\mathbf{X}^t]^\mathsf{T}\mathrm{U}_{\mathrm{B}^t}\|_\mathsf{F}]$$

$$\overset{②}{\leq} 8C_F^2\mathbb{E}_{\xi^t}[\|\bar{\mathbf{V}}^t - \mathbf{I}_k\|_\mathsf{F}^2], \tag{53}$$

where step ① uses $\mathbf{X}^{t+1} = \mathbf{X}^t + \mathrm{U}_\mathrm{B}(\bar{\mathbf{V}}^t - \mathbf{I}_k)\mathrm{U}_\mathrm{B}^\mathsf{T}\mathbf{X}^t$; step ② uses the inequality $\|\mathbf{X}\mathbf{Y}\|_\mathsf{F} \leq \|\mathbf{X}\|_\mathsf{F}\|\mathbf{Y}\|_\mathsf{sp}$ for all $\mathbf{X}$ and $\mathbf{Y}$ repeatedly, and the fact that $\|\mathbf{G}\|_\mathsf{F} \leq C_F$ for all $\mathbf{X} \in \mathrm{St}(n, r)$ and all $\mathbf{G} \in \partial F(\mathbf{X})$.

For the second term $6\mathbb{E}_{\xi^t}[\|\bar{\mathbf{V}}^t\Upsilon_1^\mathsf{T}\bar{\mathbf{V}}^t - \Upsilon_1\|_\mathsf{F}^2]$ in (52), we have::

$$6\mathbb{E}_{\xi^t}[\|\bar{\mathbf{V}}^t\Upsilon_1^\mathsf{T}\bar{\mathbf{V}}^t - \Upsilon_1\|_\mathsf{F}^2]$$

$$\overset{①}{\leq} 12\mathbb{E}_{\xi^t}[\|\bar{\mathbf{V}}^t\Upsilon_1^\mathsf{T}\bar{\mathbf{V}}^t\|_\mathsf{F}^2] + 12\mathbb{E}_{\xi^t}[\|\Upsilon_1\|_\mathsf{F}^2]$$

$$\overset{②}{\leq} 24\mathbb{E}_{\xi^t}[\|\Upsilon_1\|_\mathsf{F}^2]$$

$$\overset{③}{=} 24\mathbb{E}_{\xi^t}[\|\mathrm{mat}((\mathbf{Q} + \alpha\mathbf{I}_k)\mathrm{vec}(\bar{\mathbf{V}}^t - \mathbf{I}_k))\|_\mathsf{F}^2]$$

$$\leq 24\mathbb{E}_{\xi^t}[\|\mathbf{Q} + \alpha\mathbf{I}_k\|_\mathsf{sp}^2 \cdot \|\bar{\mathbf{V}}^t - \mathbf{I}_k)\|_\mathsf{F}]$$

$$\overset{④}{\leq} 24(L_f + \alpha)^2 \cdot \mathbb{E}_{\xi^t}[\|\bar{\mathbf{V}}^t - \mathbf{I}_k)\|_\mathsf{F}^2] \tag{54}$$

where step ① uses the triangle inequality; step ② uses the inequality $\|\mathbf{X}\mathbf{Y}\|_\mathsf{F} \leq \|\mathbf{X}\|_\mathsf{F}\|\mathbf{Y}\|_\mathsf{sp}$ for all $\mathbf{X}$ and $\mathbf{Y}$; step ③ uses the definition of $\Omega_1$ in (48); step ④ uses the fact that $\|\mathbf{Q}\|_\mathsf{sp} \leq L_f$.

For the third term $6\mathbb{E}_{\xi^t}[\|\bar{\mathbf{V}}^t\Omega_1^\mathsf{T}\bar{\mathbf{V}}^t - \Omega_1^\mathsf{T}\|_\mathsf{F}^2]$ in (52), we have:

$$
\begin{aligned}
& 6\mathbb{E}_{\xi^t}[\|\bar{\mathbf{V}}^t\Omega_1^\mathsf{T}\bar{\mathbf{V}}^t - \Omega_1^\mathsf{T}\|_\mathsf{F}^2] \\
\overset{\text{①}}{=}\;& 6\mathbb{E}_{\xi^t}[\|\bar{\mathbf{V}}^t\Omega_1^\mathsf{T}(\bar{\mathbf{V}}^t - \mathbf{I}_k) + (\bar{\mathbf{V}}^t - \mathbf{I}_k)\Omega_1^\mathsf{T}\|_\mathsf{F}^2] \\
\overset{\text{②}}{\leq}\;& 12\mathbb{E}_{\xi^t}[\|\Omega_1\|_\mathsf{sp}^2 \cdot \|\bar{\mathbf{V}}^t - \mathbf{I}_k\|_\mathsf{F}^2] \\
\overset{\text{③}}{\leq}\;& 12\mathbb{E}_{\xi^t}[\|\nabla f(\mathbf{X}^{t+1}) + \mathbb{H}^{t+1}\|_\mathsf{sp}^2 \cdot \|\bar{\mathbf{V}}^t - \mathbf{I}_k\|_\mathsf{F}^2] \\
\leq\;& 12C_F^2\mathbb{E}_{\xi^t}[\|\bar{\mathbf{V}}^t - \mathbf{I}_k\|_\mathsf{F}^2],
\end{aligned}
\tag{55}
$$

where step ① uses the fact that $-\bar{\mathbf{V}}^t\Omega_1^\mathsf{T}\mathbf{I}_k + \bar{\mathbf{V}}^t\Omega_1^\mathsf{T} = \mathbf{0}$; step ② uses the norm inequality; step ③ uses the fact that $\|\Omega_1\|_\mathsf{sp} = \|\mathrm{U}_{\mathtt{B}^t}^\mathsf{T}[\nabla f(\mathbf{X}^{t+1}) + \mathbb{H}^{t+1}][\mathbf{X}^t]^\mathsf{T}\mathrm{U}_{\mathtt{B}^t}\|_\mathsf{sp} \leq \|\nabla f(\mathbf{X}^{t+1}) + \mathbb{H}^{t+1}\|_\mathsf{sp} \leq \|\nabla f(\mathbf{X}^{t+1}) + \mathbb{H}^{t+1}\|_\mathsf{F}$ which can be derived using the norm inequality.

For the fourth term $6\mathbb{E}_{\xi^t}[\|\bar{\mathbf{V}}^t\Omega_2^\mathsf{T}\bar{\mathbf{V}}^t - \Omega_2\|_\mathsf{F}^2]$ in (52), we have:

$$
\begin{aligned}
& 6\mathbb{E}_{\xi^t}[\|\bar{\mathbf{V}}^t\Omega_2^\mathsf{T}\bar{\mathbf{V}}^t - \Omega_2\|_\mathsf{F}^2] \\
\overset{\text{①}}{\leq}\;& 12\mathbb{E}_{\xi^t}[\|\bar{\mathbf{V}}^t\Omega_2^\mathsf{T}\bar{\mathbf{V}}^t\|_\mathsf{F}^2] + 12\mathbb{E}[\|\Omega_2\|_\mathsf{F}^2] \\
\overset{\text{②}}{\leq}\;& 24\mathbb{E}_{\xi^t}[\|\Omega_2\|_\mathsf{F}^2] \\
\overset{\text{③}}{=}\;& 24\mathbb{E}_{\xi^t}[\|\mathrm{U}_{\mathtt{B}^t}^\mathsf{T}[\nabla f(\mathbf{X}^t) - \nabla f(\mathbf{X}^{t+1})][\mathbf{X}^t]^\mathsf{T}\mathrm{U}_{\mathtt{B}^t}\|_\mathsf{F}^2] \\
\overset{\text{④}}{\leq}\;& 24\mathbb{E}_{\xi^t}[\|\nabla f(\mathbf{X}^t) - \nabla f(\mathbf{X}^{t+1})\|_\mathsf{F}^2] \\
\overset{\text{⑤}}{\leq}\;& 24L_f^2\mathbb{E}_{\xi^t}[\|\mathbf{X}^t - \mathbf{X}^{t+1}\|_\mathsf{F}^2] \\
\overset{\text{⑥}}{\leq}\;& 24L_f^2\mathbb{E}_{\xi^t}[\|\bar{\mathbf{V}}^t - \mathbf{I}_k\|_\mathsf{F}^2],
\end{aligned}
\tag{56}
$$

where step ① uses the triangle inequality; step ② uses the norm inequality; step ③ uses the definition of $\Omega_2 = \mathrm{U}_{\mathtt{B}^t}^\mathsf{T}[\nabla f(\mathbf{X}^t) - \nabla f(\mathbf{X}^{t+1})][\mathbf{X}^t]^\mathsf{T}\mathrm{U}_{\mathtt{B}^t}$ in (49); step ④ uses the norm inequality; step ⑤ uses the fact that $\nabla f(\mathbf{X})$ is $L_f$-Lipschitz continuous; step ⑥ uses Lemma 2.2(c).

In view of (53), (54), (55), (56), and (52), we have:

$$
\mathbb{E}_{\xi^{t+1}}[\|\partial_\mathcal{M}\mathcal{K}(\mathbf{I}_k; \mathbf{X}^{t+1}, \mathtt{B}^{t+1})\|_\mathsf{F}^2] \leq (c_1 + c_2 + c_3 + c_4) \cdot \mathbb{E}_{\xi^t}[\|\bar{\mathbf{V}}^t - \mathbf{I}_k\|_\mathsf{F}^2],
$$

where $c_1 = 8C_F^2$, $c_2 = 24(L_f+\alpha)^2$, $c_3 = 12C_F^2$, and $c_4 = 24L_f^2$. Defining $\phi \triangleq 72(C_F^2+\alpha^2+L_f^2)$, we conclude that $\mathbb{E}_{\xi^{t+1}}[\|\partial_\mathcal{M}\mathcal{K}(\mathbf{I}_k; \mathbf{X}^{t+1}, \mathtt{B}^{t+1})\|_\mathsf{F}^2] \leq \phi\mathbb{E}_{\xi^t}[\|\bar{\mathbf{V}}^t - \mathbf{I}_k\|_\mathsf{F}^2]$.

**Part (b)**. We show that $\mathbb{E}_{\xi^t}[\mathrm{dist}^2(\mathbf{0}, \partial_\mathcal{M}F(\mathbf{X}^t))] \leq \gamma \cdot \mathbb{E}_{\xi^t}[\mathrm{dist}^2(\mathbf{0}, \partial_\mathcal{M}\mathcal{K}(\mathbf{I}_k; \mathbf{X}^t, \mathtt{B}^t))]$, where $\gamma \triangleq \mathrm{C}_n^k/\mathrm{C}_{n-2}^{k-2}$. For all $\mathbf{D}^t \triangleq \partial F(\mathbf{X}^t)[\mathbf{X}^t]^\mathsf{T} \ominus \mathbf{X}^t[\partial F(\mathbf{X}^t)]^\mathsf{T}$, we obtain:

$$
\begin{aligned}
\|\mathbf{D}^t\|_\mathsf{F}^2 \;=\;& \textstyle\sum_i \sum_{j\neq i}(\mathbf{D}_{ij}^t)^2 + \sum_i \sum_{j=i}(\mathbf{D}_{ij}^t)^2 \\
\overset{\text{①}}{=}\;& \textstyle\sum_i \sum_{j\neq i}(\mathbf{D}_{ij}^t)^2 \\
\overset{\text{②}}{=}\;& \tfrac{1}{\mathrm{C}_{n-2}^{k-2}} \textstyle\sum_{i=1}^{\mathrm{C}_n^k} \|\mathbf{U}_{\mathcal{B}_i}^\mathsf{T}\mathbf{D}^t\mathbf{U}_{\mathcal{B}_i}\|_\mathsf{F}^2 \\
\overset{\text{③}}{=}\;& \tfrac{1}{\mathrm{C}_{n-2}^{k-2}} \cdot \mathrm{C}_n^k\mathbb{E}_{\mathtt{B}^t}[\|\mathrm{U}_{\mathtt{B}^t}^\mathsf{T}\mathbf{D}^t\mathrm{U}_{\mathtt{B}^t}\|_\mathsf{F}^2] \\
\overset{\text{④}}{=}\;& \gamma\mathbb{E}_{\mathtt{B}^t}[\|\mathrm{U}_{\mathtt{B}^t}^\mathsf{T}\mathbf{D}^t\mathrm{U}_{\mathtt{B}^t}\|_\mathsf{F}^2],
\end{aligned}
\tag{57}
$$

where step ① uses the fact that $\mathbf{D}_{ii}^t = 0$ for all $i \in [n]$; step ② uses Claim (**a**) of this lemma with $\mathbf{D}_{ii}^t = 0$ for all $i \in [n]$; step ③ uses $\mathbb{E}_{\mathtt{B}^t}[\|\mathrm{U}_{\mathtt{B}^t}^\mathsf{T}\mathbf{W}\mathrm{U}_{\mathtt{B}^t}\|_\mathsf{F}^2] = \tfrac{1}{\mathrm{C}_n^k}\sum_{i=1}^{\mathrm{C}_n^k}\|\mathbf{U}_{\mathcal{B}_i}^\mathsf{T}\mathbf{W}\mathbf{U}_{\mathcal{B}_i}\|_\mathsf{F}^2$ as $\mathtt{B}^t$ are

chosen from $\{\mathcal{B}_i\}_{i=1}^{C_n^k}$ randomly and uniformly; ④ uses the definition of $\gamma$. We further derive:

$$
\begin{aligned}
\mathbb{E}_{\xi^t}\|\partial_{\mathcal{M}}F(\mathbf{X}^t)\|_{\mathsf{F}}^2 \quad &\overset{①}{=} \quad \|\partial F(\mathbf{X}^t) \ominus \mathbf{X}^t[\partial F(\mathbf{X}^t)]^{\mathsf{T}}\mathbf{X}^t\|_{\mathsf{F}}^2 \\
&\overset{②}{=} \quad \|\partial F(\mathbf{X}^t)[\mathbf{X}^t]^{\mathsf{T}}\mathbf{X}^t \ominus \mathbf{X}^t[\partial F(\mathbf{X}^t)]^{\mathsf{T}}\mathbf{X}^t\|_{\mathsf{F}}^2 \\
&\overset{③}{\leq} \quad \|\partial F(\mathbf{X}^t)[\mathbf{X}^t]^{\mathsf{T}} \ominus \mathbf{X}^t[\partial F(\mathbf{X}^t)]^{\mathsf{T}}\|_{\mathsf{F}}^2 \\
&\overset{④}{=} \quad \gamma\mathbb{E}_{\mathsf{B}^t}[\|\mathrm{U}_{\mathsf{B}^t}^{\mathsf{T}}\{\partial F(\mathbf{X}^t)[\mathbf{X}^t]^{\mathsf{T}} \ominus \mathbf{X}^t[\partial F(\mathbf{X}^t)]^{\mathsf{T}}\}\mathrm{U}_{\mathsf{B}^t}\|_{\mathsf{F}}^2] \\
&\overset{⑤}{=} \quad \gamma\|\partial_{\mathcal{M}}\mathcal{K}(\mathbf{I}_k;\mathbf{X}^t,\mathsf{B}^t)\|_{\mathsf{F}}^2
\end{aligned}
\tag{58}
$$

where step ① uses the definition of $\partial_{\mathcal{M}}F(\mathbf{X}^t)$); step ② uses $[\mathbf{X}^t]^{\mathsf{T}}\mathbf{X}^t = \mathbf{I}_k$; step ③ uses the inequality that $\|\mathbf{A}\mathbf{X}\|_{\mathsf{F}}^2 \leq \|\mathbf{A}\|_{\mathsf{F}}^2$ for all $\mathbf{X} \in \mathrm{St}(n,r)$; step ④ uses Equality (57); step ⑤ uses the definition of $\partial_{\mathcal{M}}\mathcal{K}(\mathbf{I}_k;\mathbf{X}^t,\mathsf{B}^t)$.

$\square$

## F.3 Proof of Theorem 4.6

*Proof.* We derive the following results:

$$
\begin{aligned}
\mathbb{E}_{\xi^T}[\mathrm{dist}^2(\mathbf{0}, \partial_{\mathcal{M}}F(\mathbf{X}^{T+1}))] \quad &\overset{①}{\leq} \quad \gamma \cdot \mathbb{E}_{\xi^{T+1}}[\mathrm{dist}^2(\mathbf{0}, \partial_{\mathcal{M}}\mathcal{K}(\mathbf{I}_k;\mathbf{X}^{T+1},\mathsf{B}^{T+1}))] \\
&\overset{②}{\leq} \quad \gamma \cdot \phi \cdot \mathbb{E}_{\xi^T}[\|\bar{\mathbf{V}}^T - \mathbf{I}_k\|_{\mathsf{F}}^2] \\
&\overset{③}{\leq} \quad \gamma \cdot \phi \cdot \frac{\tilde{c}}{T+1},
\end{aligned}
$$

where step ① uses Lemma 4.4(b); step ② uses Lemma 4.4(a); step ③ uses Inequality (45).

Therefore, we conclude that there exists an index $\bar{t}$ with $\bar{t} \in [0, T]$ such that the associated solution $\mathbf{X}^{\bar{t}}$ qualifies as an $\epsilon$-*critical point* of Problem (1) satisfying $\mathbb{E}_{\xi^{\bar{t}}}[\mathrm{dist}^2(\mathbf{0}, \partial_{\mathcal{M}}F(\mathbf{X}^{\bar{t}+1}))] \leq \epsilon$, provided that $T$ is sufficiently large to ensure $\gamma \cdot \phi \cdot \frac{\tilde{c}}{T+1} \leq \epsilon$.

$\square$

## F.4 Proof of Theorem 4.10

*Proof.* By Theorem 4.2(a) and Theorem 4.6, the composite function $F_\iota(\mathbf{X}) \triangleq F(\mathbf{X}) + \iota_{\mathcal{M}}(\mathbf{X})$ is monotonically non-increasing, i.e., $F_\iota(\mathbf{X}^{t+1}) \leq F_\iota(\mathbf{X}^t)$. Moreover, the sequence $\{\mathbf{X}^t\}_{t=1}^{\infty}$ has a limit point $\mathbf{X}^{\infty}$.

Since $F_\iota(\mathbf{X}) \triangleq F(\mathbf{X}) + \iota_{\mathcal{M}}(\mathbf{X})$ is a KL function by assumption, Proposition 4.9 implies that there exists an index $t_\star \in \mathbb{N}$ such that, for all $t \geq t_\star$,

$$
\frac{1}{\varphi'(F_\iota(\mathbf{X}^t) - F_\iota(\mathbf{X}^{\infty}))} \leq \mathrm{dist}(0, \partial F_\iota(\mathbf{X}^t)).
\tag{59}
$$

Since $\varphi(\cdot)$ is a concave desingularization function, we have: $\varphi(b) + (a-b)\varphi'(a) \leq \varphi(a)$. Applying the inequality above with $a = F(\mathbf{X}^t) - F(\mathbf{X}^{\infty})$ and $b = F(\mathbf{X}^{t+1}) - F(\mathbf{X}^{\infty})$, we have:

$$
\begin{aligned}
&(F(\mathbf{X}^t) - F(\mathbf{X}^{t+1}))\varphi'(F(\mathbf{X}^t) - F(\mathbf{X}^{\infty})) \\
\leq \quad &\underbrace{\varphi(F(\mathbf{X}^t) - F(\mathbf{X}^{\infty}))}_{\triangleq\varphi_t} - \varphi(F(\mathbf{X}^{t+1}) - F(\mathbf{X}^{\infty})).
\end{aligned}
\tag{60}
$$

**Part (a)**. We derive the following inequalities:

$$
\begin{aligned}
(E_{t+1})^2 \triangleq \mathbb{E}_{\xi^t}[\|\bar{\mathbf{V}}^t - \mathbf{I}_k\|_{\mathsf{F}}^2] \;\overset{①}{\leq}\;& \tfrac{2}{\alpha} \cdot \mathbb{E}_{\xi^t}[F(\mathbf{X}^t) - F(\mathbf{X}^{t+1})] \\
\overset{②}{\leq}\;& \tfrac{2}{\alpha} \cdot \mathbb{E}_{\xi^t}[(\varphi_t - \varphi_{t+1}) \cdot \tfrac{1}{\varphi'(F(\mathbf{X}^t) - F(\mathbf{X}^\infty))}] \\
\overset{③}{\leq}\;& \tfrac{2}{\alpha} \cdot \mathbb{E}_{\xi^t}[(\varphi_t - \varphi_{t+1}) \cdot \mathrm{dist}(0, \partial F_\iota(\mathbf{X}^t))] \\
\overset{④}{\leq}\;& \tfrac{2}{\alpha} \cdot \mathbb{E}_{\xi^t}[(\varphi_t - \varphi_{t+1}) \cdot \|\partial_{\mathcal{M}} F(\mathbf{X}^t)\|_{\mathsf{F}}] \\
\overset{⑤}{\leq}\;& \tfrac{2}{\alpha} \cdot \mathbb{E}_{\xi^t}[(\varphi_t - \varphi_{t+1})\sqrt{\gamma}\|\partial_{\mathcal{M}}\mathcal{K}(\mathbf{I}_k; \mathbf{X}^t, \textsc{b}^t)\|_{\mathsf{F}}] \\
\overset{⑥}{\leq}\;& \tfrac{2}{\alpha} \cdot \mathbb{E}_{\xi^t}[(\varphi_t - \varphi_{t+1})\sqrt{\gamma\phi}\|\bar{\mathbf{V}}^{t-1} - \mathbf{I}_k\|_{\mathsf{F}}] \\
\overset{⑦}{=}\;& \underbrace{\tfrac{2}{\alpha} \cdot \sqrt{\gamma\phi}}_{\triangleq\kappa} \cdot (\varphi_t - \varphi_{t+1}) \cdot E_t,
\end{aligned}
$$

where step ① uses the sufficient decrease condition as shown in Theorem 4.2; step ② uses Inequality (60); step ③ uses Inequality (59); step ④ uses Lemma A.7; step ⑤ uses Inequality (58); step ⑥ uses Lemma 4.4; step ⑦ uses the definitions of $\{\kappa, \varphi_t, E_t\}$.

**Part (b)**. Applying Lemma A.9 with $p_t = \kappa\varphi_t$ with $p_t \geq p_{t+1}$, for all $i \geq 1$, we have:

$$
\sum_{j=i}^{\infty} E_{j+1} \leq E_i + 2p_i.
$$

Using the definition of $D_t \triangleq \sum_{j=t}^{\infty} E_{j+1}$ and letting $i = t$, we obtain:

$$
D_t \leq E_t + 2p_t \overset{①}{=} E_t + 2\kappa\varphi_t \overset{②}{\leq} E_t + 2\kappa\varphi_1 \overset{③}{\leq} 2\sqrt{k} + 2\kappa\varphi_1,
$$

where step ① uses $p_t = \kappa\varphi_t$; step ② uses $\varphi_t \leq \varphi_1$; step ③ uses $E_t \triangleq \big(\mathbb{E}_{\xi^{t-1}}[\|\bar{\mathbf{V}}^{t-1} - \mathbf{I}_k\|_{\mathsf{F}}]\big)^{1/2}$ and $\|\bar{\mathbf{V}}^{t-1} - \mathbf{I}_k\|_{\mathsf{F}} \leq \|\bar{\mathbf{V}}^{t-1}\|_{\mathsf{F}} + \|\mathbf{I}_k\|_{\mathsf{F}} \leq \sqrt{k} + \sqrt{k}$. We conclude that $D_t \triangleq \sum_{j=t}^{\infty} E_{j+1}$ is always upper-bounded.

Using the fact that $\|\mathbf{X}^{t+1} - \mathbf{X}^t\|_{\mathsf{F}}^2 \leq \|\bar{\mathbf{V}}^t - \mathbf{I}_k\|_{\mathsf{F}}^2$ as shown in Lemma 2.2(*c*), we conclude that $\sum_{i=1}^{\infty} \mathbb{E}_{\xi^i}[\|\mathbf{X}^{i+1} - \mathbf{X}^i\|_{\mathsf{F}}]$ is also always upper-bounded.

$\square$

### F.5 PROOF OF THEOREM 4.11

*Proof.* We define $\varphi_t \triangleq \varphi(s^t)$, where $s^t \triangleq F(\mathbf{X}^t) - F(\mathbf{X}^\infty)$.

We define $E_{t+1} \triangleq \big(\mathbb{E}_{\xi^t}[\|\bar{\mathbf{V}}^t - \mathbf{I}_k\|_{\mathsf{F}}^2]\big)^{1/2}$, and $D_i = \sum_{j=i}^{\infty} E_{j+1}$.

We have: $D_{t-1} - D_t = E_t \leq 2\sqrt{k} \triangleq \bar{r}$.

First, we have:

$$
\begin{aligned}
\|\mathbf{X}^T - \mathbf{X}^\infty\|_{\mathsf{F}} \;\overset{①}{\leq}\;& \sum_{j=T}^{\infty} \|\mathbf{X}^j - \mathbf{X}^{j+1}\|_{\mathsf{F}} \\
\overset{②}{\leq}\;& \sum_{j=T}^{\infty} \|\bar{\mathbf{V}}^j - \mathbf{I}_k\|_{\mathsf{F}} \\
\overset{③}{=}\;& \sum_{j=T}^{\infty} E_{j+1} \\
\overset{④}{=}\;& D_T,
\end{aligned}
$$

where step ① uses the triangle inequality; step ② uses $\|\mathbf{X}^{t+1} - \mathbf{X}^t\|_{\mathsf{F}}^2 \leq \|\bar{\mathbf{V}}^t - \mathbf{I}_k\|_{\mathsf{F}}^2$, as shown in Lemma 2.2(*c*); step ③ uses the definition of $E_{t+1}$; step ④ uses the definition of $D_T$. Therefore, it suffices to establish the convergence rate of $D_T$.

Second, we obtain the following results:

$$
\begin{aligned}
\mathbb{E}_{\xi^t}[\tfrac{1}{\varphi'(s^t)}] \quad &\overset{①}{\leq} \quad \mathbb{E}_{\xi^t}[\|\mathrm{dist}(\mathbf{0}, \partial F_\iota(\mathbf{X}^t))\|_{\mathsf{F}}] \\
&\overset{②}{\leq} \quad \mathbb{E}_{\xi^t}[\|\partial_{\mathcal{M}} F(\mathbf{X}^t)\|_{\mathsf{F}}] \\
&\overset{③}{\leq} \quad \mathbb{E}_{\xi^t}[\sqrt{\gamma}\|\partial_{\mathcal{M}}\mathcal{K}(\mathbf{I}_k; \mathbf{X}^t, \mathsf{B}^t)\|_{\mathsf{F}}] \\
&\overset{④}{\leq} \quad \mathbb{E}_{\xi^t}[\sqrt{\gamma\phi}\|\bar{\mathbf{V}}^{t-1} - \mathbf{I}_k\|_{\mathsf{F}}] \\
&\overset{⑤}{\leq} \quad \sqrt{\gamma\phi}E_t,
\end{aligned}
\tag{61}
$$

where step ① uses uses Proposition 4.9 that $\mathrm{dist}(\mathbf{0}, \partial F_\iota(\mathbf{X}'))\varphi'(F_\iota(\mathbf{X}') - F_\iota(\mathbf{X}^\infty)) \geq 1$; step ② uses Lemma A.7; step ③ uses Inequality (58); step ④ uses the Riemannian subgradient lower bound for the iterates gap in Lemma 4.4; step ⑤ uses the definition of $E_t \triangleq \mathbb{E}_{\xi^{t-1}}[\|\bar{\mathbf{V}}^{t-1} - \mathbf{I}_k\|_{\mathsf{F}}^2]$.

Third, using the definition of $D_t$, we derive:

$$
\begin{aligned}
D_t \quad &\triangleq \quad \sum_{i=t}^{\infty} E_{i+1} \\
&\overset{①}{\leq} \quad E_t + 2\kappa\varphi_t \\
&\overset{②}{=} \quad E_t + 2\kappa c \cdot \{[s^t]^\sigma\}^{\frac{1-\sigma}{\sigma}} \\
&\overset{③}{=} \quad E_t + 2\kappa c \cdot \{c(1-\sigma) \cdot \tfrac{1}{\varphi'(s^t)}\}^{\frac{1-\sigma}{\sigma}} \\
&\overset{④}{=} \quad E_t + 2\kappa c \cdot \{c(1-\sigma) \cdot \sqrt{\gamma\phi}E_t\}^{\frac{1-\sigma}{\sigma}} \\
&\overset{⑤}{=} \quad D_{t-1} - D_t + 2\kappa c \cdot \{c(1-\sigma) \cdot \sqrt{\gamma\phi}(D_{t-1} - D_t)\}^{\frac{1-\sigma}{\sigma}} \\
&= \quad D_{t-1} - D_t + \underbrace{2\kappa c \cdot [c(1-\sigma)\sqrt{\gamma\phi}]^{\frac{1-\sigma}{\sigma}}}_{\triangleq \ddot{\kappa}} \cdot \{D_{t-1} - D_t\}^{\frac{1-\sigma}{\sigma}},
\end{aligned}
\tag{62}
$$

where step ① uses $\sum_{i=t}^{\infty} E_{i+1} \leq E_t + 2\kappa\varphi_t$, as shown in Theorem 4.10(**b**); step ② uses the definitions that $\varphi_t \triangleq \varphi(s^t)$, and $\varphi(s) = cs^{1-\sigma}$; step ③ uses $\varphi'(s) = c(1-\sigma) \cdot [s]^{-\sigma}$, leading to $[s^t]^\sigma = c(1-\sigma) \cdot \tfrac{1}{\varphi'(s^t)}$; step ④ uses Inequality (61); step ⑤ uses the fact that $E_t = D_{t-1} - D_t$.

We consider three cases for $\sigma \in [0, 1)$.

**Part (a)**. We consider $\sigma = 0$. We have from Inequality (61):

$$
\begin{aligned}
0 \quad &\leq \quad \mathbb{E}_{\xi^t}[-\tfrac{1}{\varphi'(s^t)} + \sqrt{\gamma\phi}E_t] \\
&\overset{①}{=} \quad \mathbb{E}_{\xi^t}[-\tfrac{1}{c(1-\sigma)\cdot[s^t]^{-\sigma}} + \sqrt{\gamma\phi}E_t] \\
&\overset{②}{=} \quad \mathbb{E}_{\xi^t}[-\tfrac{1}{c} + \sqrt{\gamma\phi}E_t],
\end{aligned}
\tag{63}
$$

where step ① uses $\varphi'(s) = c(1-\sigma) \cdot [s]^{-\sigma}$; step ② uses $\sigma = 0$ and $E_t = D_{t-1} - D_t$.

Since $E_t \to 0$, and $\gamma, \phi, c > 0$, Inequality (63) results in a contradiction $E_t \geq \tfrac{1}{c\sqrt{\gamma\phi}} > 0$. Therefore, there exists $t'$ such that $D_t = 0$ for all $t > t'$, ensuring that the algorithm terminates in a finite number of steps.

**Part (b)**. We consider $\sigma \in (0, \tfrac{1}{2}]$. We define $w \triangleq \tfrac{1-\sigma}{\sigma} \geq 1$. We have from Inequality (62):

$$
\begin{aligned}
D_t \quad &\leq \quad D_{t-1} - D_t + (D_{t-1} - D_t)^w \cdot \ddot{\kappa} \\
&\overset{①}{\leq} \quad D_{t-1} - D_t + (D_{t-1} - D_t) \cdot \bar{r}^{w-1} \cdot \ddot{\kappa} \\
&\leq \quad D_{t-1} \cdot \tfrac{\bar{r}^{w-1} \cdot \ddot{\kappa} + 1}{\bar{r}^{w-1} \cdot \ddot{\kappa} + 2},
\end{aligned}
\tag{64}
$$

where step ① uses the fact that $x^w \leq x \cdot \bar{r}^{w-1}$ for all $\sigma \in (0, \tfrac{1}{2}]$, and $x = D_{t-1} - D_t \in [0, \bar{r}]$. Therefore, we have:

$$
D_T \leq D_1 \cdot \left(\tfrac{\bar{r}^{w-1} \cdot \ddot{\kappa} + 1}{\bar{r}^{w-1} \cdot \ddot{\kappa} + 2}\right)^{T-1}.
$$

**Part (c).** We consider $\sigma \in (\frac{1}{2}, 1)$. We define $w \triangleq \frac{1-\sigma}{\sigma} \in (0, 1)$, and $\tau \triangleq 1/w - 1 \in (0, \infty)$. We have from Inequality (62):

$$
\begin{aligned}
D_t &\leq D_{t-1} - D_t + \ddot{\kappa} \cdot (D_{t-1} - D_t)^{\frac{1-\sigma}{\sigma}} \\
&\overset{①}{=} \ddot{\kappa}(D_{t-1} - D_t)^w + (D_{t-1} - D_t)^w \cdot (E_t)^{1-w} \\
&\overset{②}{\leq} \ddot{\kappa}(D_{t-1} - D_t)^w + (D_{t-1} - D_t)^w \cdot \overline{r}^{1-w} \\
&= (D_{t-1} - D_t)^w \cdot \underbrace{(\ddot{\kappa} + \overline{r}^{1-w})}_{\triangleq \dot{\kappa}},
\end{aligned}
$$

where step ① uses the definition of $w$ and the fact that $D_{t-1} - D_t = E_t$; step ② uses the fact that $\max_{x \in (0, \overline{r}]} x^{1-w} \leq \overline{r}^{1-w}$ if $w \in (0, 1)$. We further obtain:

$$
\underbrace{[D_t]^{1/w}}_{=[D_t]^{\tau+1}} \leq (D_{t-1} - D_t) \cdot \dot{\kappa}^{1/w}.
$$

Applying Lemma A.10 with $a = \dot{\kappa}^{1/w}$, we have:

$$
D_T \leq \mathcal{O}(T^{-1/\tau}) \overset{①}{=} \mathcal{O}(T^{-\frac{1}{1/w-1}}) \overset{②}{=} \mathcal{O}(T^{-\frac{1}{\frac{\sigma}{1-\sigma}-1}}) = \mathcal{O}(T^{-\frac{1-\sigma}{2\sigma-1}}),
$$

where step ① uses $\tau \triangleq 1/w - 1$; step ② uses $w \triangleq \frac{1-\sigma}{\sigma}$.

$\square$

# G  ADDITIONAL EXPERIMENT DETAILS AND RESULTS

This section provides additional experimental details and results for our proposed methods. We first introduce nonnegative PCA as an additional application, describe the datasets and experimental settings, and specify the compared baselines for $\ell_1$-regularized SPCA and nonnegative PCA. We then report extended results on $\ell_0$-regularized SPCA, $\ell_1$-regularized SPCA, and nonnegative PCA, demonstrating the effectiveness and robustness of our algorithms across these settings.

## G.1  ADDITIONAL APPLICATION: NONNEGATIVE PCA

Nonnegative PCA is an extension of PCA that imposes nonnegativity constraints on the principal vector (Zass & Shashua, 2006; Qian et al., 2021). This constraint leads to a nonnegative representation of loading vectors and it helps to capture data locality in feature selection. Nonnegative PCA can formulated as: $\min_{\mathbf{X} \in \mathrm{St}(n,r)} -\frac{1}{2}\langle \mathbf{CX}, \mathbf{X} \rangle$, $s.t.\ \mathbf{X} \geq \mathbf{0}$, where $\mathbf{C} \in \mathbb{R}^{n \times n}$ is the covariance matrix of the data.

## G.2  DATA SETS

To generate the data matrix $\mathbf{A} \in \mathbb{R}^{m \times n}$, we consider 10 publicly available real-world or randomly generated data sets: 'w1a', 'TDT2', '20News', 'sector', 'E2006', 'MNIST', 'Gisette', 'Caltech', 'Cifar', 'randn'. We randomly select a subset of examples from the original data set. The size of $\mathbf{A} \in \mathbb{R}^{m \times n}$ is chosen from the following set $(m, n) \in \{(2477, 300), (500, 1000), (8000, 1000), (6412, 1000), (2000, 1000), (60000, 784), (3000, 1000), (1000, 1000), (500, 1000)\}$. We scale the matrix $\mathbf{A}$ to have unit Frobenius norm by setting $\mathbf{A} = \frac{\mathbf{A}}{\|\mathbf{A}\|_{\mathsf{F}}}$ and let $\mathbf{C} = \mathbf{A}^{\mathsf{T}}\mathbf{A} \in \mathbb{R}^{n \times n}$.

## G.3  ADDITIONAL EXPERIMENT SETTINGS

▶ **Compared Methods on $L_1$-Regularized SPCA.** We benchmark **OBCD** against the following state-of-the-art algorithms: (i) Randomized Submanifold Subgradient Method (RSSM) (Cheung et al., 2024); (ii) Linearized Alternating Direction Method of Multiplier (LADMM) (He & Yuan, 2012); (iii) Riemannian Subgradient Method (RSubGrad) (Li et al., 2021); (iv) ADMM (Lai & Osher, 2014); (v) Manifold Proximal Gradient Method (ManPG) (Chen et al., 2020). For RSSM

and RSubGrad, the subgradient $\mathbf{G}^t \in \partial F(\mathbf{X}^t)$ at iterate $\mathbf{X}^t$ is taken as $\mathbf{G}^t = -\mathbf{C}\mathbf{X}^t + \lambda \mathrm{sign}(\mathbf{X}^t)$, since $\mathrm{sign}(\mathbf{X})$ is a valid subgradient of $\|\mathbf{X}\|_1$. All competing methods are initialized with a random matrix, producing five variants: RSSM (rnd), LADMM (rnd), RSubGrad (rnd), ADMM (rnd), and ManPG (rnd). For **OBCD**, we employ a random working-set rule with identity initialization, denoted by **OBCD-R**(id).

▶ **Compared Methods on Nonnegative PCA**. For Nonnegative PCA, we compare **OBCD** with two leading infeasible approaches: *(i)* Linearized ADMM (LADMM) (He & Yuan, 2012; Lai & Osher, 2014), *(ii)* Penalty-based Splitting Method (PSM) (Yuan, 2024; Chen, 2012), and *(iii)* Riemannian ADMM (RADMM) (Li et al., 2024a). Since LADMM, PSM, and RADMM are infeasible methods and may violate the nonnegativity constraints, we evaluate the quality of intermediate solutions using a surrogate objective, $f(\mathbf{X}) + 1000\|\min(\mathbf{0}, \mathbf{X})\|_\mathsf{F}$ with $\mathbf{X} \in \mathrm{St}(n, r)$, which penalizes any violation of feasibility.

### G.4 Additional Experiment Results

▶ **Results on $L_0$-Regularized SPCA**. For $\lambda \in \{10, 50, 100, 500\}$, Figures 3-6 present the convergence curves of the compared methods on $L_0$-regularized SPCA. Across all setting, **OBCD-R** consistently achieves lower objective values than competing methods, further reinforcing the conclusions drawn in the main paper.

▶ **Results on $L_1$-Regularized SPCA**. For $\lambda \in \{10, 50, 100, 500\}$, Table 2 and Figures 7-10 report objective values obtained by all methods with $r = 20$. Two observations follow. *(i)* ManPG is generally faster than LADMM, ADMM and RSubGrad, which aligns with the findings reported in (Chen et al., 2020). *(ii)* **OBCD-R** consistently achieves lower objective values compared with {LADMM, ADMM, RSubGrad, ManPG}, demonstrating its superior solution quality.

▶ **Results on Nonnegative PCA**. For $r \in \{10, 20, 40, 80\}$, Table 3 reports objective values and feasibility violations measured by $\|\min(\mathbf{0}, \mathbf{X})\|_\mathsf{F}$, while Figures 11-14 show the surrogate objective $f(\mathbf{X}) + 1000\|\min(\mathbf{0}, \mathbf{X})\|_\mathsf{F}$. Two key conclusions can be drawn. *(i)* The proposed methods generally achieve the best overall performance, and **OBCD-R** often substantially outperforms LADMM, PSM, and RADMM by locating stronger stationary points.

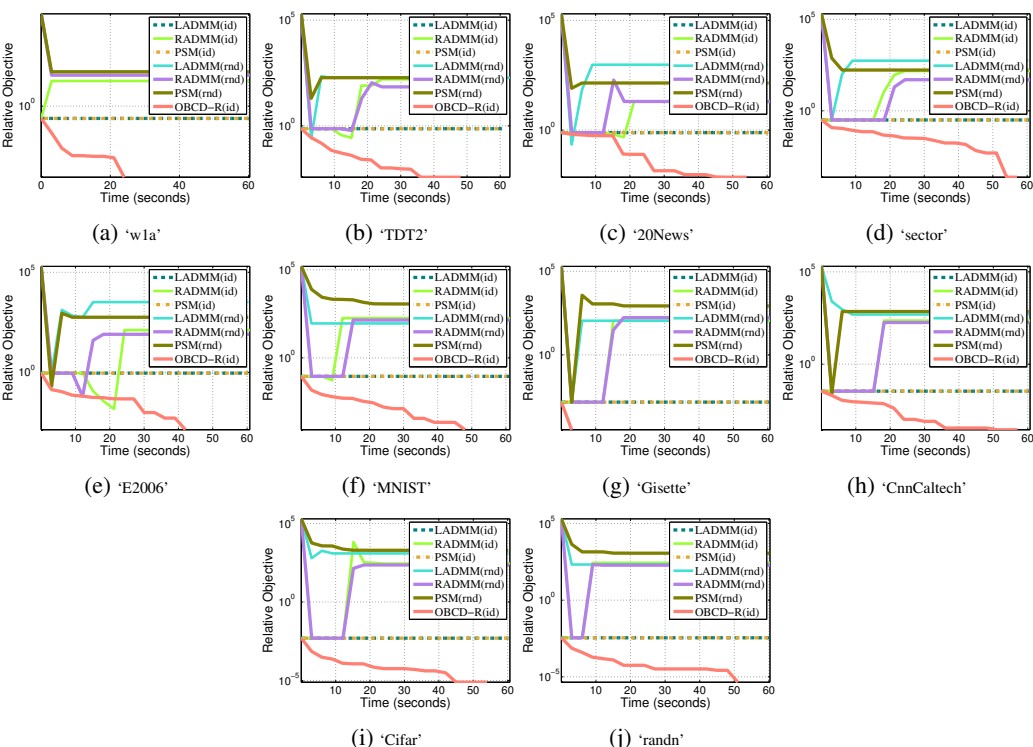

Figure 3: The convergence curve for solving $L_0$-regularized SPCA with $\lambda = 10$.

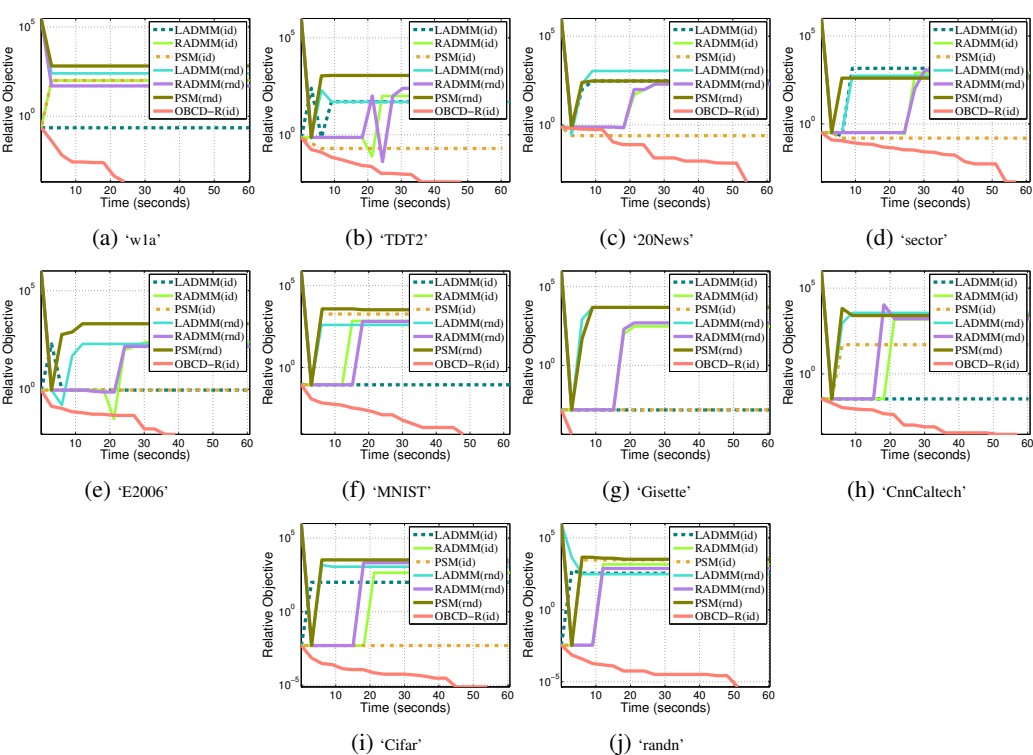

Figure 4: The convergence curve for solving $L_0$-regularized SPCA with $\lambda = 50$.

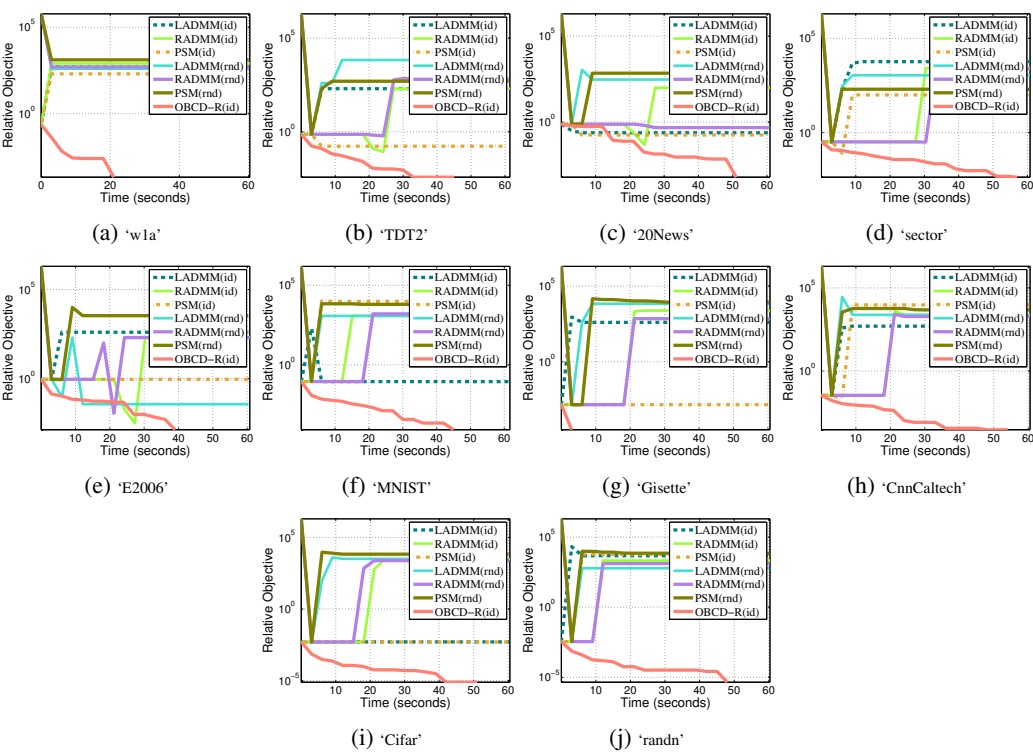

Figure 5: The convergence curve for solving $L_0$-regularized SPCA with $\lambda = 100$.

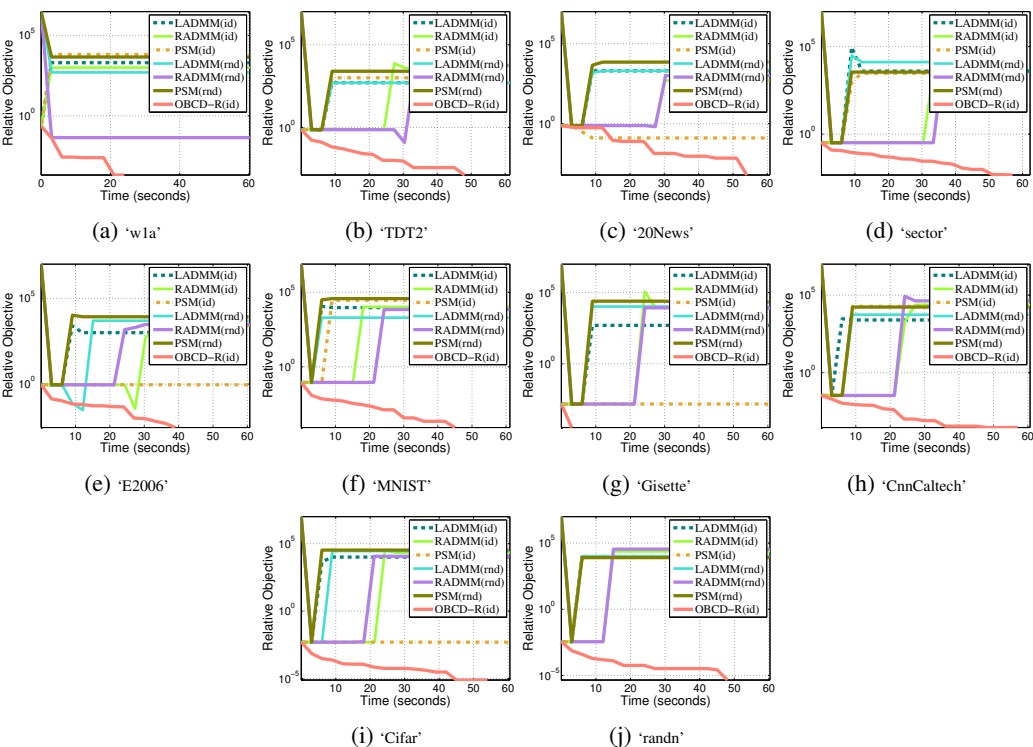

Figure 6: The convergence curve for solving $L_0$-regularized SPCA with $\lambda = 500$.

| data-m-n | RSSM (rnd) | LADMM (rnd) | RSubGrad (rnd) | ADMM (rnd) | ManPG (rnd) | OBCD-R (id) |
|---|---|---|---|---|---|---|
| | | $r = 20$, $\lambda = 10$, time limit=60 | | | | |
| w1a-2477-300 | 1676.362 | 199.961 | 207.918 | 648.546 | 199.949 | 199.833 |
| TDT2-500-1000 | 4798.905 | 199.997 | 376.695 | 2756.315 | 199.999 | 199.636 |
| 20News-8000-1000 | 5099.667 | 203.159 | 458.525 | 2976.634 | 199.997 | 199.673 |
| sector-6412-1000 | 5088.999 | 211.558 | 257.937 | 2646.919 | 199.990 | 199.848 |
| E2006-2000-1000 | 4791.094 | 201.933 | 240.895 | 2873.292 | 200.000 | 199.541 |
| MNIST-60000-784 | 4491.492 | 199.990 | 304.146 | 3077.644 | 199.992 | 199.950 |
| Gisette-3000-1000 | 5096.530 | 203.597 | 361.631 | 3054.472 | 199.990 | 199.989 |
| CnnCaltech-3000-1000 | 5274.750 | 203.177 | 287.583 | 2952.906 | 199.990 | 199.977 |
| Cifar-1000-1000 | 5326.610 | 199.990 | 452.860 | 3007.068 | 199.990 | 199.987 |
| randn-500-1000 | 5299.246 | 207.757 | 267.307 | 2908.559 | 199.990 | 199.988 |

| data-m-n | RSSM (rnd) | LADMM (rnd) | RSubGrad (rnd) | ADMM (rnd) | ManPG (rnd) | OBCD-R (id) |
|---|---|---|---|---|---|---|
| | | $r = 20$, $\lambda = 50$, time limit=60 | | | | |
| w1a-2477-300 | 11896.991 | 1017.039 | 1014.312 | 1948.020 | 999.949 | 999.833 |
| TDT2-500-1000 | 24811.350 | 1142.577 | 5689.161 | 13596.188 | 999.999 | 999.643 |
| 20News-8000-1000 | 25660.045 | 1085.026 | 4852.847 | 15234.296 | 999.997 | 999.673 |
| sector-6412-1000 | 25685.661 | 1076.243 | 5056.712 | 13985.491 | 999.990 | 999.834 |
| E2006-2000-1000 | 23945.851 | 1085.356 | 4102.980 | 13800.413 | 1000.000 | 999.933 |
| MNIST-60000-784 | 22829.255 | 1036.685 | 3035.519 | 15166.657 | 999.992 | 999.949 |
| Gisette-3000-1000 | 25696.928 | 1125.509 | 4866.266 | 15083.925 | 999.990 | 999.989 |
| CnnCaltech-3000-1000 | 26443.995 | 1075.923 | 5648.585 | 14435.979 | 999.990 | 999.977 |
| Cifar-1000-1000 | 26174.415 | 1101.272 | 6080.349 | 14828.673 | 999.990 | 999.987 |
| randn-500-1000 | 25917.437 | 1237.580 | 4616.156 | 14999.881 | 999.990 | 999.988 |

| data-m-n | RSSM (rnd) | LADMM (rnd) | RSubGrad (rnd) | ADMM (rnd) | ManPG (rnd) | OBCD-R (id) |
|---|---|---|---|---|---|---|
| | | $r = 20$, $\lambda = 100$, time limit=60 | | | | |
| w1a-2477-300 | 25212.531 | 2024.330 | 2142.546 | 4172.640 | 1999.949 | 1999.833 |
| TDT2-500-1000 | 49303.568 | 2210.215 | 13770.257 | 27221.640 | 1999.999 | 1999.636 |
| 20News-8000-1000 | 52028.247 | 2204.356 | 12741.678 | 30561.467 | 1999.997 | 1999.673 |
| sector-6412-1000 | 51434.623 | 2222.103 | 17521.186 | 27816.620 | 1999.990 | 1999.834 |
| E2006-2000-1000 | 48063.148 | 2140.058 | 11210.402 | 27411.269 | 2000.000 | 1999.933 |
| MNIST-60000-784 | 46090.059 | 2057.976 | 11107.393 | 30906.421 | 1999.992 | 1999.949 |
| Gisette-3000-1000 | 51396.503 | 2202.300 | 15971.871 | 30698.736 | 1999.990 | 1999.989 |
| CnnCaltech-3000-1000 | 53046.484 | 2230.728 | 9917.898 | 29326.239 | 1999.990 | 1999.977 |
| Cifar-1000-1000 | 52183.021 | 2282.490 | 16736.350 | 30070.764 | 1999.990 | 1999.987 |
| randn-500-1000 | 52275.431 | 2309.568 | 14891.818 | 30522.549 | 1999.990 | 1999.988 |

| data-m-n | RSSM (rnd) | LADMM (rnd) | RSubGrad (rnd) | ADMM (rnd) | ManPG (rnd) | OBCD-R (id) |
|---|---|---|---|---|---|---|
| | | $r = 20$, $\lambda = 500$, time limit=60 | | | | |
| w1a-2477-300 | 144765.556 | 9999.940 | 26452.425 | 14711.906 | 9999.949 | 9999.834 |
| TDT2-500-1000 | 243550.365 | 11006.292 | 177896.188 | 137815.999 | 9999.999 | 9999.636 |
| 20News-8000-1000 | 257513.893 | 10188.884 | 193633.121 | 152343.022 | 9999.997 | 9999.675 |
| sector-6412-1000 | 260801.229 | 9999.915 | 199887.443 | 138927.601 | 9999.990 | 9999.834 |
| E2006-2000-1000 | 236514.992 | 10535.514 | 135563.372 | 143898.385 | 10000.000 | 9999.933 |
| MNIST-60000-784 | 228035.432 | 10306.371 | 146677.728 | 145588.796 | 9999.992 | 9999.948 |
| Gisette-3000-1000 | 261983.906 | 10313.107 | 202913.350 | 152724.051 | 9999.990 | 9999.989 |
| CnnCaltech-3000-1000 | 259056.451 | 10418.351 | 166856.613 | 149325.559 | 9999.990 | 9999.977 |
| Cifar-1000-1000 | 262258.151 | 10874.860 | 195776.730 | 150353.857 | 9999.990 | 9999.987 |
| randn-500-1000 | 257825.619 | 10219.431 | 80831.264 | 137050.323 | 9999.990 | 9999.988 |

Table 2: Comparisons of objective values for $L_1$-regularized SPCA. The $1^{st}$, $2^{nd}$, and $3^{rd}$ best results are colored with red, green and blue, respectively.

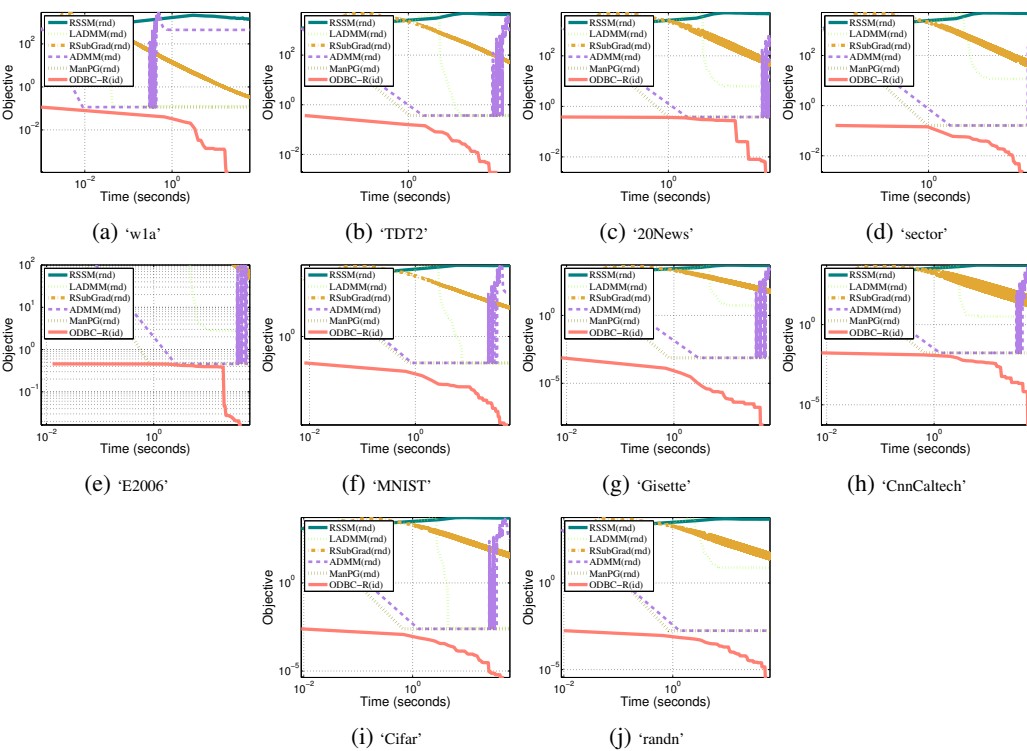

(a) 'w1a'  (b) 'TDT2'  (c) '20News'  (d) 'sector'

(e) 'E2006'  (f) 'MNIST'  (g) 'Gisette'  (h) 'CnnCaltech'

(i) 'Cifar'  (j) 'randn'

Figure 7: The convergence curve for solving $L_1$-regularized SPCA with $\lambda = 10$.

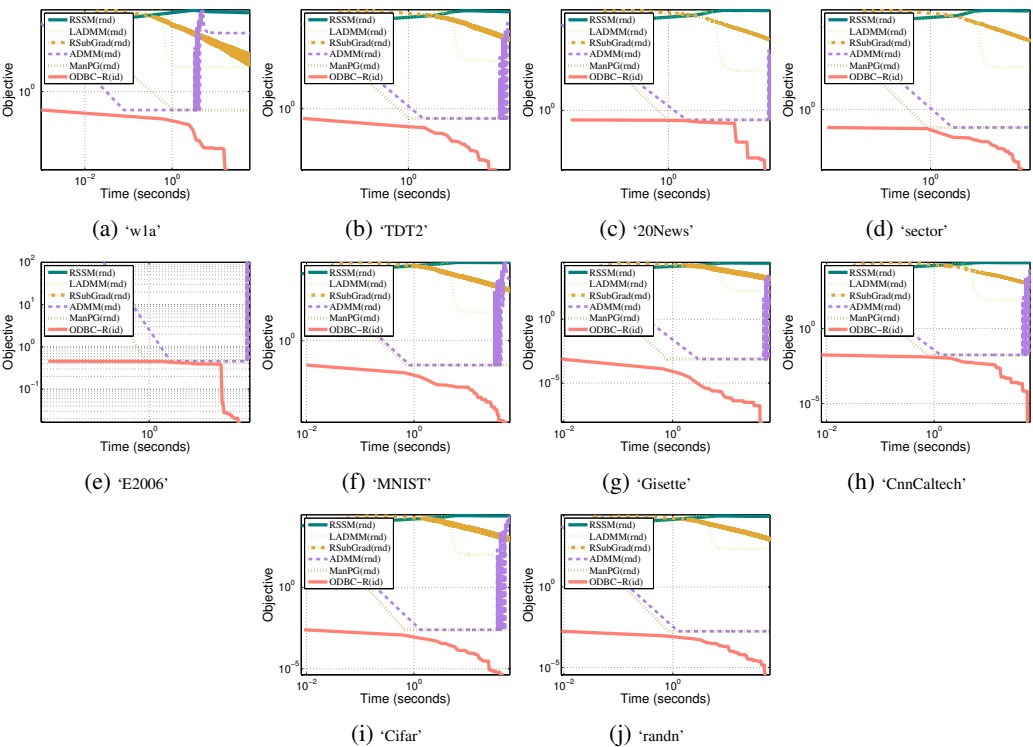

Figure 8: The convergence curve for solving $L_1$-regularized SPCA with $\lambda = 50$.

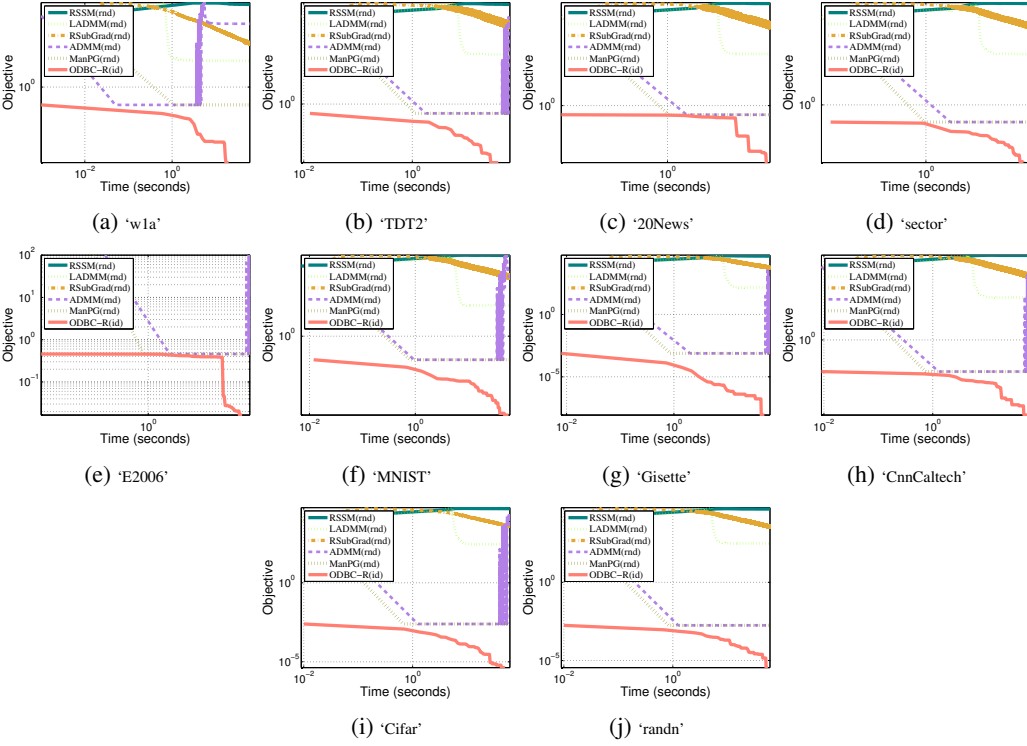

Figure 9: The convergence curve for solving $L_1$-regularized SPCA with $\lambda = 100$.

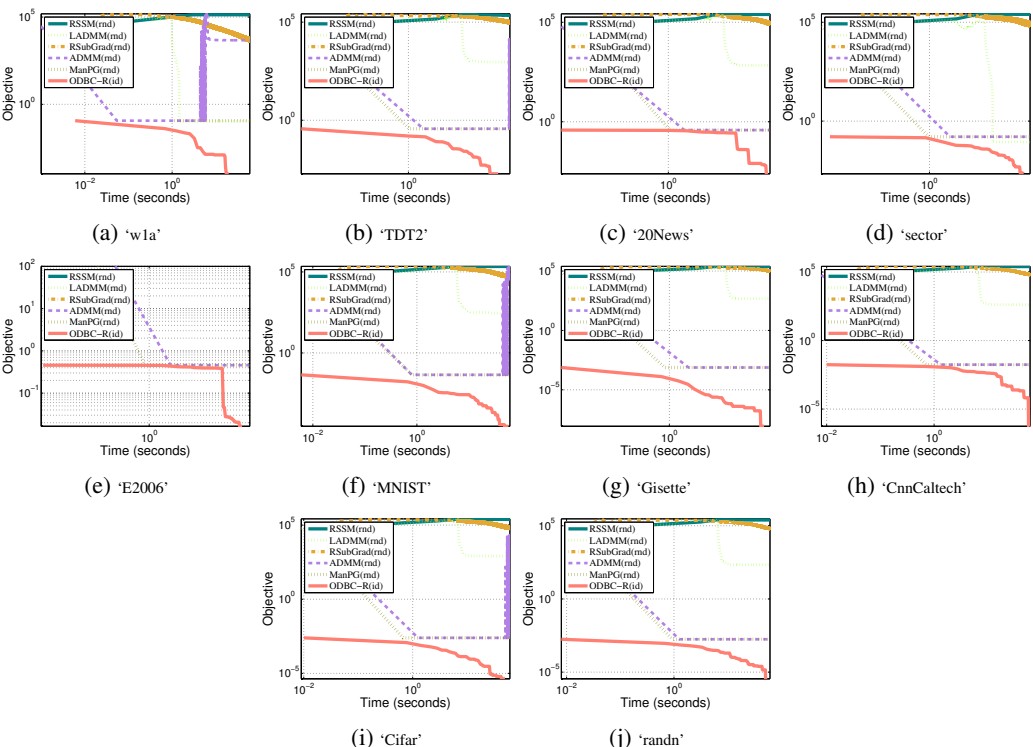

(a) 'w1a'  (b) 'TDT2'  (c) '20News'  (d) 'sector'

(e) 'E2006'  (f) 'MNIST'  (g) 'Gisette'  (h) 'CnnCaltech'

(i) 'Cifar'  (j) 'randn'

Figure 10: The convergence curve for solving $L_1$-regularized SPCA with $\lambda = 500$.

| data-m-n | ADMM (rnd) | PSM (rnd) | RADMM (rnd) | OBCD-R(id) (id) | data-m-n | ADMM (rnd) | PSM (rnd) | RADMM (rnd) | OBCD-R (id) |
|---|---|---|---|---|---|---|---|---|---|
| | | $r = 10$, time limit=60 | | | | | $r = 20$, time limit=60 | | |
| w1a-2477-300 | -4.08e-02, 0e+00 | -4.71e-02, 0e+00 | -1.11e-02, 0e+00 | -1.67e-01, 7e-15 | w1a-2477-300 | -3.73e-02, 0e+00 | -5.36e-02, 0e+00 | -3.68e-02, 0e+00 | -2.17e-01, 3e-15 |
| TDT2-500-1000 | -1.64e-01, 0e+00 | -6.70e-02, 0e+00 | -2.82e-03, 0e+00 | -3.32e-01, 4e-15 | TDT2-500-1000 | -1.73e-03, 0e+00 | -9.53e-02, 0e+00 | -5.01e-03, 0e+00 | -3.71e-01, 2e-15 |
| 20News-8000-1000 | -4.82e-02, 0e+00 | -9.14e-02, 0e+00 | -8.43e-03, 0e+00 | -3.49e-01, 2e-15 | 20News-8000-1000 | -1.31e-03, 0e+00 | -3.14e-02, 0e+00 | -7.71e-03, 0e+00 | -3.78e-01, 4e-15 |
| sector-6412-1000 | -5.70e-03, 0e+00 | -5.84e-03, 0e+00 | -3.30e-03, 0e+00 | -1.21e-01, 1e-15 | sector-6412-1000 | -9.91e-03, 0e+00 | -1.55e-02, 0e+00 | -1.17e-02, 0e+00 | -1.67e-01, 4e-15 |
| E2006-2000-1000 | -3.13e-01, 0e+00 | -3.39e-01, 0e+00 | -6.71e-03, 0e+00 | -4.42e-01, 1e-14 | E2006-2000-1000 | -1.20e-03, 0e+00 | -3.56e-01, 0e+00 | -1.55e-03, 0e+00 | -4.62e-01, 1e-14 |
| MNIST-60000-784 | -3.57e-02, 0e+00 | -9.10e-02, 0e+00 | -3.00e-02, 0e+00 | -2.78e-01, 2e-14 | MNIST-60000-784 | -1.70e-02, 0e+00 | -9.40e-02, 0e+00 | -3.47e-02, 0e+00 | -2.95e-01, 2e-14 |
| Gisette-3000-1000 | -1.41e-01, 0e+00 | -2.34e-01, 0e+00 | -6.84e-02, 0e+00 | -3.72e-01, 2e-18 | Gisette-3000-1000 | -2.23e-02, 0e+00 | -2.31e-01, 0e+00 | -6.05e-02, 0e+00 | -3.80e-01, 7e-19 |
| CnnCaltech-3000-1000 | -2.28e-02, 0e+00 | -6.58e-02, 0e+00 | -2.10e-02, 0e+00 | -1.38e-01, 0e+00 | CnnCaltech-3000-1000 | -1.05e-02, 0e+00 | -6.87e-02, 0e+00 | -3.34e-02, 0e+00 | -1.52e-01, 2e-26 |
| Cifar-1000-1000 | -1.73e-01, 0e+00 | -2.91e-01, 0e+00 | -7.86e-02, 0e+00 | -4.47e-01, 0e+00 | Cifar-1000-1000 | -2.37e-02, 0e+00 | -2.87e-01, 0e+00 | -1.12e-01, 0e+00 | -4.54e-01, 0e+00 |
| randn-500-1000 | -4.91e-03, 0e+00 | -5.10e-03, 0e+00 | -4.77e-03, 0e+00 | -1.24e-02, 2e-14 | randn-500-1000 | -1.00e-02, 0e+00 | -9.90e-03, 0e+00 | -9.55e-03, 0e+00 | -2.11e-02, 2e-14 |

| data-m-n | ADMM (rnd) | PSM (rnd) | RADMM (rnd) | OBCD-R (id) | data-m-n | ADMM (rnd) | PSM (rnd) | RADMM (rnd) | OBCD-R (id) |
|---|---|---|---|---|---|---|---|---|---|
| | | $r = 40$, time limit=60 | | | | | $r = 80$, time limit=60 | | |
| w1a-2477-300 | -6.45e-02, 0e+00 | -1.07e-01, 0e+00 | -8.56e-02, 0e+00 | -3.00e-01, 7e-15 | w1a-2477-300 | -1.28e-01, 0e+00 | -1.70e-01, 0e+00 | -1.34e-01, 0e+00 | -3.90e-01, 1e-16 |
| TDT2-500-1000 | -3.50e-02, 0e+00 | -9.89e-02, 0e+00 | -3.57e-02, 0e+00 | -4.09e-01, 6e-15 | TDT2-500-1000 | -9.80e-02, 0e+00 | -4.97e-02, 0e+00 | -4.55e-02, 0e+00 | -4.49e-01, 2e-14 |
| 20News-8000-1000 | -1.92e-02, 0e+00 | -3.43e-02, 0e+00 | -1.11e-01, 0e+00 | -4.14e-01, 2e-14 | 20News-8000-1000 | -2.93e-02, 0e+00 | -3.04e-02, 0e+00 | -2.23e-02, 0e+00 | -4.47e-01, 3e-14 |
| sector-6412-1000 | -8.70e-02, 0e+00 | -2.38e-02, 0e+00 | -3.70e-02, 0e+00 | -2.25e-01, 4e-15 | sector-6412-1000 | -7.99e-02, 0e+00 | -3.82e-02, 0e+00 | -3.39e-02, 0e+00 | -2.96e-01, 5e-15 |
| E2006-2000-1000 | -8.36e-03, 0e+00 | -3.64e-01, 0e+00 | -2.68e-02, 0e+00 | -4.75e-01, 2e-14 | E2006-2000-1000 | -3.09e-03, 0e+00 | -3.31e-01, 0e+00 | -1.39e-01, 0e+00 | -4.89e-01, 2e-14 |
| MNIST-60000-784 | -2.09e-02, 0e+00 | -1.09e-01, 0e+00 | -4.67e-02, 0e+00 | -2.89e-01, 3e-14 | MNIST-60000-784 | -5.06e-02, 0e+00 | -9.95e-02, 0e+00 | -8.13e-02, 0e+00 | -3.03e-01, 3e-14 |
| Gisette-3000-1000 | -2.59e-02, 0e+00 | -2.63e-01, 0e+00 | -1.47e-01, 0e+00 | -3.69e-01, 6e-20 | Gisette-3000-1000 | -6.51e-02, 0e+00 | -2.64e-01, 0e+00 | -2.35e-01, 0e+00 | -3.56e-01, 0e+00 |
| CnnCaltech-3000-1000 | -2.03e-02, 0e+00 | -8.75e-02, 0e+00 | -4.74e-02, 0e+00 | -1.49e-01, 0e+00 | CnnCaltech-3000-1000 | -4.77e-02, 0e+00 | -1.02e-01, 0e+00 | -8.91e-02, 0e+00 | -1.61e-01, 0e+00 |
| Cifar-1000-1000 | -2.65e-02, 0e+00 | -3.25e-01, 0e+00 | -1.60e-01, 0e+00 | -4.43e-01, 0e+00 | Cifar-1000-1000 | -6.69e-02, 0e+00 | -3.21e-01, 0e+00 | -2.29e-01, 0e+00 | -4.24e-01, 0e+00 |
| randn-500-1000 | -2.01e-02, 0e+00 | -2.03e-02, 0e+00 | -2.00e-02, 0e+00 | -3.08e-02, 5e-16 | randn-500-1000 | -4.03e-02, 0e+00 | -4.02e-02, 0e+00 | -3.95e-02, 0e+00 | -5.31e-02, 3e-14 |

Table 3: Comparisons of objective values and the violation of the nonnegative constraints ($\| \min(\mathbf{0}, \mathbf{X})\|_\mathsf{F}$) for nonnegative PCA for all the compared methods. The $1^{st}$, $2^{nd}$, and $3^{rd}$ best results are colored with red, green and blue, respectively.

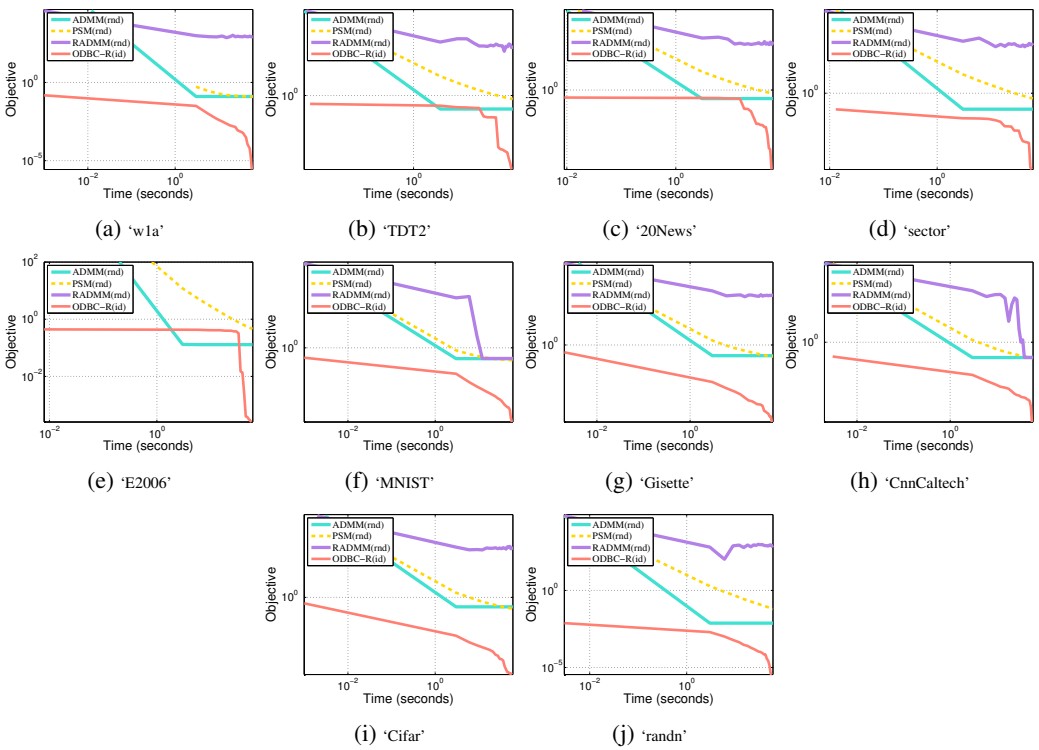

Figure 11: The convergence curve of the surrogate objective $f(\mathbf{X}) + 1000\| \min(\mathbf{0}, \mathbf{X})\|_\mathsf{F}$ with $\mathbf{X} \in \mathrm{St}(n, r)$ for solving the nonnegative PCA problem with $r = 10$.

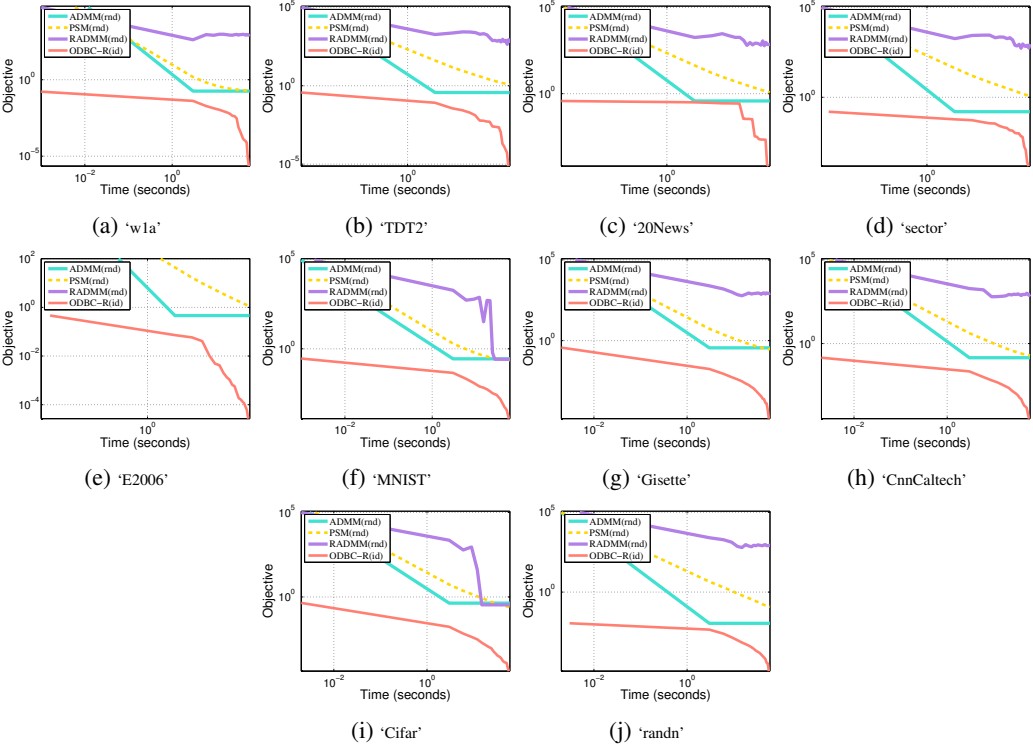

Figure 12: The convergence curve of the surrogate objective $f(\mathbf{X}) + 1000\| \min(\mathbf{0}, \mathbf{X})\|_\mathsf{F}$ with $\mathbf{X} \in \mathrm{St}(n, r)$ for solving the nonnegative PCA problem with $r = 20$.

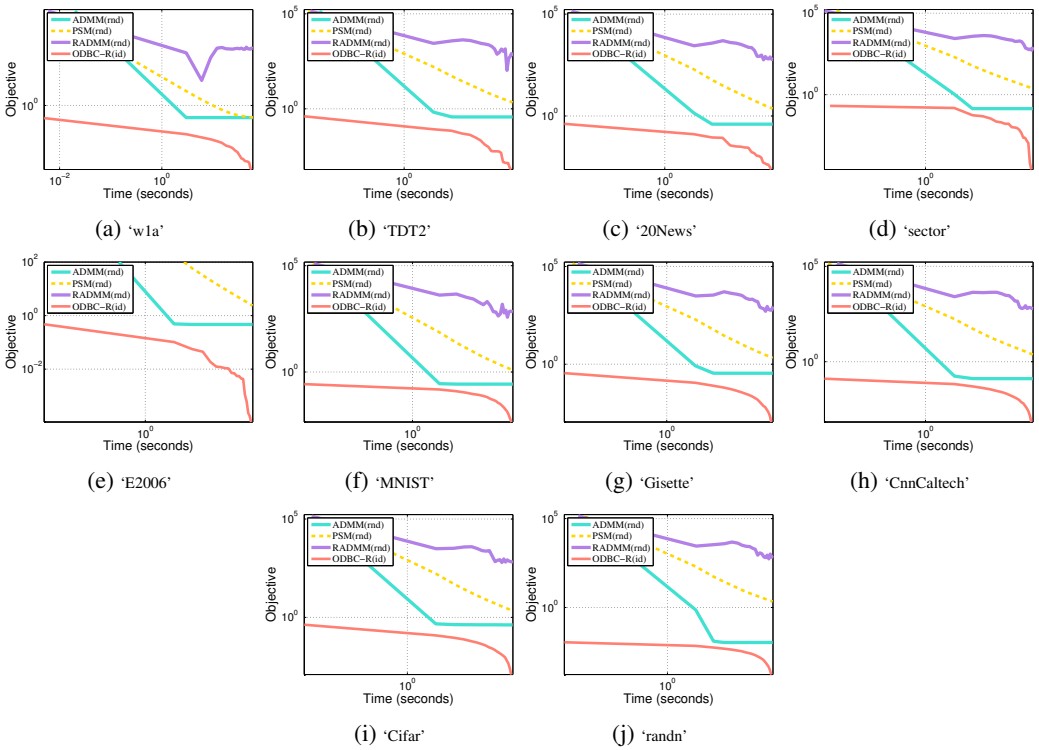

Figure 13: The convergence curve of the surrogate objective $f(\mathbf{X}) + 1000\|\min(\mathbf{0}, \mathbf{X})\|_\mathsf{F}$ with $\mathbf{X} \in \mathrm{St}(n, r)$ for solving the nonnegative PCA problem with $r = 40$.

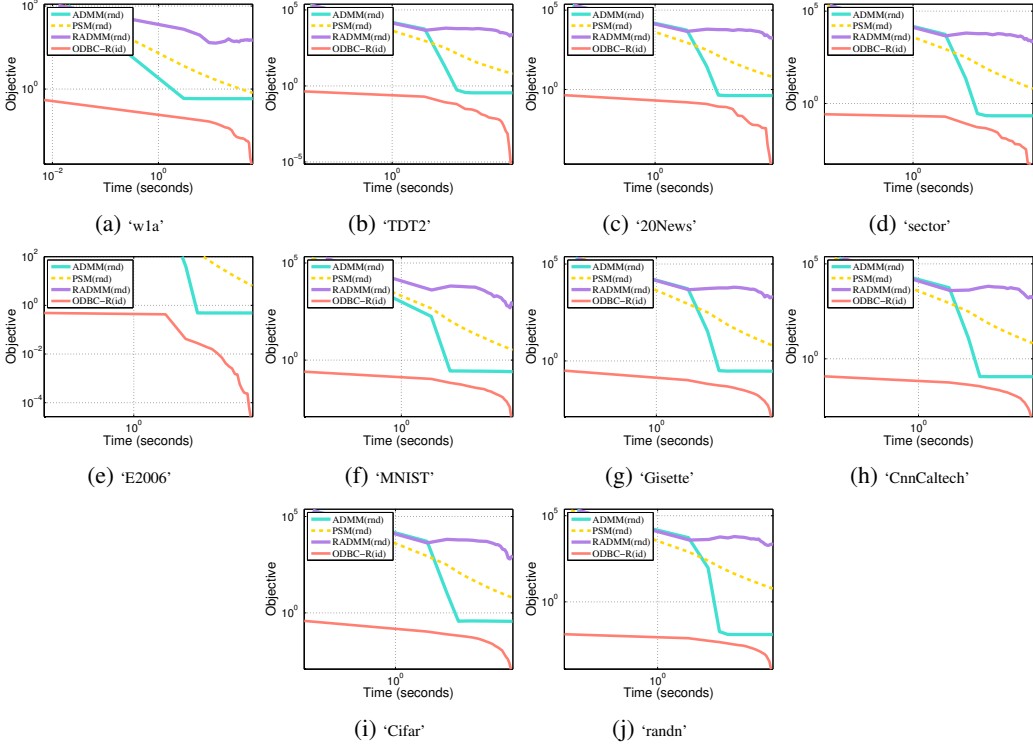

Figure 14: The convergence curve of the surrogate objective $f(\mathbf{X}) + 1000\|\min(\mathbf{0}, \mathbf{X})\|_\mathsf{F}$ with $\mathbf{X} \in \mathrm{St}(n, r)$ for solving the nonnegative PCA problem with $r = 80$.

