# OpenReview forum: "A Block Coordinate Descent Method for Nonsmooth Composite Optimization under Orthogonality Constraints"
_ICLR.cc/2026/Conference — ICLR 2026 Poster_

### Official Review · Reviewer_X2XH · 2025-10-19

**Soundness:** 3
**Presentation:** 3
**Contribution:** 3
**Rating:** 8
**Confidence:** 2

**Summary:**

A new approach called OBCD is proposed for minimizing the sum of a smooth and a nonsmooth convex functions over the nonconvex set of orthogonal matrices.

**Strengths:**

Optimization under orthogonality constraints is an important topic with many applications. The paper is well written. My expertise of the topic is limited, though. I know about projection methods, as there is a surge of interest for orthogonalization recently in the context of the Muon optimizer, see for instance Grishina et al. "Accelerating Newton-Schulz Iteration for Orthogonalization via Chebyshev-type Polynomials". The proposed method maintains feasibility throughout the process, which is a different approach than relying on approximate orthogonalization.

**Weaknesses:**

I don't see limitations.

**Questions:**

N/A

---

> ### Author Response · Authors · 2025-11-19
>
> We thank the reviewer for the encouraging feedback.

---

> ### Comment · Reviewer_X2XH · 2025-11-26
>
> After reading all reviews and rebuttals carefully, I confirm my positive evaluation and keep my score.

---

### Official Review · Reviewer_Jeff · 2025-10-27

**Soundness:** 3
**Presentation:** 2
**Contribution:** 3
**Rating:** 6
**Confidence:** 3

**Summary:**

The authors propose an algorithm for minimizing the sum of a smooth and nonsmooth function on the Stiefel manifold.
Relevant applications include sparse Principal Component Analysis (PCA) with l1 norm and l0 norm regularization.
The algorithm is a descent method, and features a block coordinate update, which requires computing the corresponding coordinate block of gradient, as opposed to the full gradient.
In theory, the algorithm converges in expectation to a relevant point with classical complexity (Th. 4.2), and under additional assumptions (including continuity of $h$, which rules the important l0 norm example and leaves l1 norm) the last iterate converges to a relevant point with classical complexity. We note that the notion of "relevant point" is slightly stronger than the usual notion of critical point (0 in limiting subdifferential).
In practice, the algorithm performs well on PCA with l0 regularization.

**Strengths:**

- the submission is clearly written,
- the problem is motivated by a relevant application (sparse PCA with l0 and l1 norm regularization),
- the algorithm is derived and formulated in a rigorous and clear way (up to one detail, discussed below),
- the theoretical analysis presents a notion of stationary points (BS$k$-points) that is stronger than the customary critical point (zero in subdifferential), and a proof that the algorithm converges to such points on average. I am quite familiar with composite optimization, but less so on block coordinate schemes so this idea may be common in that field.
- the algorithm combines known ideas (block coordinate descent, majorization minimization) suitably, and successfully applies these ideas to minimization with l0 norm regularization. As far as I know, this task is highly challenging, and few methods for it are available even in the unconstrained case.

**Weaknesses:**

Here is a list of points that prevent me from providing a better assessment to the submission.
- The nonsmooth function $h$ has strong, restrictive assumptions. This severely limits the applicability of the method.
  + Assumption (ii): $h$ is assumed to be coordinate separable, with the same expression for each coordinate.
  + Assumption (iii) on function $h$ is shown to be tractable in theory for three values of $h$ only (the $\ell1$ norm, the indicator of a polyhedron, and the $\ell0$ norm when $k=2$).
- l. 122-128: I disagree with part of the positioning relative with the literature:
  + contrary to what point (ii) implies, I think that the proposed method is not applicable to general nonsmooth composite problems. Indeed it relies on an assumption on the nonsmooth function, which is arguably restrictive (see point below) but certainly not valid for "general nonsmooth composite problems".
  + point (iii) asserts that it is a "limitation" for methods to be "infeasible", that is to generate iterates that are only asymptotically feasible. Yet, infeasible methods are accepted for solving large-scale optimization problems under general constraints (e.g. with the Augmented Lagrangian method), and Stiefel manifold constraints specifically (e.g. with the recent Landing methods, see next point).
  + point (iv), "they often lack rigorous convergence guarantees" should be more precise on the type of convergence guarantees that previous methods fail (for instance: convergence of average, last iterate, complexity guarantee).
- l. 78-107: Literature review misses one recent line of work of so-called "landing methods" for optimization on Stiefel manifolds: [1-4]. These methods also target large-scale minimization on the Stiefel manifold; the paper should discuss these methods.
- Algorithm 1: The algorithm seems to depend on a parameter $\alpha$ (l. 194, Lemma 2.3, Th. 4.2, among other occurrences), yet this parameter does not appear in the algorithm statement, and it is unclear what condition $\alpha$ should meet so that the convergence guarantees hold, and particularly so the sufficient decrease condition of Theorem 4.2(a).
- l. 374: the condition $\| \partial h(X) \|_{sp}<l_h$ is ill-posed: the "sp" norm of a set is not defined. Besides, this condition may appear more natural if it were implied by Lipschitz continuity of $h$.
- l. 392: the assumption that $F_i$ is a KL function is not discussed at all. It is thus unclear whether it holds for any of the application cases, and the three discussed $h$ functions.
- l. 395: Proposition 4.8, from previous work, is stated without providing a precise reference, that is, both the paper and statement references.
- l. 404 & 412: "the continuity assumption made in lemma 4.4" is unclear. To what condition does this sentence refer to exactly? Maybe writing an additional assumption for this would clarify the situation.
- Experiments: it is unclear whether the three baselines generate feasible iterates. In addition, these three baselines involve operator splitting. Such methods usually involve a reformulation of the problem with additional constraints, which are satisfied asymptotically only. Finally, it is not clear whether the three baselines have convergence guarantees on  the problem with $h = \| \cdot \|_{0}$, which is discontinuous; this aspect conditions the interpretation of the whole experimental section.

References:
- [1] Goyens, Absil & Feppon (2026) Geometric Design of the Tangent Term in Landing Algorithms for Orthogonality Constraints, Springer Nature Switzerland.
- [2] Ablin & Peyré (2022) Fast and Accurate Optimization on the Orthogonal Manifold without Retraction, PMLR.
- [3] Vary, Ablin & Gao et al. (2024) Optimization without Retraction on the Random Generalized Stiefel Manifold, PMLR.
- [4] Gao, Vary & Ablin et al. (2022) Optimization Flows Landing on the Stiefel Manifold$\star$, IFAC-PapersOnLine.


Minor points, that do not impact the assessment of the paper:
- l. 35: clarity would improve by mentioning explicitly that $f$ is assumed to be differentiable and $H$-smooth
- l. 39, footnote: Is $f$ assumed to have expression $1/2 \|X\|_{H}^{2}$ on the whole paper?
- l. 40 & 296: what are the definitions of $F$ "closed", and $F$ lower semicontinuous? How do they differ?
- l. 230: is the
- l. 305: The writing of Definition 3.3 is confusing: the sentence "Furthermore $\lambda \in [\partial F(X)]^\top X$" can read as an additional condition for a point $X$ to be critical, but the notion  of criticality does not involve $\lambda$.
- l. 313-318: what is the purpose of this paragraph?

**Questions:**

Any comment and detail on each of the listed weak points is welcome.

Below are some suggestions in the form of questions; any answer on the following questions is welcome, but no answer is also fine.
- connection to (variable metric) proximal gradient method? That would connect assumption (iii) to the notion of "prox-friendly" nonsmooth function $h$, standard in nonsmooth optimization. That may also help and connect
- Th. 3.6: globally optimal points are BS2-points. Are they also BSk points, for $2 \le k \le n-1$?
- Assumption (iii) is reminiscent of the  might appear more natural if connected to the proximal gradient operator.
- optimization methods on problems with l0-norm regularization usually face the combinatorial difficulty of the l0 norm in some way. I am surprised that it doesn't show in your analysis. Do you have some intuition to share on this? I am also surprised that the complexity of OBCD with parameter $k$ in Theorem 4.2 does not depend on $k$. Again, do you have some intuition to share on this?

---

> ### Author Response · Authors · 2025-11-19
>
> Thank you for your efforts in evaluating our manuscript.
>
> ---
>
> **Q1.** The nonsmooth function $h$ has strong, restrictive assumptions. This severely limits the applicability of the method.
>
> **A1.** This restriction is inherent to classical coordinate-descent–type methods (see, e.g., Tseng & Yun, 2009, Mathematical Programming). Nevertheless, our framework still covers an important class of problems, including smooth minimization under orthogonality constraints and sparse PCA with $\ell_0$-, $\ell_1$-, and capped $\ell_1$ penalties, so the proposed methods remain highly applicable in practice.
>
> ---
>
> **Q2.** l. 122-128: I disagree with part of the positioning relative with the literature: contrary to what point (ii) implies, I think that the proposed method is not applicable to general nonsmooth composite problems. Indeed it relies on an assumption on the nonsmooth function, which is arguably restrictive (see point below) but certainly not valid for "general nonsmooth composite problems".
>
> **A2.** We agree and have revised the text accordingly: *they do not accommodate coordinate-wise nonsmooth composite objectives*.
>
> ---
>
> **Q3.** point (iii) asserts that it is a "limitation" for methods to be "infeasible", that is to generate iterates that are only asymptotically feasible. Yet, infeasible methods are accepted ...
>
> **A3.** We agree that infeasible methods are widely used and often very effective (e.g., augmented Lagrangian and recent landing methods).
>
> We consider scenarios where each iterate is used immediately and orthogonality must be enforced at every step—such as online PCA or subspace tracking on the Stiefel manifold—so temporary constraint violations are undesirable. In such settings, maintaining feasibility at all iterations becomes a practical advantage.
>
> ---
>
> **Q4.** point (iv), "they often lack rigorous convergence guarantees" should be more precise on the type of convergence guarantees that previous methods fail (for instance: convergence of average, last iterate, complexity guarantee).
>
> **A4.** We have revised it to: *they often lack rigorous last-iterate convergence guarantees*.
>
> ---
>
> **Q5.** l. 78-107: Literature review misses one recent line of work of so-called "landing methods" for optimization on Stiefel manifolds: [1-4]. These methods also target large-scale minimization on the Stiefel manifold; the paper should discuss these methods.
>
> **A5.** We thank the reviewer for pointing out this important line of work. In the revised manuscript, we have added a dedicated discussion of landing methods in the related work section.
>
> ---
>
> **Q6.** Algorithm 1: The algorithm seems to depend on a parameter $\alpha$.
>
> **A6.** We have explicitly included the proximal parameter in Algorithm 1 as: "proximal parameter $\alpha>0$".
>
> Our analysis only requires $\alpha>0$, and we set $\alpha=10^{-6}$ in all experiments.
>
> ---
>
> **Q7.** l. 374: the condition $| \partial h(X) |_{sp}<l_h$ is ill-posed: the "sp" norm of a set is not defined. Besides, this condition may appear more natural if it were implied by Lipschitz continuity of $h$.
>
> **A7.** We have revised it to:
>
> "Assume that $F(\cdot)$ is $C_F$-Lipschitz continuous on $St(n,r)$, i.e., $\|\mathbf{G}\|_{fro} \leq C_F$ for all $X\in St(n,r)$ and all $\mathbf{G}\in \partial F(X)$."
>
> ---
>
> **Q8.** l. 392: the assumption that $F_i$ is a KL function is not discussed at all. It is thus unclear whether it holds for any of the application cases, and the three discussed $h$ functions.
>
> **A8.** We have added the following clarification:
>
> "Semi-algebraic functions constitute a broad class of KL functions, including real polynomials, norm functions $\|x\|_p$ with $p \ge 0$, rank functions, and indicator functions of sets such as the Stiefel manifold and the positive semidefinite cone \cite{Attouch2010}."
>
> The three choices of $h$ in our applications are semi-algebraic, so each $F$ is indeed a KL function.
>
> ---
>
> **Q9.** l. 395: Proposition 4.8, from previous work, is stated without providing a precise reference, that is, both the paper and statement references.
>
> **A9.** We have clarified the origin of Proposition 4.8 by explicitly citing the source papers.
>
> ---
>
> **Q10.** l. 404 & 412: "the continuity assumption made in lemma 4.4" is unclear.
>
> **A10.** We have revised it to:
>
> Assume that $F(\cdot)$ is $C_F$-Lipschitz continuous on $St(n,r)$, i.e., $\|\mathbf{G}\|_{fro} \leq C_F$ for all $X\in St(n,r)$ and all $\mathbf{G}\in \partial F(X)$.
>
>
> ---

---

> ### Author Response · Authors · 2025-11-19
>
> **Q11.** Experiments: it is unclear whether the three baselines generate feasible iterates. In addition, these three baselines involve operator splitting. Such methods usually involve a reformulation of the problem with additional constraints, which are satisfied asymptotically only ..
>
> **A11.**
>
> 1. All three baselines handle the orthogonality constraint and the $\ell_0$ term via an operator-splitting strategy. We therefore evaluate the objective (F(X) = f(X) + h(X)) using the current feasible iterate (that satisfies the orthogonality constraint).
>
>
> 2. Since the exact $\ell_0$ norm is discontinuous, we use a numerical practice and count entries with $|X_{ij}| > 10^{-6}$ as nonzeros when computing $h(X)$, which yields stable results for all methods.
>
> ---
>
> **Q12.** l. 35: clarity would improve by mentioning explicitly that $f$ is assumed to be differentiable and $H$-smooth
>
> **A12.** We also assume that $f(\cdot)$ is differentiable in the revision.
>
>
> ---
>
> **Q13.** l. 39, footnote: Is $f$ assumed to have expression $1/2 |X|_{H}^{2}$ on the whole paper?
>
> **A13.** We assume that $f(\cdot)$ is H-smooth, without loss of generality.
>
>
> ---
>
> **Q14.** l. 40 & 296: what are the definitions of $F$ "closed", and $F$ lower semicontinuous? How do they differ?
>
> **A14.** In our original text, "closed" and "lower semicontinuous" were used redundantly; what we actually need is that "F" is proper and lower semicontinuous.
>
> ---
>
>
> **Q15.** l. 305: The writing of Definition 3.3 is confusing: the sentence "Furthermore $\lambda \in [\partial F(X)]^\top X$" can read as an additional condition for a point $X$ to be critical, but the notion of criticality does not involve $\lambda$.
>
> **A15.** We have changed it to: "Moreover, the corresponding multiplier satisfies $\mathbf{\Lambda}\in [\partial F(\check{X})]^T\check{X}$."
>
> ---
>
> **Q16.** l. 313-318: what is the purpose of this paragraph?
>
> **A16.** This paragraph introduces "Optimality Conditions for the Subproblems" that is later used to prove the optimality hierarchy in Theorem 3.6 and the Riemannian subgradient lower bound in Lemma 4.4(a). We now state this role explicitly in the revised manuscript.
>
>
> ---
>
> **Q17.** connection to (variable metric) proximal gradient method? That would connect assumption (iii) to the notion of "prox-friendly" nonsmooth function $h$, standard in nonsmooth optimization. That may also help and connect
>
>
> **A17.** We have added the following clarification to the revised manuscript:
>
> "This assumption is analogous to the ``prox-friendly'' condition in (variable-metric) proximal gradient methods, but instead of a standard proximal operator for \textit{a single nonsmooth term} in the \textit{full} space, our subproblem jointly handles \textit{two nonsmooth components} (the function $h(\cdot)$ and the orthogonality constraint) in the low-dimensional $k\times k$ space."
>
> ---
>
> **Q18.** Th. 3.6: globally optimal points are BS2-points. Are they also BSk points, for $2 \le k \le n-1$?
>
>
> **A18.**
>
> 1. The answer is affirmative.
>
> 2. In the revised version, we extend Theorem 3.1 and add Corollary 3.2, establishing that the representation result also holds for all $k > 2$.
>
> 3. By applying the same reasoning used in the original Theorem 3.7 (previously for $k=2$), we have shown that every globally optimal point is a $BS_k$ point for all $2 \le k \le n$.
>
> ---
>
> **Q19.** Assumption (iii) is reminiscent of the might appear more natural if connected to the proximal gradient operator.
>
>
> **A19.** We have revised the text and added some clarification to explicitly connect Assumption (iii) to the "prox-friendly" condition.
>
> ---
>
> **Q20.** optimization methods on problems with l0-norm regularization usually face the combinatorial difficulty of the l0 norm in some way. I am surprised that it doesn't show in your analysis. Do you have some intuition to share on this? I am also surprised that the complexity of OBCD with parameter $k$ in Theorem 4.2 does not depend on $k$. Again, do you have some intuition to share on this?
>
> **A20.** The combinatorial difficulty of the $\ell_0$ term is absorbed into each $k\times k$ subproblem, so Theorem 4.2 counts only the number of such subproblem calls needed to reach an $\varepsilon$-$BS_k$ point. The dependence on $k$ is implicit: it appears in the definition of an $\varepsilon$-$BS_k$ point,
> $$\frac{1}{C_n^k} \sum_{i=1}^{C_n^k} dist (I_k,\arg\min_{V} K(V; \ddot{X}, B_i) )^2 \leq \epsilon,$$
> which averages over $C_n^k$ blocks, and in the per-iteration cost of solving the $k\times k$ subproblems in the break-point search procedure, even though $k$ does not appear explicitly in the iteration bound.
>
> ---

---

> > ### Comment · Reviewer_Jeff · 2025-11-26
> >
> > I thank the authors for the detailed answers and corresponding updates. Going over the math again, I have one last question.
> >
> > The KL property is a local assumption, with neighborhoods on the space of iterates $X \in \Upsilon$ and function values both ($F(X') \in (F(X^\infty), F(X^\infty + \eta)$. However, Th. 4.10(a) states an inequality valid for all $t$, including at times when the iterates and the function values do not necessarily lie in the neighborhoods that guarantee the KL inequality. How do you manage to use the KL property at all iterates of the sequence? Specifically, why does equation (59) hold for any $t$? If not, how are the later results (Th. 4.10(b), 4.11) affected?
> >
> > In view of the subsequential convergence provided by Th. 4.6, I would think that (i) there exists a (random) $t_0$ at which both the iterate and function value lie in the proper neighborhood, and that (ii) iterates remain in that neighborhood for $t \ge t_0$, so that the results still mostly apply.

---

> ### Author Response · Authors · 2025-11-26
> **Response to the comment on the KL property**
>
> We thank the reviewer for the insightful comment.
>
> Indeed, the KL property is *local* and the KL inequality holds only when the iterate and its function value lie in a neighborhood of a limit point.
>
> In our analysis, we in fact use the KL inequality only for **sufficiently large** indices. By Theorems 4.2 and 4.6, there exists a subsequence $X^{t}$ converging to a limit point $X^\infty$ with $F_{\iota}(X^{t}) \to F_{\iota}(X^\infty)$. By the local KL property at $X^\infty$, there exist a neighborhood $\Upsilon$ of $X^{\infty}$ and an index $t_\star$ such that
> $$ X^t \in \Upsilon, \ F_{\iota}(X^t) \in (F_{\iota}(X^\infty), F_{\iota}(X^\infty)+\eta),\text{for all}~t \ge t_\star.$$
> Therefore, Equation (59) should be read as
> $$\frac{1}{\varphi' (F_{\iota}(X^t) - F_{\iota}(X^{\infty}) )} \leq \text{dist}(0,\partial F_{\iota}(X^t)),\text{for all } t \ge t_\star.$$
>
> The later results, Theorems 4.10(b) and 4.11, rely only on this **tail** inequality, and the statements and rates remain valid.
>
> We have revised the text around Theorem 4.10 to clearly state the “large enough $t_\star$” requirement and avoid the impression that the KL inequality is assumed to hold for all $t$.

---

### Official Review · Reviewer_TMgr · 2025-10-31

**Soundness:** 3
**Presentation:** 2
**Contribution:** 2
**Rating:** 6
**Confidence:** 3

**Summary:**

The authors study the problem of minimizing a nonconvex nonsmooth function $F(X)$ over the space of $n \times r$ orthogonal matrices (the Stiefel manifold $St(n,r)$). This is a challenging class of problems with wide applications. Common approaches for such problems include projection-based methods, Riemannian methods using tangent-space surrogates and retractions, or Block Majorization-Minimization (BMM) methods that iteratively minimize a surrogate directly on the manifold.

The authors propose an approach, **OBCD**, which falls under the BCD/BMM umbrella but with a novel and distinct design. Instead of updating a manifold block $X_k$ directly, the method subsamples $k$ rows and finds a small $k \times k$ orthogonal matrix $V$ that *transforms* this block. This "row-wise" update ($X^{t+1}(\mathcal{B},:) \leftarrow \overline{V}^{t}X^{t}(\mathcal{B},:)$) is a key contribution, as it is inherently feasible and avoids the standard tangent-space/retraction machinery.

While the general BCD/BMM idea is known, this paper's novelty lies in the specifics of its framework for the Stiefel manifold:

1.  It defines a new optimality condition, the **"block-k stationary point" ($BS_k$-point)**.  Theorem 3.6 shows that this condition is **stronger** than the standard critical point condition. The authors justify this by showing their $k=2$ solver uses both rotations and reflections, allowing it to escape suboptimal points.

2.  It provides a **constructive and exact solver** for its nonsmooth subproblem. Appendix B introduces a novel "Breakpoint Searching Method (BSM)" that finds the *exact global solution* for the $k=2$ subproblem with $l_0$, $l_1$, or non-negativity regularizers. This is a non-trivial technical result that provides a solid foundation for the algorithm.

On this theoretical foundation, the authors derive a comprehensive convergence analysis, including an $\mathcal{O}(1/\epsilon)$ iteration complexity for an $\epsilon$-$BS_k$-point (Theorem 4.2) and a full non-ergodic (last-iterate) convergence analysis under the KL property (Theorem 4.10). The presentation is clear, with assumptions formally stated, the algorithm well-defined, and the theoretical claims rigorously established.

**Strengths:**

The authors study the problem of minimizing a nonconvex nonsmooth function $F(X)$ over the space of $n \times r$ orthogonal matrices (the Stiefel manifold $St(n,r)$). This is a challenging class of problems with wide applications. Common approaches for such problems include projection-based methods, Riemannian methods using tangent-space surrogates and retractions, or Block Majorization-Minimization (BMM) methods that iteratively minimize a surrogate directly on the manifold.

The authors propose an approach, **OBCD**, which falls under the BCD/BMM umbrella but with a novel and distinct design. Instead of updating a manifold block $X_k$ directly, the method subsamples $k$ rows and finds a small $k \times k$ orthogonal matrix $V$ that *transforms* this block. This "row-wise" update ($X^{t+1}(\mathcal{B},:) \leftarrow \overline{V}^{t}X^{t}(\mathcal{B},:)$) is a key contribution, as it is inherently feasible and avoids the standard tangent-space/retraction machinery.

While the general BCD/BMM idea is known, this paper's novelty lies in the specifics of its framework for the Stiefel manifold:

1.  It defines a new optimality condition, the **"block-k stationary point" ($BS_k$-point)**. This is a significant contribution, as Theorem 3.6 proves this condition is **provably stronger** than the standard critical point condition. The authors justify this by showing their $k=2$ solver uses both rotations and reflections, allowing it to escape suboptimal points.

2.  It provides a **constructive and exact solver** for its nonsmooth subproblem. Appendix B introduces a novel "Breakpoint Searching Method (BSM)" that finds the *exact global solution* for the $k=2$ subproblem with $l_0$, $l_1$, or non-negativity regularizers. This is a non-trivial technical result that provides a solid foundation for the algorithm.

On this theoretical foundation, the authors derive a comprehensive convergence analysis, including an $\mathcal{O}(1/\epsilon)$ iteration complexity for an $\epsilon$-$BS_k$-point (Theorem 4.2) and a full non-ergodic (last-iterate) convergence analysis under the KL property (Theorem 4.10). The presentation is clear, with assumptions formally stated, the algorithm well-defined, and the theoretical claims rigorously established.

**Weaknesses:**

### 1. Unclear Practicality for the `k > 2` Case

A primary weakness is the gap between the well-analyzed $k=2$ case and the general $k > 2$ case. The paper's core assumption (Asm-iii) is that the $k \times k$ nonsmooth subproblem can be solved "exactly and efficiently." The authors provide an impressive, detailed proof of this for $k=2$ using their novel Breakpoint Searching Method (Appendix B).

However, for $k > 2$, this assumption is highly questionable. Solving a general $k \times k$ nonsmooth composite problem over the Stiefel manifold $St(k,k)$ is not a trivial task, and the paper provides no algorithm or justification for it.

The paper *does* offer a fallback in Algorithm 1, suggesting one can "Alternatively, find a local solution $\overline{V}^t$ such that $\mathcal{K}(\overline{V}^{t};X^{t},B)\le\mathcal{K}(I_{k};X^{t},B)$". But the paper provides no discussion on *how* to find such a local solution. For a general nonsmooth, nonconvex subproblem, even finding a point that guarantees this simple descent from the identity matrix ($I_k$) is a non-trivial problem in itself. Without a proposed method, the practical application of OBCD for block sizes $k > 2$ remains unclear.

### 2. Overstated Novelty Claim

In the "Summary" of the related work (Section 1.2), the paper claims: "To our knowledge, this represents the first application of BCD methods to solve nonsmooth composite optimization problems under orthogonality constraints...". This claim appears to be incorrect. The paper's own literature review (Section 1.2, under "Minimizing Nonsmooth Functions...") cites existing work on "Block Majorization Minimization (BMM) on Riemannian manifolds," which directly addresses nonsmooth problems on manifolds, including the Stiefel manifold. This contradiction in the paper's own text weakens the positioning of its contribution. Rather than claiming to be the first of its kind, the authors could emphasize the tailored approach for solving problems on Stiefel manifold efficiently.

### 3. Mismatch Between Theory and Experiments

There is a disconnect between the theoretical contributions and the experimental validation. The paper provides a strong theoretical justification (in Appendix B) for handling $l_0$, $l_1$, and non-negativity constraints. However, the experiments in Section 5 are conducted *only* on $L_0$-norm-based SPCA. This is a missed opportunity to not validate the new, non-trivial solvers for the $l_1$ and non-negativity constraint problems, which would have made the experimental section much more comprehensive and demonstrated the full power of the proposed subproblem solvers.

### Minor comments

While the paper's literature review is adequate, the authors may consider including the following recent paper on Euclidean BMM as part of the BCD section:

Hanbaek Lyu and Yuchen Li, “Block majorization-minimization with diminishing radius for constrained nonconvex optimization.”  SIAM Journal on Optimization, Vol. 35, Iss. 2 (2025)

**Questions:**

1. The authors mention in L120 that [Gao et al. 2019] studies a similar problem with columewise updates, whereas the proposed method is rowwise. Besides these algorithmic design choices, what are the important differences? Pros/cons on the computational cost? Theoretical property?

2.  The definition of the "block-k stationary point" ($BS_k$-point) (Definition 3.5) is based on $I_k$ being the *global minimizer* of the subproblem. If one only finds a local solution $\overline{V}^t \neq I_k$ that satisfies the descent condition (as suggested in Algorithm 1) and the algorithm converges, does the limit point have any meaningful theoretical properties? Does the hierarchy in Theorem 3.6 still hold, or is the $BS_k$-point definition fundamentally tied to the (impractical) global solution of the subproblem for $k > 2$?

---

> ### Author Response · Authors · 2025-11-19
>
> Thank you for the reviewer's valuable comments and detailed suggestions. Below we reply point‑by‑point, using the reviewer’s wording for clarity.
>
> ---
>
> **Q1.** Unclear Practicality for the k > 2 Case .. The paper does offer a fallback in Algorithm 1, suggesting one can "Alternatively, find a local solution $\overline{V}^t$ such that $\mathcal{K}(\overline{V}^{t};X^{t},B)\le\mathcal{K}(I_{k};X^{t},B)$" .. finding a point that guarantees this simple descent from the identity matrix ($I_k$) is a non-trivial problem in itself.
>
>
> **A1.**  In the case $k = 2$, one can efficiently compute $\overline{V}^t$ satisfying
> $$\mathcal{K}(\overline{V}^{t};X^{t},B)\le\mathcal{K}(I_{k};X^{t},B),\ \overline{V}^{t} \in 2 \times 2.$$
> The same idea extends naturally to the general case $k > 2$. Specifically, we adopt a coordinate-descent–type strategy on the $k \times k$ subproblem (NOT the original $n\times r$ problem), where each iteration updates a $2 \times 2$ block using the same procedure as in the $k=2$ case. By randomly updating these $2 \times 2$ blocks, we obtain a **nested scheme** that monotonically decreases $\mathcal{K}(\cdot;X^t,B)$ and yields
> $$\mathcal{K}(\overline{V}^{t};X^{t},B)\le\mathcal{K}(I_{k};X^{t},B),\ \overline{V}^{t} \in k \times k,\ \text{with}\ k>2$$
> for the resulting $\overline{V}^t$. This block-coordinate strategy preserves the practical appeal of the $k=2$ case, since each step only requires solving a low-dimensional $2 \times 2$ subproblem.
>
> Finally, we note that when $h = 0$, the subproblem in our OBCD algorithm reduces to computing a small $k \times k$ SVD, which is a standard and highly efficient operation in practice.
>
>
> ---
>
> **A2.** Overstated Novelty Claim. The paper claims: "To our knowledge, this represents the first application of BCD methods to solve nonsmooth composite optimization problems under orthogonality constraints...".
>
> **Q2.** Thanks for the suggestion. We have changed it to: "our methods overcome these limitations by using a tailored block coordinate descent framework for efficient composite optimization on the Stiefel manifold, with strong optimality and convergence guarantees."
>
> ---
>
> **Q3.** Mismatch Between Theory and Experiments .. the experiments in Section 5 are conducted only on $L_0$-norm-based SPCA. This is a missed opportunity to not validate the new, non-trivial solvers for the $l_1$ and non-negativity constraint problems, which would have made the experimental section much more comprehensive and demonstrated the full power of the proposed subproblem solvers.
>
> **A3.**  We have included the experiments on L0-SPCA, L1-SPCA, and nonnegative PCA in Appendix Section G.
>
> The corresponding code for all three variants is now included in the supplementary material for verification and reproduction.
>
>
> ---
>
> **Q4.** Minor comments: Literature review.
>
> **A4.** We have incorporated the suggested reference into the revised manuscript.
>
>
> ---
>
> **Q5.** The authors mention in L120 that [Gao et al. 2019] .. what are the important differences? Pros/cons on the computational cost? Theoretical property?
>
> **A5.**
>
> 1. Gao et al. (2019) handle only **smooth** objectives and use an **infeasible** scheme that enforces the Stiefel constraint only in the limit, with mainly critical-point guarantees and local **Q-linear convergence** under additional assumptions.
>
> 2. Our method, by contrast, treats general **nonsmooth** composite problems, maintains **feasibility** at every iterate, and enjoys **non-ergodic (last-iterate) convergence guarantees**, while operating on small subproblems with comparable per-iteration cost.
>
> ---
>
> **Q6.** The definition of the "block-k stationary point" ($BS_k$-point) (Definition 3.5) is based on $I_k$ being the global minimizer of the subproblem.. Does the limit point have any meaningful theoretical properties? Does the hierarchy in Theorem 3.6 still hold, or is the $BS_k$-point definition fundamentally tied to the (impractical) global solution of the subproblem for $k > 2$?
>
> **A6.**
>
> 1. If each subproblem is solved so as to satisfy a descent condition, namely $\mathcal{K}(\overline{V}^{t};X^{t},B)\le\mathcal{K}(I_{k};X^{t},B)$, then any limit point of the iterates remains a critical point of the original problem. In this setting, the hierarchy in Theorem 3.6 no longer strictly holds; the resulting "local block-𝑘 stationary’’ notion essentially coincides with the standard critical point.
>
> 2. The original $𝐵𝑆_𝑘$-point relies on global solutions of the $k\times k$ subproblems. Allowing local solutions offers a practical relaxation that drops the stronger optimality requirement yet maintains critical-point guarantees.

---

### Official Review · Reviewer_ufBN · 2025-11-05

**Soundness:** 3
**Presentation:** 2
**Contribution:** 3
**Rating:** 6
**Confidence:** 4

**Summary:**

The paper proposes a block coordinate descent (OBCD) algorithm for nonsmooth composite optimization on the Stiefel manifold. The method updates $k$ blocks at each iteration while preserving orthogonality, and achieves convergence to a stationary point under suitable assumptions. The authors also design an exact solver for the $k=2$ subproblem when the nonsmooth term $h$ is coordinate-wise separable, by reducing it to a one-dimensional breakpoint-search problem. Theoretical results include global convergence and iteration complexity bounds, and experiments on sparse PCA show promising performance.

**Strengths:**

* They propose a first BCD method for solving nonsmooth composite optimization problems under orthogonality constraints.
* The breakpoint-search solver for the $k=2$ subproblem is elegant and provides an exact and efficient solution when $h$ is separable.
* The experimental results on sparse PCA are convincing.

**Weaknesses:**

* (Assumption 3 (exact subproblem solution).) This assumption is quite restrictive, as exact solutions are only possible when $h$ is coordinate-wise separable (e.g., $\ell_0$ or $\ell_1$ penalties). For general non-separable regularizers such as the nuclear norm, this assumption may not hold. Could the authors discuss (i) whether the framework can be extended to non-separable $h$, and (ii) whether their convergence analysis remains valid if each subproblem is solved only approximately or to a stationary point? A relaxed ``inexact subproblem'' assumption might make the method more generally applicable.
* (Bounded subgradient assumption (Lemma 4.4).) I think the assumption $||\partial h(X)||_{sp} \le \ell_h$ appears problematic for $h(X)=||X||_0$ (please correct me if I am wrong). However, you use the $\ell_0$-norm in your experiments. It would be better justify this assumption.
* (Relation to prior work ([1]).) I think problem (1) is a special case of [1] when $H=L_f \mathcal{I}_{nr}$. It would strengthen the contribution to explain clearly how this work differs from and improves upon [1].
* (Experiment) Would the authors also include experimental comparisons with the method proposed in [1]? Such a comparison would better demonstrate the advantages of the proposed OBCD algorithm.

[1] Cheung, Andy Yat-Ming, et al. "Randomized Submanifold Subgradient Method for Optimization over Stiefel Manifolds." arXiv preprint arXiv:2409.01770 (2024).

**Questions:**

Please see above.

---

> ### Author Response · Authors · 2025-11-18
>
> We sincerely appreciate the reviewer’s thoughtful comments and suggestions.
>
> ---
>
> **Q1.** (Assumption 3 (exact subproblem solution).) This assumption is quite restrictive, as exact solutions are only possible when $h$ is coordinate-wise separable...(i) whether the framework can be extended to non-separable $h$.
>
> **A1.** Our coordinate descent–type method indeed cannot directly handle coordinate-wise (or, more precisely, row-wise) non-separable regularizers. This limitation is inherent to classical coordinate descent methods (see Tseng & Yun, 2009, Mathematical Programming).
>
> Given that smooth minimization under orthogonality constraints and sparse PCA with $\ell_0$-, $\ell_1$-, and capped-$\ell_1$ penalties already cover many important applications, we develop a lightweight **feasible** coordinate-descent solver specifically tailored to this setting.
>
> ---
>
> **Q2.** Could the authors discuss (ii) whether their convergence analysis remains valid if each subproblem is solved only approximately or to a stationary point? A relaxed ``inexact subproblem'' assumption might make the method more generally applicable.
>
> **A2.** Our convergence analysis remains valid under inexact subproblem solutions, provided the approximate solution $\bar{V}^t$ satisfies the mild improvement condition
>
> $$\mathcal{K}(\bar{V}^t;X^t,B) \leq \mathcal{K}(I_k;X^t,B).$$
>
> This condition simply requires that the chosen update does not increase the objective relative to the current iterate. Under this requirement, the sufficient descent condition still holds:
>
> $$\tfrac{\alpha}{2}||X^{t+1} - X^t||_F^2 \leq \tfrac{\alpha}{2} ||\bar{V}^t - I_k||_F^2 \leq F(X^t) - F(X^{t+1})$$
>
> and all subsequent convergence arguments remain unchanged.
>
> ---
>
> **Q3.** (Bounded subgradient assumption (Lemma 4.4).) I think the assumption $||\partial h(X)||_{sp} \le \ell_h$ appears problematic for $h(X)=||X||_0$ (please correct me if I am wrong). However, you use the $\ell_0$-norm in your experiments.
>
> **A3.**
>
> 1. The subgradient of the $\ell_0$-norm is not necessarily bounded, so the assumption $|\partial h(X)|_{sp} \le \ell_h$ does not hold when $h(X) = |X|_0$. Nevertheless, for $\ell_0$-regularized PCA, OBCD still attains the standard $O(\varepsilon^{-1})$ iteration complexity for reaching an $\varepsilon$-stationary point, though without the additional non-ergodic rate guarantee.
>
> 2. The bounded-subgradient condition holds for regularizers such as the $\ell_1$-norm and the capped-$\ell_1$ penalty. Consequently, for $\ell_1$-regularized PCA, OBCD enjoys non-ergodic convergence. For completeness, we have **included $\ell_1$-regularized PCA experiments in Appendix Section~G.**
>
> ---
>
> **Q4.** (Relation to prior work ([1]) I think problem (1) is a special case of [1] when $H=L_f \mathcal{I}_{nr}$. It would strengthen the contribution to explain clearly how this work differs from and improves upon [1].
>
> [1] Cheung, Andy Yat-Ming, et al. "Randomized Submanifold Subgradient Method for Optimization over Stiefel Manifolds." arXiv preprint arXiv:2409.01770 (2024).
>
> **A4.** The optimization models considered in [1] do not fully subsume our problems.
>
>
> 1. For $\ell_0$-regularized PCA, the objective is not weakly convex, violating a key assumption in [1]. Consequently, the RSSM algorithm in [1] cannot be applied, as one cannot reliably choose subgradients for the $\ell_0$-norm that both guarantee meaningful descent and promote sparsity.
>
> 2. For $\ell_1$-regularized PCA, [1] is applicable. We have included comparisons with RSSM (as well as Linearized ADMM, ADMM, Riemannian subgradient methods, and adaptive semi-smooth Newton methods) in Appendix Section G. Our method OBCD consistently achieves solutions of higher quality, which we attribute to the stronger stationarity achieved by OBCD.
>
> Key methodological differences between OBCD and RSSM include:
>
> 1. Update rule: Our method updates several rows per iteration, whereas [1] updates several columns.
>
> 2. Stationarity guarantee: We obtain block-k stationary points, which are stronger than the stationarity conditions guaranteed by RSSM.
>
> 3. Convergence rate: We establish non-ergodic convergence rates, while such guarantees for RSSM in deterministic settings are unknown.
>
> 4. Handling nonsmooth and discontinuous terms: Our method explicitly leverages the structural properties of nonsmooth (and even discontinuous) regularizers, leading to different algorithmic behavior and convergence properties compared to [1].
>
>
> ---
>
> **Q5.** (Experiment) Would the authors also include experimental comparisons with the method proposed in [1]?
>
> **A5.** Because RSSM [1] cannot be applied to $\ell_0$-PCA, we added comparisons on $\ell_1$-PCA in the revised manuscript, where RSSM is applicable. The baselines include RSSM, linearized ADMM, ADMM, Riemannian subgradient methods, and adaptive semi-smooth Newton methods. Appendix Section G shows that our method consistently produces superior solutions.

---

> > ### Comment · Reviewer_ufBN · 2025-11-22
> >
> > I appreciate the authors for the response, which addresses some of my concerns. I will raise the score.

---

### Meta-Review · Area_Chair_3AGu · 2026-01-06

**Summary:**

The authors propose OBCD method to solve nonsmooth optimization problem with orthogonality constraints, which is very hard because the manifold is non-convex. They update small rows of matrix using a new "Breakpoint Searching Method" that stay on the manifold always and find better stationary points than usual methods.

The reviewers were mostly positive about this paper but highlighted few concerns, the major one are listed below:

1. Many reviewers noticed the assumption for $h$ is very restrictive because it must be coordinate-wise separable.
It means the method cannot do things like nuclear norm.
2. One reviewer is worried that solving subproblem for more than 2 rows is too difficult and the paper doesn't show how.
3. Some reviewers feel the authors claim they are the first to do BCD for this problem, but other papers like "landing methods" or "BMM" already exist.
4. Some reviewers had deep questions about KL property and if it works for all iterates.

**Reviewer Concerns:**

# Addressed Concerns
1. Reviewer ask how KL inequality works for all iterates since it is local property.
 The authors clarify that they only need it for "sufficiently large" indices in the tail of the sequence. It seems reviewer has been satisfied with this math explanation.
2. One reviewer expressed that the problem might be same as [1] and want more experiments.  Authors add new experiments for $\ell_1$-PCA and show that OBCD is better because it find stronger stationary points. The reviewer was happy with the rebuttal.
3.  the claim of being "first" is not true.  Authors change the text to say they are "tailored" framework instead of first ever.
4. it can be hard to solve subproblem for big $k$. Authors explain they can use "nested scheme" where they solve the subproblem by updating $2 \times 2$ blocks inside.

# Outstanding Concerns
*  This concern that nonsmooth term $h$ is restrictive is still somewhat there.  The authors admit that their method "cannot directly handle coordinate-wise... non-separable regularizers." While they argue many applications use separable terms, the method is still not "general" for all nonsmooth problems.
*  Even though authors explain the nested scheme, some reviewers feel there is "gap between the well-analyzed $k=2$ case and the general $k>2$ case." The paper focus mostly on $k=2$ and doesn't provide the same deep analysis or experiments for very large $k$, so the practical advantage for large blocks is still not fully proven in the main text.

**Reviewer Scores:**

*  Reviewer ufBN (Initial: 6): This reviewer already said they "will raise the score" after the authors added new experiments and explained the difference from the RSSM paper.
*  Reviewer TMgr (Initial: 6): Even though the authors explained the "nested scheme" for the $k > 2$ case, I feel the reviewer would not be inclined his score and would keep 6.
*  Reviewer Jeff (Initial: 6): This reviewer was very active in the discussion, especially about the KL property and the "tail" inequality. After the authors clarified some confusions, I feel that he would increase his rating.
*  Reviewer X2XH (Initial: 8): This reviewer did not have many concerns to start with and they have already been positive. I feel they would keep the score of 8.

---

### Decision · Program_Chairs · 2026-01-26

Accept (Poster)